# When Kernels Multiply, Clusters Unify: Fusing Embeddings with the Kronecker Product

**Youqi Wu**[*]
yqwu24@cse.cuhk.edu.hk

**Jingwei Zhang**[*]
jwzhang22@cse.cuhk.edu.hk

**Farzan Farnia**[*]
farnia@cse.cuhk.edu.hk

## Abstract

State-of-the-art embeddings often capture distinct yet complementary discriminative features: For instance, one image embedding model may excel at distinguishing fine-grained textures, while another focuses on object-level structure. Motivated by this observation, we propose a principled approach to fuse such complementary representations through *kernel multiplication*. Multiplying the kernel similarity functions of two embeddings allows their discriminative structures to interact, producing a fused representation whose kernel encodes the union of the clusters identified by each parent embedding. This formulation also provides a natural way to construct *joint kernels* for paired multi-modal data (e.g., image–text tuples), where the product of modality-specific kernels inherits structure from both domains. We highlight that this kernel product is mathematically realized via the *Kronecker product* of the embedding feature maps, yielding our proposed *KrossFuse* framework for embedding fusion. To address the computational cost of the resulting high-dimensional Kronecker space, we further develop *RP−KrossFuse*, a scalable variant that leverages random projections for efficient approximation. As a key application, we use this framework to bridge the performance gap between cross-modal embeddings (e.g., CLIP, BLIP) and unimodal experts (e.g., DINOv2, E5). Experiments show that RP−KrossFuse effectively integrates these models, enhancing modality-specific performance while preserving cross-modal alignment. The project code is available at https://github.com/yokiwuuu/KrossFuse.

## 1 Introduction

The representation learning literature features a wide range of embedding models, each excelling at distinct and often complementary discriminative features. For example, one image encoder may specialize in distinguishing fine-grained categories such as dog breeds, while another captures broader semantic distinctions such as traffic signs. This contrast is illustrated in Figure 1: the DINOv2 image embedding [53] tends to form clearer clusters for dog breeds but shows less separation among traffic signs, whereas the CLIP image embedding [60] exhibits the opposite trend, achieving better separation for traffic signs while mixing the two dog breeds in the kernel heatmaps. Such observations motivate the following question: *how can we systematically fuse multiple embeddings to obtain a single representation that combines the discriminative strengths of all its parent embeddings?*

In this work, we view each embedding as inducing a *kernel similarity function* that assigns a similarity score to every pair of samples. Evaluating this function over a reference dataset produces a *kernel similarity structure*—a matrix that reflects how the embedding perceives relationships among samples

---

[*]Department of Computer Science & Engineering, The Chinese University of Hong Kong

39th Conference on Neural Information Processing Systems (NeurIPS 2025).

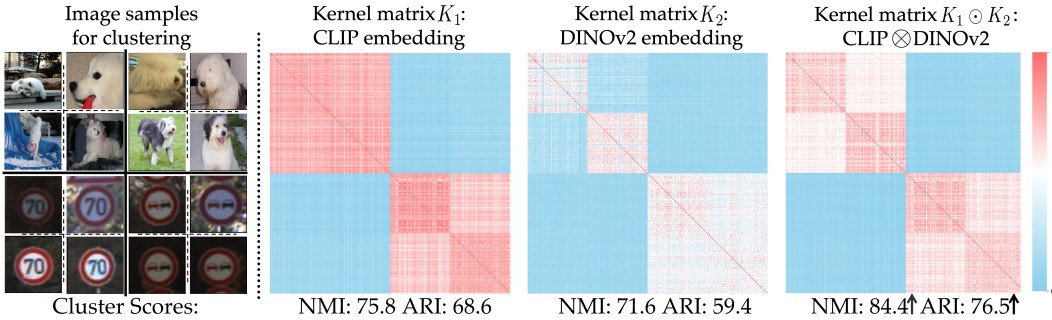

| Image samples for clustering | Kernel matrix $K_1$: CLIP embedding | Kernel matrix $K_2$: DINOv2 embedding | Kernel matrix $K_1 \odot K_2$: CLIP $\otimes$ DINOv2 |
|---|---|---|---|
| Cluster Scores: | NMI: 75.8 ARI: 68.6 | NMI: 71.6 ARI: 59.4 | NMI: 84.4↑ ARI: 76.5↑ |

Figure 1: Heatmaps of RBF kernel similarity matrices for an image dataset with four groundtruth clusters (two dog classes in ImageNet and two traffic sign classes in GTSRB) (left) $K_1$ for CLIP, (middle) $K_2$ for DINOv2, (right) $K_1 \odot K_2$ elementwise product for CLIP and DINOv2's Kronecker product. Unlike CLIP and DINOv2, their Kronecker product could cluster the four image classes.

and which samples it tends to cluster together. This kernel-based view allows us to analyze and combine embeddings directly in the kernel space, where our goal is to construct a fused representation whose similarity structure reflects a *strict union* of the parent cluster structures: two samples should appear similar only if *all* parent embeddings agree that they are similar.

A natural and principled way to capture this "all must agree" logic is through *kernel multiplication*, defining the fused kernel $k_{\psi_{\text{fuse}}}$ as the product of the individual kernel similarity functions $k_{\psi_1}$ and $k_{\psi_2}$ of embeddings $\psi_1$ and $\psi_2$:

$$k_{\psi_{\text{fuse}}}(x,y) \;=\; k_{\psi_1}(x,y) \cdot k_{\psi_2}(x,y). \tag{1}$$

This formulation provides a simple yet effective mechanism for combining the discriminative patterns of multiple embeddings: assuming normalized kernel similarity scores bounded by 1, the fused similarity becomes small whenever any parent embedding separates the two samples[2]. The empirical effect of this operation is illustrated in Figure 1 (right), where multiplying the CLIP and DINOv2 kernels clearly separates all four classes that each model alone fails to isolate.

A feature representation whose kernel similarity function equals the product of the individual kernels is obtained by taking the *Kronecker product* of the individual embedding feature maps, which is well known in the kernel methods literature [65, 7]. Building on this insight, we introduce *KrossFuse*, a general framework for embedding fusion based on Kronecker-product embeddings. Unlike simple concatenation of embedding feature vectors, which corresponds to the summation of individual kernels and lacks the discussed cluster unification property, KrossFuse yields a representation whose kernel unifies sample clusters across parent embeddings' similarities as displayed in Figures 1,2.

Beyond fusing representations within a single modality, KrossFuse also offers a principled way to construct *joint kernels for multi-modal data* (e.g., image–text tuples). Such joint kernels are applicable to generative and retrieval systems (e.g., text-to-image models) that rely on paired data for training and evaluation, where the product of modality-specific kernels captures the joint structure across domains. Figure 2 illustrates this effect in a text-to-image generation setting with SD-XL Turbo model [64]: while the kernel structures of the DINOv2 image, CLIP text, and their concatenated feature vectors fail to clearly separate the six clusters defined by three professions (firefighter, chef, police officer) and two genders (male, female), their *Hadamard product*—using the kernel of the Kronecker-fused embeddings—can distinctly separate all six groups.

While conceptually elegant, the Kronecker formulation incurs a significant computational cost: the dimensionality of the fused embedding equals the product of the input dimensions (for instance, fusing 512-dimensional CLIP and 768-dimensional DINOv2 embeddings yields a 393,216-dimensional vector). To address this scalability barrier, we propose *RP-KrossFuse*, a random-projection-based extension that efficiently approximates the Kronecker feature space while preserving kernel similarities. This approach retains the theoretical properties of KrossFuse while making it practical for large-scale, high-dimensional embeddings.

---

[2]For non-negative kernels such as the Gaussian RBF or normalized even-degree polynomial kernels, the near-zero kernel similarity score directly implies separation, whereas for kernels that can take negative values (e.g., linear or cosine similarity kernels), the fusion operates through orthogonality—if two samples are nearly orthogonal in any parent space, their fused similarity remains near zero, thus preserving separation.

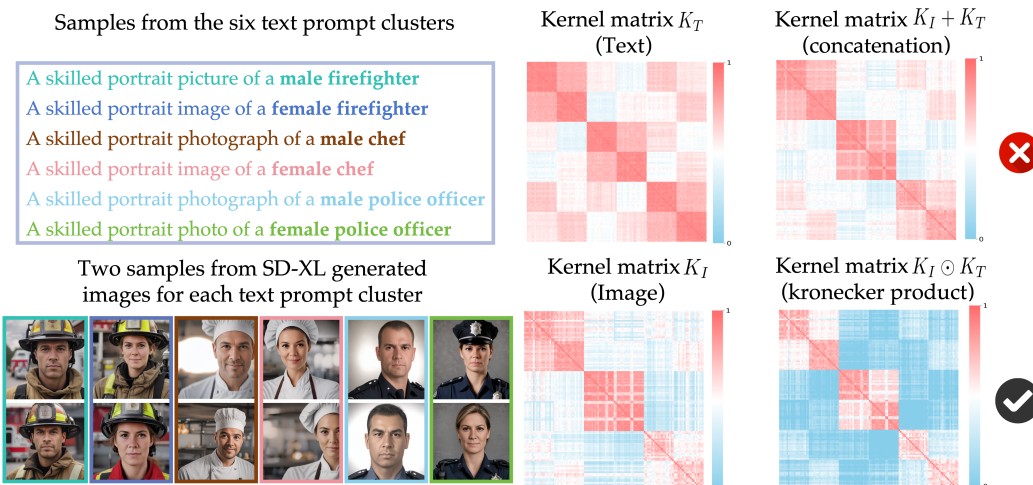

Figure 2: Kernel similarity heatmaps for (text,image) data with 6 underlying clusters. While the kernel matrix of the concatenated CLIP text and DINOv2 image embeddings blur cluster boundaries, the kernel matrices' Hadamard product (for Kronecker-fused embedding) separates all the 6 groups.

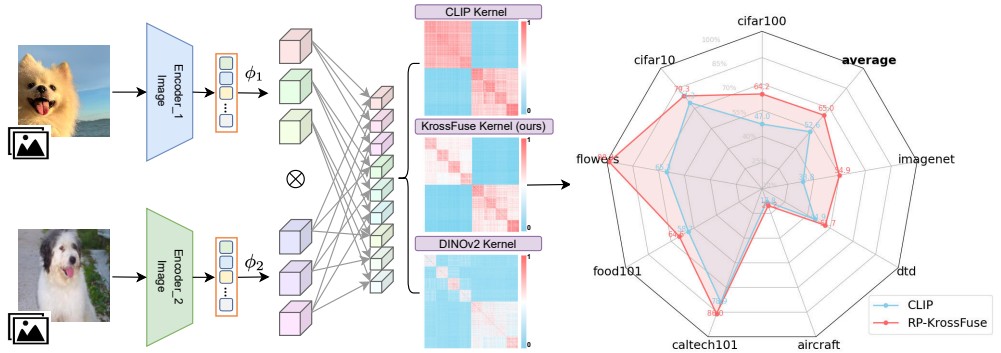

Figure 3: The Kronecker product fusion of embeddings in our proposed KrossFuse: The RP-KrossFuse fusion (implemented with Random Projection) of CLIP and DINOv2 could improve the averaged few-shot classification accuracy over CLIP on 9 benchmark image datasets.

We further apply the KrossFuse framework to address the performance gap between cross-modal and uni-modal embeddings. Cross-modal models such as CLIP, ALIGN [32], and BLIP [38] achieve alignment across modalities but often lag behind modality-specific experts such as DINOv2 and E5 [74] on domain-specific benchmarks. The embedding fusion in this setting poses a significant challenge: the uni-modal expert lacks an embedding map for the other modality (e.g., DINOv2 provides no text encoder). We demonstrate that KrossFuse naturally extends to this case by defining a *symmetrized embedding* for the shared modality and an *imputed constant map* for the missing modality, ensuring that the Kronecker product remains well-defined and balanced across domains.

We perform several experiments to demonstrate that RP-KrossFuse effectively fuses CLIP with uni-modal image (DINOv2) and text (Sentence-RoBERTa) embeddings, achieving competitive modality-specific accuracy while preserving strong cross-modal alignment. For example, Figure 3 shows the improved results of the KrossFuse fusion of DINOv2 and CLIP in few-shot classification over the CLIP model. Our results suggest the RP-KrossFuse framework provides a scalable, training-free mechanism to unify embedding structures. In summary, our contributions are:

- We propose *KrossFuse*, an embedding fusion framework that applies the principle of *kernel multiplication* to unify the cluster structures of embeddings.

- We develop *RP-KrossFuse*, a scalable, random-projection-based extension that efficiently approximates the Kronecker feature space.

- We apply KrossFuse to fuse cross-modal and uni-modal embeddings, demonstrating that it can enhance modality-specific performance while preserving cross-modal alignment.

## 2 Related Work

**Cross-modal Embeddings.** Recent advances in cross-modal embeddings have bridged the gap between visual and textual modalities. CLIP [60] is a pioneer model in this field using a contrastive learning approach, enabling remarkable zero-shot capabilities. Subsequent models, such as ALIGN [32] and FLORENCE [79], scaled datasets and refined learning objectives for improved performance. Variants like CoCa [78] introduced captioning objectives, while BLIP and BLIP-2 [38, 39] employed bootstrapping techniques to generate synthetic image-text pairs. Recent works, including OpenFlamingo [5], PaLI [10] and ImageBind [18], extended capabilities to few-shot scenarios and multilingual, multi-modal scaling. However, a trade-off could exist: strong cross-modal alignment often comes at the price of relatively lowering the performance in single modalities. Our work focuses on this trade-off and explores fusing cross-modal and uni-modal strengths into a unified embedding.

**Combining Representation of Embeddings.** Unifying the strengths of different embedding models has been studied in the representation learning literature. The EVA [15] and X-CLIP [45] methods enhance modality-specific performance while preserving cross-modal capabilities. Bertalign [44] aligns multilingual embeddings via parallel corpus supervision, and ImageBind [18] unifies six modalities using image-paired data. VLMo [6] balances modality-specific traits with cross-modal interaction, while FLAVA [67] and UniCLIP [37] propose unified frameworks for joint representation learning. Different from these training-based strategies, our method offers a training-free approach by applying the Kronecker product of the involved embeddings. By leveraging the symmetrized Kronecker product, our proposed KrossFuse aims to fuse cross-modal and uni-modal embeddings, preserving alignment while improving single-modality performance. To our knowledge, the task of fusing cross-modal and uni-modal embeddings has not been explored exclusively in the literature.

**Random Projections and Kronecker Products.** Random projection is a well-established method for dimensionality reduction. The Johnson-Lindenstrauss (JL) lemma [34] guarantees distance preservation in random projection with high probability. Sparse random projections [1] and fast JL transforms [2] have been shown to further improve the computational efficiency. The Kronecker product combined with random projection has not been utilized in embedding fusion in the context of cross-modal embeddings. In the existing literature, random projections are applied separately before computing the Kronecker product. We unify the two operations in the RP-KrossFuse approach by using the Hadamard product of random projected embeddings and demonstrate the approximation of the kernel matrix of the Kronecker product output.

**Kernel Embedding Methods for Generative AI.** Kernel embeddings are used for evaluation and guidance of generative models. The Kernel Inception Distance (KID) [8] introduced kernel-based evaluation, followed by entropy-based measures including diversity metrics [17, 55, 17, 31, 81, 54, 56, 82]. Kernel formulations are extended to distributed [75] and online evaluation settings [26, 25, 63], as well as explainable [29, 20, 19] and diversity-guided [30] embedding. It is relevant to explore how *KrossFuse* could be extended to these frameworks through its product-kernel formulation.

## 3 Preliminaries

### 3.1 Kernel Functions and Feature Maps

Consider a data vector $x \in \mathcal{X}$. A function $k : \mathcal{X} \times \mathcal{X} \rightarrow \mathbb{R}$ is called a kernel function if for every $n \in \mathbb{N}$ and set of points $x_1, \ldots, x_n \in \mathcal{X}$, the following kernel matrix $K_X \in \mathbb{R}^{n \times n}$ will be positive semi-definite (PSD):

$$K_X := \begin{bmatrix} k(x_1, x_1) & \cdots & k(x_1, x_n) \\ \vdots & \ddots & \vdots \\ k(x_n, x_1) & \cdots & k(x_n, x_n) \end{bmatrix}$$

The Moore-Aronszajn Theorem implies that $k$ is a kernel function if there is a feature map $\phi : \mathcal{X} \rightarrow \mathbb{R}^s$ such that $k(x, y) = \langle \phi(x), \phi(y) \rangle$ is the inner product (denoted by $\langle \cdot, \cdot \rangle$) of the representations $\phi(x)$ and $\phi(y)$. An example is the linear kernel $k_{\mathrm{lin}}(x, y) = x^\top y$ provided by the standard inner product. Another example is the Gaussian (RBF) kernel function with bandwidth parameter $B > 0$:

$$k(x, y) = \exp\left(-\frac{\left\|x - y\right\|_2^2}{B}\right)$$

The Schur product theorem shows that for every two kernel functions $k_1$, $k_2$, their product $k(x, y) := k_1(x, y)k_2(x, y)$ will also be a kernel function. It can be seen that the feature map $\phi$ of the product kernel $k$ is the Kronecker product of the feature maps $\phi_1$, $\phi_2$ of the kernels $k_1, k_2$, i.e.:

$$\phi(x) = \phi_1(x) \otimes \phi_2(x)$$

In the above $\otimes$ denotes the Kronecker product which for matrix $A \in \mathbb{R}^{m \times d}$ with entries $a_{i,j}$'s and matrix $B \in \mathbb{R}^{s \times l}$ is defined as:

$$A \otimes B := \begin{bmatrix} a_{1,1}B & \cdots & a_{1,d}B \\ \vdots & \ddots & \vdots \\ a_{m,1}B & \cdots & a_{m,d}B \end{bmatrix} \in \mathbb{R}^{ms \times dl}$$

### 3.2 Cross-Modal Embedding Maps and Kernel Spaces

Consider two data domains $\mathcal{X}$ (e.g. for image modality) and $\mathcal{T}$ (e.g. for text modality). We call an embedding map $\psi = (\psi_X, \psi_T)$ cross-modal for these two domains if it offers modality-specific maps $\phi_X : \mathcal{X} \to \mathcal{Z}$ and $\phi_T : \mathcal{T} \to \mathcal{Z}$ that respectively map a vector $x \in \mathcal{X}$ in the first domain and a vector $t \in \mathcal{T}$ in the second domain to a shared embedding space $\mathcal{Z}$, i.e. we have $\psi_X(x), \psi_T(t) \in \mathcal{Z}$. The standard cross-modal embeddings are trained such that their outputs for relevant paired data point $(x, t)$ is properly aligned. This property can be mathematically formulated as for a proper kernel function $k : \mathcal{Z} \times \mathcal{Z} \to \mathbb{R}$ the kernel function $k(\psi_X(x), \psi_T(t))$ is supposed to attain high values for relevant paired sample $(x, t) \sim P_{X,T}$ drawn from a ground-truth joint distribution $P_{X,T}$.

## 4 KrossFuse: Kronecker Fusion of Embeddings

### 4.1 Fusing Uni-modal Embeddings using their Kronecker Product

Consider two uni-modal embedding maps $\gamma_1 : \mathcal{X} \to \mathcal{Z}_1$ and $\gamma_2 : \mathcal{X} \to \mathcal{Z}_2$. To fuse $\gamma_1$ and $\gamma_2$, we analyze the kernel similarity functions of the two embeddings. Considering kernel functions $k_1 : \mathcal{Z}_1 \times \mathcal{Z}_1 \to \mathbb{R}$ and $k_2 : \mathcal{Z}_2 \times \mathcal{Z}_2 \to \mathbb{R}$ operating in the embedding spaces, each of the embeddings $\gamma_1$ and $\gamma_2$ provide a kernel function for inputs $x, y \in \mathcal{X}$:

$$k_{\gamma_1}(x, y) = k_1(\gamma_1(x), \gamma_1(y)), \quad k_{\gamma_2}(x, y) = k_2(\gamma_2(x), \gamma_2(y)). \tag{2}$$

Note that if $\phi_1, \phi_2$ denote the feature maps of kernels $k_1, k_2$, i.e. $k_1(x, y) = \langle \phi_1(x), \phi_1(y) \rangle$ and $k_2(x, y) = \langle \phi_2(x), \phi_2(y) \rangle$, then the feature map of $k_{\gamma_1}$ and $k_{\gamma_2}$ will be $\phi_1 \circ \gamma_1$ and $\phi_2 \circ \gamma_2$, respectively.

In fusing the embeddings, we set the fused kernel function to be the product of the marginal kernel functions $k_{\gamma_1} \cdot k_{\gamma_2}$. As shown in Figure 1, this implies that the similarity score between inputs $x, y$ will be low if either of the kernel functions assign a low similarity score, i.e, that embedding distinguishes the input types. As a result, the inputs $x, y$ will be clustered differently in the kernel-based method if their kernel similarity score is minor according to at least one of the embeddings. In the following proposition, we show that the Kronecker product of the embeddings' feature map $\phi_1 \circ \gamma_1$ and $\phi_2 \circ \gamma_2$ possesses the mentioned kernel product property. We defer the proof of the theoretical statements to the Appendix.

**Proposition 1.** *Consider feature maps $\phi_1 : \mathcal{Z}_1 \to \mathbb{R}^{d_1}$, $\phi_2 : \mathcal{Z}_2 \to \mathbb{R}^{d_2}$ and their corresponding kernel functions $k_1$, $k_2$. Then, given the kernel functions defined in* (2)*, the product kernel function $k_{\gamma_1}(x, y) \cdot k_{\gamma_2}(x, y)$ has the feature map $\phi_{\gamma_1, \gamma_2} : \mathcal{X} \to \mathbb{R}^{d_1 d_2}$ defined using the Kronecker product:*

$$\phi_{\gamma_1, \gamma_2}(x) = \phi_1(\gamma_1(x)) \otimes \phi_2(\gamma_2(x))$$

Therefore, the feature map to combine the two embeddings follows from the Kronecker multiplication of $\phi_1 \circ \gamma_1$ and $\phi_2 \circ \gamma_2$, which maps an input $x \in \mathcal{X}$ to a space of dimension $d_1 d_2$, i.e. the product of the dimensions of maps $\phi_1$ and $\phi_2$.

**Remark 1.** *The discussed Kronecker product combination of two embeddings can be further extended to multiple $m$ embeddings $\gamma_1, \ldots, \gamma_m$. Assuming kernel functions $k_1, k_2, \ldots, k_m$ (with feature maps*

$\phi_1, \ldots, \phi_m$) to operate on the $m$ embedded data, the feature map corresponding to the unified product kernel $\prod_{i=1}^{m} k_i(\gamma_i(x), \gamma_i(y))$ will be

$$\phi_{\gamma_1, \ldots, \gamma_m}(x) = \bigotimes_{i=1}^{k} \phi_i\big(\gamma_i(x)\big) \in \mathbb{R}^{d_1 d_2 \cdots d_m}$$

*The above implies that, using the above feature map, the resulting kernel function could distinguish the dissimilarity of inputs $x, y$, if one of the embeddings can differentiate the two data points.*

## 4.2 Extending KrossFuse for Kronecker Fusion of Uni-modal and Cross-Modal Embeddings

We earlier discussed how to fuse unimodal representations $\gamma_1$ and $\gamma_2$ via their Kronecker product, such that the kernel similarity function of the fused embedding is the product of their kernels. However, the challenge in combining a cross-modal embedding $\psi = (\psi_X, \psi_T)$ operating on two modalities in $\mathcal{X}, \mathcal{T}$ and a uni-modal embedding $\gamma = (\gamma_X)$ of only the modality $\mathcal{X}$ is the missing operator of $\gamma$ to apply to the nons-shared modality $\mathcal{T}$. For example, if we suppose $\psi$ represents the CLIP cross-modal model applying to image $\mathcal{X}$ and text $\mathcal{T}$ domains and $\gamma$ denotes the DINOv2 embedding applying to single image $\mathcal{X}$, then we do not have the text part $\gamma_T$ to multiply to the text part of CLIP model.

To apply the Kronecker product-based fusion of the embeddings, we propose the *KrossFuse* method, where we define the following symmetrized cross-modal embedding $\widetilde{\gamma} = (\widetilde{\phi}_{\gamma,X}, \widetilde{\phi}_{\gamma,T})$ to play the role of the uni-modal embedding $\gamma = (\gamma_X)$ in the Kronecker fusion process:

$$\widetilde{\phi}_{\gamma,X}(x) := \frac{1}{\sqrt{2}} \Big[ \sqrt{\frac{C}{d}} + \phi\big(\gamma_X(x)\big), \ \sqrt{\frac{C}{d}} - \phi\big(\gamma_X(x)\big) \Big]^\top$$

$$\widetilde{\phi}_{\gamma,T}(t) := \sqrt{\frac{C}{2d}} \cdot \underbrace{\big[1, \ldots, 1\big]}_{2d \text{ times}}^\top \tag{3}$$

Here, $C > 0$ is defined as a hyperparameter constant determining the constant similarity score of every two data points in the missing modality. Note that $\phi$ denotes the feature map of the given kernel function for the single-modality embedding, and $d$ denotes the dimension of $\phi$'s output vector.

Given the symmetrized cross-modal embedding $\widetilde{\gamma}$ which also applies to the non-shared modality $\mathcal{T}$, KrossFuse combines the cross-modal embedding $\psi = (\psi_X, \psi_T)$ (e.g. CLIP) and the uni-modal embedding $\gamma$ (e.g. DINOv2) by taking the Kronecker product of $\psi$ and $\widetilde{\gamma}$ in each modality as:

$$E_X(x) := \phi\big(\Psi_X(x)\big) \otimes \widetilde{\phi}_{\gamma,X}(x), \tag{4}$$

$$E_T(t) := \phi\big(\Psi_T(t)\big) \otimes \widetilde{\phi}_{\gamma,T}(t) \tag{5}$$

**Proposition 2.** *Given the combined cross-modal embedding in* (4) *and kernel function* $k(x, y) = \langle \phi(x), \phi(y) \rangle$ *for feature map* $\phi$, *the following inner product will hold for every inputs* $x, x' \in \mathcal{X}$ *from the shared modality and inputs* $t, t' \in \mathcal{T}$ *from the non-shared modality:*

$$\big\langle E_X(x), E_X(x') \big\rangle = k\big(\psi_X(x), \psi_X(x')\big) \Big( C + k\big(\gamma_X(x), \gamma_X(x')\big) \Big),$$

$$\big\langle E_T(t), E_T(t') \big\rangle = C \cdot k\big(\psi_T(t), \psi_T(t')\big),$$

$$\big\langle E_X(x), E_T(t) \big\rangle = C \cdot k\big(\psi_X(x), \psi_T(t)\big).$$

As a result, for the merged KrossFuse embedding, the inner product (i.e. the resulting kernel function) between the transformation of two inputs from either of the modalities will be the product of $C$ and the kernel function of the cross-modal embedding (e.g. CLIP model), except the case of two inputs from the shared embedding where the value will be added to the product of the kernels from both the cross-modal and uni-modal embeddings.

**Remark 2.** *The KrossFuse framework can be similarly applied to fuse a cross-modal embedding with multiple uni-modal embeddings. Such a multi-embedding fusion involves the Kronecker product of the modified embedding for every uni-modal embedding in the unification. Therefore, the KrossFuse algorithm can be applied to fuse each of the modalities of a cross-modal embedding with specialized uni-modal models. For example, each of the text and image embedding of CLIP can be merged with text-specific (e.g. E5) and image-specific (e.g. DINOv2) models.*

---

**Algorithm 1** Kernel Feature Fusion of Two Embeddings

---

1: **Input:** Image samples $\{x_i\}_{i=1}^{N_x}$, First image encoder $\gamma_1$, Second image encoder $\gamma_2$, Kernel maps $\phi_1, \phi_2$, Projected dim $l$
2: $U_1 \sim \text{Uniform}[-\sqrt{3}, \sqrt{3}]^{d_{\phi_1} \times l}/\sqrt{l}$
3: $U_2 \sim \text{Uniform}[-\sqrt{3}, \sqrt{3}]^{d_{\phi_2} \times l}/\sqrt{l}$
4: Initialize $Z^{\text{img}} \in \mathbb{R}^{N_x \times l}$
5: **for** batch $\mathcal{B}_x$ in $\{x_i\}$ **do**
6:     $\psi_{1,\mathcal{B}_x} \leftarrow \phi_1(\gamma_1(\mathcal{B}_x)), \psi_{2,\mathcal{B}_x} \leftarrow \phi_2(\gamma_2(\mathcal{B}_x))$
7:     $Z^{\text{img}}_{\mathcal{B}_x} \leftarrow (\psi_{1,\mathcal{B}_x} U_1) \odot (\psi_{2,\mathcal{B}_x} U_2)$
8: **end for**
9: **Return** $Z^{\text{img}}$

---

## 5   RP-KrossFuse: Scalable Embedding Fusion via Random Projection

Although the Kronecker product in KrossFuse can combine multiple embeddings, the dimension of the merged feature vector will be the product of the dimension of individual models, which would be computationally challenging in standard applications. To lower the computational costs, we propose a scalable application of random projection that can preserve the inner product (kernel function values) with limited feature size. The proposed extension, which we call RP-KrossFuse, applies random projection to each cross-modality embedding (modified cross-modality embedding for an original uni-modal embedding). To do this, for each of the cross-modality embeddings $\psi_1 = (\psi_{X,1}, \psi_{T,1})$ and $\psi_2 = (\psi_{X,2}, \psi_{T,2})$, we generate random matrix a $U_i \in \mathbb{R}^{l \times d_1}$ whose entries are independent random variables with uniform distribution over $[-\sqrt{3}, \sqrt{3}]$, that has unit variance. Then, the RP-KrossFuse embedding of each of inputs $x \in \mathcal{X}$ and $t \in \mathcal{T}$ will be

$$\widetilde{\psi}_X(x) = \frac{1}{\sqrt{l}} \big( U_1 \psi_{1,X}(x) \big) \odot \big( U_2 \psi_{2,X}(x) \big) \tag{6}$$

$$\widetilde{\psi}_T(t) = \frac{1}{\sqrt{l}} \big( U_1 \psi_{1,T}(t) \big) \odot \big( U_2 \psi_{2,T}(t) \big) \tag{7}$$

In the above, $\odot$ denotes the element-wise Hadamard product. This formulation leads to Algorithm 1 for uni-modal fusion and Algorithm 2 for cross-modal fusion. Note that computing the above RP-KrossFuse embeddings requires $\mathcal{O}\big( l(d_1 + d_2) \big)$ operations. Also, the output has $l$ dimensions, which for a properly bounded $l$ will be significantly cheaper than $d_1 d_2$ dimensions in the Kronecker product of the two embedded vectors. Theorem 1 shows the above method can preserve the KrossFuse inner products with high probability.

**Theorem 1.** *Consider $n$ input pairs $(t_i, x_i)_{i=1}^n$. Suppose $\max\{\|\psi_{X,j}(x_i)\|^2, \|\psi_{T,j}(t_i)\|^2\} \leq B$ is norm-bounded for every $j \in \{1,2\}$ and index $i \in \{1, \ldots, n\}$. Then, for any $\delta > 0$, we have with probability at least $1 - \delta$:*

$$\left| \widetilde{\psi}_T(t_i)^\top \widetilde{\psi}_T(t_j) - \psi_{1,2,T}(t_i)^\top \psi_{1,2,T}(t_j) \right| \leq \sqrt{\frac{2B^2 \log(2n^2/\delta)}{l}},$$

$$\left| \widetilde{\psi}_X(x_i)^\top \widetilde{\psi}_X(x_j) - \psi_{1,2,X}(x_i)^\top \psi_{1,2,X}(x_j) \right| \leq \sqrt{\frac{2B^2 \log(2n^2/\delta)}{l}}$$

In the Appendix, we have also proved Theorem 2 on the extension of the random projection approach to infinite-dimensional shift-invariant kernels, e.g., RBF kernels, via the framework of random Fourier features [61]. We defer the discussion of the shift-invariant kernels to the Appendix A.4.

## 6   Numerical Results

To evaluate the effectiveness of our proposed fusion methods, KrossFuse and RP-KrossFuse, we performed several numerical experiments regarding unimodal and cross-modal embedding tasks (see appendix B for implementation details). We tested the proposed and baseline fusion methods in application to the following embedding models: cross-modal (image,text) embeddings of CLIP [60], OpenCLIP [28] and SigLIP [80], image modality embeddings of DINOv2 [53], Unicom [3], and

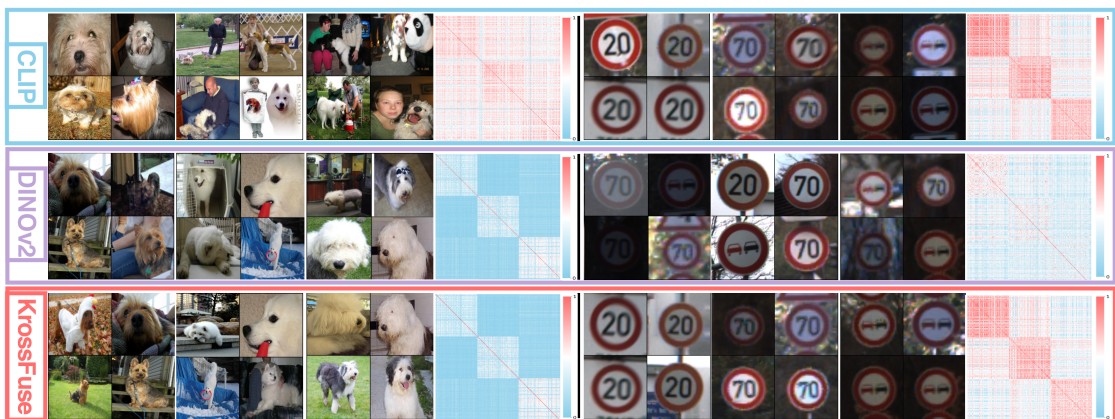

Figure 4: Clustering results and kernel matrix heatmaps for CLIP, DINOv2, and KrossFuse on ImageNet dog breeds and GTSRB dataset. While CLIP could not fully separate all the dog categories and DINOv2 struggled in clustering traffic signs, the KrossFuse fusion captured the clusters.

text modality embeddings of S-RoBERTa [62], E5 [74]. Unless otherwise specified, the additional numerical results are provided in appendix C.

**Visualization of kernel similarity matrix for the Kronecker product of embeddings.** To visualize how the Kronecker product of two embeddings can capture the clusters identified by each of the embeddings, we present the numerical results of performing kernel-based clustering on the following image datasets: CUB-200-2011 [73], Oxford Flowers [51], DTD [11], Image-Woof [23] consisting of ten dog breeds from ImageNet-1K [13], GTSRB [69] and typographic attack images by introducing mislabeled red text into 10 ImageNet subclasses following the reference [47]. The visualizations of the kernel matrices for these datasets highlight how CLIP and DINOv2 could assign differently structured kernel similarity scores in different image types. We present three representative image clustering results, visualized with kernel matrix heatmaps in Figure 4. In the Image-Woof dataset, the DINOv2 similarity map could visually capture the existing three clusters, while the CLIP embedding did not completely separate the three clusters. In contrast, on the three traffic sign classes from GTSRB dataset, the CLIP embedding could differentiate the three groundtruth clusters in its kernel matrix, while the DINOv2 assigned kernel matrix did not display the three clusters. On the other hand, the Kronecker product kernel matrix in KrossFuse shows clear block-diagonal structures on both datasets.

**Unimodal Classification Results.** To assess the performance of the RP-Krossfuse fusion of representations CLIP and DINOv2, we performed linear probe classification on the following standard image datasets: ImageNet [13], GTSRB [69], and SVHN [50], as well as on out-of-distribution (OOD) benchmarks: ImageNet-A [22] and ImageNet-R [21]. The results are shown in Table 1, where KrossFuse is compared with the four baselines (see detailed implementation in appendix B.2: (1) KPoMRP, which utilizes the Kronecker product of marginal random projections; (2) GATE, a simplified implementation of the Mixture-of-Experts (MoE) [66] paradigm; (3) ATTN, a lightweight feature-level attentional fusion method [84]; and (4) COMM, a MLP projector fusion framework proposed in [33]. Notably, RP-KrossFuse obtained 84.1% accuracy on ImageNet, which was improving upon the DINOv2, CLIP image embeddings and also the baselines. While DINOv2 performed competitively on OOD benchmarks, RP-KrossFuse reached better accuracy scores on GTSRB and SVHN. We note that CLIP and DINOv2 would likely excel on certain image categories, and the RP-KrossFuse method seems to consistently fuse the strengths of the two embeddings.

For the text experiments, we used the SentEval toolkit [12] to evaluate the RP-KrossFuse embeddings compared to CLIP and S-RoBERTa on the following NLP classification benchmarks: MR [58], CR [24], SUBJ [57], MPQA [76], SST2 [68], TREC [72], and MRPC [14]. Comprehensive evaluation can be found in table 2.

**Zero-shot Cross-modal Alignment.** To test whether the improved unimodal performance of our fused embeddings comes without compromising CLIP's zero-shot cross-modal alignment, we visualized the cosine similarity distributions of positive and negative image-text pairs on the MSCOCO

Table 1: Linear probe evaluation of embeddings on various image benchmarks (IN: ImageNet). [†]Projection dimension of RP-KrossFuse and KPoMRP: typically 3000 (except 5000 for IN).

| Embedding | IN | GTSRB | SVHN | IN-A | IN-R |
|---|---|---|---|---|---|
| CLIP [60] | 73.2 | 83.1 | 63.6 | 23.2 | 60.0 |
| DINOv2 [53] | 83.3 | 72.5 | 60.5 | 48.5 | 68.8 |
| GATE [66] | 81.4 | 82.2 | 66.0 | 38.9 | 59.1 |
| ATTN [84] | 79.5 | 77.3 | 64.6 | 38.9 | 61.5 |
| KPoMRP[†] | 79.4 | 71.8 | 49.4 | 34.8 | 55.2 |
| COMM [33] | 82.7 | 76.7 | 65.5 | 44.7 | 63.3 |
| RP-KrossFuse[†] | 84.1 | 82.7 | 66.9 | 47.6 | 67.4 |

Table 2: Linear probe evaluation of frozen features of variants of CLIP, Sroberta, RP-KrossFuse and three baselines using the SentEval toolkit on text benchmarks. Test accuracy (%) are based on a 5-fold cross-validation. The projection dimension of KPoMRP and RP-KrossFuse is 3000.

| Embedding | Arch | Fused | MR | CR | SUBJ | MPQA | SST2 | TREC | MRPC | Avg |
|---|---|---|---|---|---|---|---|---|---|---|
| CLIP [60] | ViT-B/32 | ✗ | 75.8 | 83.1 | 92.5 | 86.4 | 82.0 | 83.0 | 70.1 | 81.8 |
| | ViT-L/14 | ✗ | 78.1 | 85.3 | 93.8 | 87.0 | 83.9 | 86.4 | 67.7 | 83.2 |
| KPoMRP | ViT-B/32 | ✔ | 73.8 | 78.5 | 86.6 | 82.1 | 80.3 | 74.8 | 70.8 | 78.1 |
| | ViT-L/14 | ✔ | 72.5 | 80.3 | 86.0 | 83.1 | 74.0 | 78.2 | 68.1 | 77.5 |
| GATE [66] | ViT-B/32 | ✔ | 84.8 | 87.2 | 94.4 | 88.5 | 91.8 | 89.3 | 65.5 | 85.9 |
| | ViT-L/14 | ✔ | 84.3 | 87.8 | 94.5 | 88.3 | 91.4 | 89.3 | 67.6 | 86.2 |
| ATTN [84] | ViT-B/32 | ✔ | 85.7 | 86.3 | 93.4 | 88.8 | 91.9 | 88.5 | 66.2 | 85.8 |
| | ViT-L/14 | ✔ | 85.7 | 85.9 | 94.3 | 87.8 | 92.4 | 86.5 | 65.9 | 85.5 |
| RP-KrossFuse | ViT-B/32 | ✔ | 85.8 | 88.7 | 94.4 | 89.1 | 89.7 | 95.0 | 73.6 | **88.0** |
| | ViT-L/14 | ✔ | 86.0 | 88.1 | 94.8 | 89.3 | 89.8 | 95.2 | 73.6 | **88.1** |
| SRoBERTa [62] | TF-L24 | ✗ | 85.1 | 86.8 | 93.7 | 87.7 | 89.1 | 93.2 | 68.1 | 86.2 |

validation dataset in Figure 5. The overlapping curves suggest that KrossFuse could preserve the geometric distribution of both positive and negative image text pairs. We report zero-shot image-to-text and text-to-image retrieval results on MSCOCO [42] and Flickr30k [77] in table 6 of the appendix. We can observe that the differences between CLIP and RP-KrossFuse are mostly below 1%, suggesting that RP-KrossFuse maintains strong zero shot cross-modal alignment, with retrieval performance comparable to CLIP.

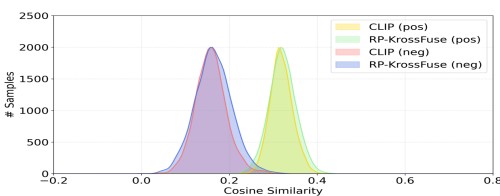

Figure 5: The cosine similarity distributions in MSCOCO.

**Cross-modal Few-shot Learning.** To evaluate how RP-KrossFuse applies to cross-modal few-shot learning scenarios, we used samples from one modality to enhance few shot learning in another modality following the work in [43] which simulates data-limited learning tasks. This cross-modal adaptation task is enabled by multimodal foundation models like CLIP, which map different modalities into a shared representation space, thereby allowing text samples to augment image samples. For generic object and scene image benchmarks, we adopt the standard prompt "a photo of a [class]" for the text modality. To further show the benefits of enhanced text representations, we extend this evaluation to the MSCOCO dataset, where ground-truth captions can be directly used as text samples. As shown in Table 3, our method of cross modality could perform better than CLIP and the other baselines, particularly in 1 and 2 shot cases, highlighting the gains reached by the fusion method in cross-modal transfer learning.

**Ablation Studies.** To test the effect of the different RP-KrossFuse components, we evaluated the classification accuracy on ImageNet and the average accuracy across seven NLP benchmarks in SentEval. As shown in Figure 6: (a) Fusing CLIP with image expert embeddings (UniCom, DINOv2) could also improve performance over the individual CLIP embedding. (b) Fusing text-

Table 3: Cross-modal few-shot classification results across datasets. "Ours" = RP-KrossFuse (proj. dim. 3000); "I"/"T" denote image/text domains.

| Shots | Method | Caltech [16] | Food [9] | DTD [11] | Aircraft [46] | ImageNet [13] | MSCOCO [42] | Average |
|---|---|---|---|---|---|---|---|---|
| 1 | CLIP [60] (I) | 70.9 | 37.8 | 35.4 | 14.6 | 24.3 | 8.7 | 32.0 |
| | CLIP [60] (I+T) | 78.9 | 58.7 | 44.9 | 17.8 | 33.8 | 31.6 | 44.3 |
| | DINOv2 [53] (I) | 84.3 | 57.9 | 47.2 | 15.4 | 54.0 | 16.4 | 45.8 |
| | Ours (I) | 84.6 | 55.7 | 48.3 | 19.4 | 51.8 | 21.5 | 46.9 |
| | Ours (I+T) | **86.0** | **64.6** | **51.7** | **20.3** | **54.9** | **43.5** | **53.5** |
| 2 | CLIP [60] (I) | 78.9 | 47.8 | 44.2 | 18.2 | 30.2 | 11.2 | 38.4 |
| | CLIP [60] (I+T) | 82.7 | 60.7 | 47.3 | 19.8 | 36.0 | 47.2 | 49.0 |
| | DINOv2 [53] (I) | 88.3 | 63.4 | 57.3 | 17.3 | 61.9 | 23.1 | 51.9 |
| | Ours (I) | 89.2 | 63.6 | 57.3 | 23.6 | 60.9 | 36.8 | 55.2 |
| | Ours (I+T) | **90.1** | **68.0** | **59.5** | **24.8** | **62.1** | **51.5** | **59.3** |
| 4 | CLIP [60] (I) | 83.3 | 57.7 | 51.9 | 20.6 | 36.8 | 23.9 | 45.7 |
| | CLIP [60] (I+T) | 84.6 | 64.8 | 52.0 | 21.1 | 42.4 | 57.5 | 53.7 |
| | DINOv2 [53] (I) | 90.4 | 69.8 | 64.0 | 20.9 | 67.0 | 38.5 | 58.4 |
| | Ours (I) | 90.8 | 71.8 | 64.4 | 28.1 | 66.6 | 52.8 | 62.4 |
| | Ours (I+T) | **91.1** | **73.8** | **65.0** | **28.2** | **67.2** | **58.1** | **63.9** |
| 8 | CLIP [60] (I) | 84.5 | 65.5 | 53.7 | 24.2 | 42.1 | 44.9 | 52.5 |
| | CLIP [60] (I+T) | 85.8 | 68.7 | 54.6 | 24.6 | 45.2 | 61.2 | 56.7 |
| | DINOv2 [53] (I) | 91.4 | 73.0 | 69.2 | 24.5 | 70.4 | 53.6 | 63.7 |
| | Ours (I) | 92.0 | 75.9 | 69.2 | **31.7** | 70.4 | 55.1 | 65.7 |
| | Ours (I+T) | **92.2** | **76.9** | **69.4** | **31.7** | **70.7** | **61.9** | **67.1** |

Figure 6: Ablation Studies. (a) (b) Effect of fusing different image and text expert embeddings. (c) Effect of kernel function. (d) Effect of random projected dimension.

expert embeddings (S-RoBERTa, E5) could bring considerable gains compared to the CLIP text encoder. (c) All kernel-based fusions led to performance improvements, with the Cosine and RBF kernels yielding slightly higher gains than the linear kernel. (d) Increasing the random projection dimension would lead to gradually saturated accuracy, suggesting RP-KrossFuse could converge to KrossFuse as the dimension reaches around 3000.

## 7    Conclusion and Limitations

The proliferation of powerful embedding models across vision, language, and other modalities underscores the need for principled methods to unify complementary representations. This work introduced KrossFuse, a Kronecker-product framework for embedding fusion grounded in the kernel product principle, and its scalable variant RP–KrossFuse, which employs random projection to efficiently approximate the high-dimensional Kronecker feature space. The proposed formulation provides a simple, training-free mechanism to integrate embeddings from diverse sources—such as cross-modal and domain-specific models—while preserving the discriminative structure of each.

Despite its efficiency and generality, RP–KrossFuse introduces a few additional hyperparameters, primarily the scaling coefficient $C$ and the projection dimension $l$. The fused embedding may require a higher projection dimension (e.g., 3,000) than individual encoders such as CLIP or DINOv2. While this difference may affect direct dimensional comparability, it aligns with standard practice in representation learning, where embeddings of varying sizes are routinely compared. The use of random projection is a deliberate design choice that enables scalability without the prohibitive cost of operating in the full Kronecker feature space (on the order of $8 \times 10^5$ dimensions). When strict dimensional parity or compactness is desired, post-hoc dimensionality reduction via PCA can be applied to the fused embedding. Future work may extend this framework to non-visual modalities, multi-way fusion scenarios, or attention-based adaptation modules that refine the fused embedding for task-specific objectives.

## Acknowledgments

The work of Farzan Farnia is partially supported by a grant from the Research Grants Council of the Hong Kong Special Administrative Region, China, Project 14209920, and is partially supported by CUHK Direct Research Grants with CUHK Project No. 4055164 and 4937054. The authors acknowledge the support from the Hong Kong Research Grants Council (RGC) and the Hong Kong PhD Fellowship Scheme (HKPFS) award supporting Youqi Wu's research. Also, the authors sincerely thank the anonymous reviewers for their insightful comments and constructive suggestions.

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

## A Proofs

### A.1 Proof of Proposition 1

To show the proposition, we only need to note the following about the inner product of the feature map $\phi_{\gamma_1,\gamma_2}$ for two samples $x, y$:

$$
\begin{aligned}
\phi_{\gamma_1,\gamma_2}(x)^\top \phi_{\gamma_1,\gamma_2}(y) &= \Big( \phi_1\big(\gamma_1(x)\big) \otimes \phi_2\big(\gamma_2(x)\big) \Big)^\top \Big( \phi_1\big(\gamma_1(y)\big) \otimes \phi_2\big(\gamma_2(y)\big) \Big) \\
&= \Big( \phi_1\big(\gamma_1(x)\big)^\top \phi_1\big(\gamma_1(y)\big) \Big) \otimes \Big( \phi_2\big(\gamma_2(x)\big)^\top \phi_2\big(\gamma_2(y)\big) \Big) \\
&= k_{\gamma_1}(x,y) \otimes k_{\gamma_2}(x,y) \\
&= k_{\gamma_1}(x,y) \cdot k_{\gamma_2}(x,y).
\end{aligned}
$$

### A.2 Proof of Proposition 2

To show the proposition, we validate the claimed identities one by one. For the first equation, note that

$$
\begin{aligned}
\langle E_X(x), E_X(x') \rangle &= \Big( \phi\big(\Psi_X(x)\big) \otimes \widetilde{\phi}_{\gamma,X}(x) \Big)^\top \Big( \phi\big(\Psi_X(x')\big) \otimes \widetilde{\phi}_{\gamma,X}(x') \Big) \\
&= \Big( \phi\big(\Psi_X(x)\big)^\top \phi\big(\Psi_X(x')\big) \Big) \otimes \Big( \widetilde{\phi}_{\gamma,X}(x)^\top \widetilde{\phi}_{\gamma,X}(x') \Big) \\
&= k\big(\Psi_X(x), \Psi_X(x')\big) \Big( \widetilde{\phi}_{\gamma,X}(x)^\top \widetilde{\phi}_{\gamma,X}(x') \Big) \\
&= k\big(\Psi_X(x), \Psi_X(x')\big) \Big( \frac{1}{2} \cdot \frac{C}{d} \cdot 2d + \frac{1}{2} \cdot 2 \cdot \phi_{\gamma,X}(x)^\top \phi_{\gamma,X}(x') \Big) \\
&= k\big(\Psi_X(x), \Psi_X(x')\big) \Big( C + k\big(\gamma_X(x), \gamma_X(x')\big) \Big)
\end{aligned}
$$

For the second identity, note that

$$
\begin{aligned}
\langle E_T(t), E_T(t') \rangle &= \Big( \phi\big(\Psi_T(t)\big) \otimes \widetilde{\phi}_{\gamma,T}(t) \Big)^\top \Big( \phi\big(\Psi_T(t')\big) \otimes \widetilde{\phi}_{\gamma,T}(t') \Big) \\
&= \Big( \phi\big(\Psi_T(t)\big)^\top \phi\big(\Psi_T(t')\big) \Big) \otimes \Big( \widetilde{\phi}_{\gamma,T}(t)^\top \widetilde{\phi}_{\gamma,T}(t') \Big) \\
&= k\big(\Psi_T(t), \Psi_T(t')\big) \Big( \widetilde{\phi}_{\gamma,T}(t)^\top \widetilde{\phi}_{\gamma,T}(t') \Big) \\
&= k\big(\Psi_T(t), \Psi_T(t')\big) \Big( \frac{C}{2d} \cdot 2d \Big) \\
&= k\big(\Psi_T(t), \Psi_T(t')\big) \cdot C
\end{aligned}
$$

Finally, for the last identity, we can complete the proof as:

$$
\begin{aligned}
\langle E_X(x), E_T(t) \rangle &= \Big( \phi\big(\Psi_X(x)\big) \otimes \widetilde{\phi}_{\gamma,X}(x) \Big)^\top \Big( \phi\big(\Psi_T(t)\big) \otimes \widetilde{\phi}_{\gamma,T}(t) \Big) \\
&= \Big( \phi\big(\Psi_X(x)\big)^\top \phi\big(\Psi_T(t)\big) \Big) \otimes \Big( \widetilde{\phi}_{\gamma,X}(x)^\top \widetilde{\phi}_{\gamma,T}(t) \Big) \\
&= k\big(\Psi_X(x), \Psi_T(t)\big) \Big( \widetilde{\phi}_{\gamma,X}(x)^\top \widetilde{\phi}_{\gamma,T}(t) \Big) \\
&= k\big(\Psi_T(t), \Psi_T(t)\big) \Big( \frac{1}{2} \cdot \frac{C}{d} \cdot 2d \Big) \\
&= k\big(\Psi_X(x), \Psi_T(t)\big) \cdot C.
\end{aligned}
$$

### A.3 Proof of Theorem 1

To show this statement, we observe that the Kronceker product embedding can be written as:

$$\psi_{1,2,X}(x_i)^\top \psi_{1,2,X}(x_j) = \Big(\psi_{1,X}(x_i) \otimes \psi_{2,X}(x_i)\Big)^\top \Big(\psi_{1,X}(x_j) \otimes \psi_{2,X}(x_j)\Big)$$

$$= \Big(\psi_{1,X}(x_i)^\top \psi_{1,X}(x_j)\Big) \otimes \Big(\psi_{2,X}(x_i)^\top \psi_{2,X}(x_j)\Big)$$

$$= \Big(\psi_{1,X}(x_i)^\top \psi_{1,X}(x_j)\Big) \cdot \Big(\psi_{2,X}(x_i)^\top \psi_{2,X}(x_j)\Big)$$

$$= \mathbb{E}_{\mathbf{u}_1}\Big[\psi_{1,X}(x_i)^\top \mathbf{u}_1^\top \mathbf{u}_1 \psi_{1,X}(x_j)\Big] \cdot \mathbb{E}_{\mathbf{u}_2}\Big[\psi_{2,X}(x_i)^\top \mathbf{u}_2^\top \mathbf{u}_2 \psi_{2,X}(x_j)\Big]$$

$$= \mathbb{E}_{\mathbf{u}_1,\mathbf{u}_2}\Big[\Big(\psi_{1,X}(x_i)^\top \mathbf{u}_1^\top \mathbf{u}_1\big(\psi_{1,X}(x_j)\big)\Big) \cdot \Big(\psi_{2,X}(x_i)^\top \mathbf{u}_2^\top \mathbf{u}_2 \psi_{2,X}(x_j)\Big)\Big]$$

where in the above $\mathbf{u}_1 \in \mathbb{R}^{d_1}$ and $\mathbf{u}_2 \in \mathbb{R}^{d_2}$ are independent random vectors with zero mean and covariance matrix $I_{d_1 \times d_1}$ and $I_{d_2 \times d_2}$. It can be seen that each row of the matrix $U_1$ and $U_2$ with uniformly-drawn entries over $[-\sqrt{3}, \sqrt{3}]$ satisfies this property. As a result, for every row $\mathbf{u}_{1,i}$ of $U_1$ and row $\mathbf{u}_{2,i}$ of $U_2$ (with randomly drawn entries with the specified distribution) we have:

$$\mathbb{E}_{\mathbf{u}_{1,i},\mathbf{u}_{2,i}}\Big[\widetilde{\psi}_{X,\mathbf{u}_{1,i},\mathbf{u}_{2,i}}(x_i)^\top \widetilde{\psi}_{X,\mathbf{u}_{1,i},\mathbf{u}_{2,i}}(x_j)\Big] = \psi_{1,2,X}(x_i)^\top \psi_{1,2,X}(x_j)$$

Note that we have $\widetilde{\psi}_X(x_i)^\top \widetilde{\psi}_X(x_j) = \frac{1}{l}\sum_{i=1}^{l} \widetilde{\psi}_{X,u_{1,i},u_{2,i}}(x_i)^\top \widetilde{\psi}_{X,u_{1,i},u_{2,i}}(x_j)$. On the other hand, we know that each random variable in this empirical mean is bounded as $\big|\widetilde{\psi}_{X,u_{1,2,i}}(x_i)^\top \widetilde{\psi}_{X,u_{1,2,i}}(x_j)\big| \leq \|\widetilde{\psi}_{X,u_{1,2,i}}(x_i)\|_2 \|\widetilde{\psi}_{X,u_{1,2,i}}(x_i)\|_2 \leq B$. Therefore, we can apply Hoeffding's inequality to show that

$$\mathbb{P}\Big(\big|\widetilde{\psi}_X(x_i)^\top \widetilde{\psi}_X(x_j) - \psi_{1,2,X}(x_i)^\top \psi_{1,2,X}(x_j)\big| \geq \epsilon\Big) \leq 2\exp\Big(\frac{-l\epsilon^2}{2B^2}\Big)$$

Setting $\delta = 2n^2 \exp\Big(\frac{-l\epsilon^2}{2B^2}\Big)$ which implies that $\epsilon = \sqrt{\frac{2B^2 \log(2n^2/\delta)}{l}}$, shows that for every $1 \leq i,j \leq n$ we have that

$$\mathbb{P}\Big(\big|\widetilde{\psi}_X(x_i)^\top \widetilde{\psi}_X(x_j) - \psi_{1,2,X}(x_i)^\top \psi_{1,2,X}(x_j)\big| \geq \sqrt{\frac{2B^2 \log(2n^2/\delta)}{l}}\Big) \leq \frac{\delta}{n^2}$$

Thus, applying the union bound to all $n^2$ pairs $(i,j) \in \{1,\ldots,n\}^2$ shows that

$$\mathbb{P}\Big(\forall i,j : \big|\widetilde{\psi}_X(x_i)^\top \widetilde{\psi}_X(x_j) - \psi_{1,2,X}(x_i)^\top \psi_{1,2,X}(x_j)\big| \leq \sqrt{\frac{2B^2 \log(2n^2/\delta)}{l}}\Big) \geq 1-\delta.$$

We can similarly prove the above result for the other modality:

$$\mathbb{P}\Big(\forall i,j : \big|\widetilde{\psi}_T(t_i)^\top \widetilde{\psi}_T(t_j) - \psi_{1,2,T}(t_i)^\top \psi_{1,2,T}(t_j)\big| \leq \sqrt{\frac{2B^2 \log(2n^2/\delta)}{l}}\Big) \geq 1-\delta.$$

This completes the proof.

### A.4 Extending KrossFuse to Infinite-dimension Shift-Invariant Kernels

As we discussed in the main text, a feasible application of KrossFuse requires a feature map $\phi : \mathcal{X} \to \mathbb{R}^s$ mapping to a finite-dimensional space with $s < \infty$. However, this assumption does not apply to popular shift-invariant kernels including the Gaussian (RBF) kernel $k_{\text{gaussian}(\sigma)}(x,y) = \exp\big(-\|x-y\|_2^2/2\sigma^2\big)$ and the Laplace kernel $k_{\text{laplace}(\eta)}(x,y) = \exp\big(-\|x-y\|_1/\eta\big)$.

To extend the KrossFuse application to a general shift-invariant kernel $k(x,y) = \kappa(x-y)$ without any assumption on the finiteness of its feature map, we propose the application of the random Fourier feature (RFF) framework in [61]. Note that if a shift-invariant map $k(x,y) = \kappa(x-y)$ for $\kappa : \mathcal{X} \to \mathbb{R}$ satisfies the positive semi-definite property of a kernel function, Bochner's theorem shows that the Fourier transform $\widehat{\kappa} : \mathbb{R}^d \to \mathbb{R}$ of $\kappa$ will take real non-negative values everywhere, $\widehat{\kappa}(\omega) \geq 0, \forall \omega$. Note that we define the Fourier transform as follows where $\langle \omega, x \rangle$ denotes the standard inner product in the $\mathcal{X}$ space.

$$\widehat{\kappa}(\omega) = \frac{1}{(2\pi)^d}\int_{\mathcal{X}} \kappa(x)\exp\big(-i\langle \omega, x \rangle\big)\mathrm{d}x. \tag{8}$$

Given the above definition, it can be seen that the synthesis equation implies $\kappa(0) = \int_\omega \widehat{\kappa}(\omega)\mathrm{d}\omega$, which means $\widehat{\kappa}$ is a valid probability density function (PDF) for every normalized shift-invariant kernel $k$ satisfying $k(x,x) = \kappa(0) = 1$.

Therefore, the synthesis equation shows that

$$
\begin{aligned}
k(x,y) &= \kappa(x-y) \\
&= \int_\mathcal{X} \widehat{\kappa}(\omega)\exp\big(i\omega^\top(x-y)\big)\mathrm{d}\omega \\
&= \int_\mathcal{X} \widehat{\kappa}(\omega)\cos\big(\omega^\top(x-y)\big)\mathrm{d}\omega \\
&= \mathbb{E}_{\omega\sim\widehat{\kappa}}\Big[\cos\big(\omega^\top(x-y)\big)\Big] \\
&= \mathbb{E}_{\omega\sim\widehat{\kappa}}\Big[\cos\big(\omega^\top x\big)\cos\big(\omega^\top y\big) + \sin\big(\omega^\top x\big)\sin\big(\omega^\top y\big)\Big] \\
&= \mathbb{E}_{\omega\sim\widehat{\kappa}}\Big[\big[\cos\big(\omega^\top x\big),\sin\big(\omega^\top x\big)\big]^\top\big[\cos\big(\omega^\top y\big),\sin\big(\omega^\top y\big)\big]\Big]
\end{aligned}
$$

Note that given a single embedding $\gamma : \mathcal{X} \to \mathbb{R}^d$, the above characterization leads to the standard RFF framework, where for a RFF feature size $r \in \mathbb{N}$, we draw IID random samples $\omega_1, \ldots, \omega_r \sim \widehat{\kappa}$ and define the following RFF proxy map $\phi_r : \mathbb{R}^d \to \mathbb{R}^{2r}$:

$$
\phi_r(z) = \frac{1}{\sqrt{r}}\Big[\cos\big(\omega_1^\top z\big),\sin\big(\omega_1^\top z\big),\ldots,\cos\big(\omega_r^\top z\big),\sin\big(\omega_r^\top z\big)\Big] \tag{9}
$$

However, the application of the RFF framework to the Kroncker product of the embeddings $\gamma_1$ and $\gamma_2$ remains unclear. In this work, we propose a joint sampling of RFF features for the Kronceker product of embeddings. More specifically, suppose we consider shift-invariant kernel functions $k_1(x,x') = \kappa_1(x-x')$ for embedding map $\gamma_1$ and $k_2(x,x') = \kappa_2(x-x')$ for embedding map $\gamma_2$. Then, we consider the joint probability density function $M(\omega_1,\omega_2) = \widehat{\kappa}_1(\omega_1)\cdot\widehat{\kappa}_2(\omega_2)$ for independent variables $\omega_1,\omega_2$.

For applying the RFF framework to the Kronecker product of the embeddings under shift-invaraint kernels, we propose IID sampling $(\omega_1^{(i)},\omega_2^{(i)}) \sim M$, i.e., we draw the $r$ samples jointly, instead of generating them separately for each samples and consider the grid-based pairing of the drawn samples. Given the $r$ drawn samples $(\omega_1^{(i)},\omega_2^{(i)})_{i=1}^r$, we define the following joint RFF feature map:

$$
\begin{aligned}
\widehat{\phi}_r(z_1,z_2) = \frac{1}{\sqrt{r}}\Big[ &\cos\big(z_1^\top\omega_1^{(1)} + z_2^\top\omega_2^{(1)}\big),\sin\big(z_1^\top\omega_1^{(1)} + z_2^\top\omega_2^{(1)}\big), \\
&\ldots,\cos\big(z_1^\top\omega_1^{(r)} + z_2^\top\omega_2^{(r)}\big),\sin\big(z_1^\top\omega_1^{(r)} + z_2^\top\omega_2^{(r)}\big)\Big]
\end{aligned} \tag{10}
$$

Similar to the standard case in the single embedding application of Fourier features, we can prove the following proposition:

**Theorem 2.** *Consider two embedding maps $\gamma_1 : \mathcal{X} \to \mathbb{R}^{d_1}$ and $\gamma_2 : \mathcal{X} \to \mathbb{R}^{d_2}$, applied with shift-invariant kernel similarity functions $k_1 : \mathbb{R}^{d_1} \times \mathbb{R}^{d_1} \to \mathbb{R}$ and $k_2 : \mathbb{R}^{d_2} \times \mathbb{R}^{d_2} \to \mathbb{R}$, respectively. Assume $(\omega_1^{(i)},\omega_2^{(i)})_{i=1}^r$ are drawn independently according to $\widehat{\kappa}_1 \times \widehat{\kappa}_2$, and define the proxy feature map $\widehat{\phi}_r : \mathbb{R}^{d_1} \times \mathbb{R}^{d_2} \to \mathbb{R}^{2r}$ as in (10). Then, for every $x,y \in \mathcal{X}$ and every $\delta > 0$, the following holds with probability at least $1 - \delta$:*

$$
\Big| k_1\big(\gamma_1(x),\gamma_1(y)\big)\cdot k_2\big(\gamma_2(x),\gamma_2(y)\big) - \big\langle\phi_r\big(\gamma_1(x),\gamma_2(x)\big),\phi_r\big(\gamma_1(y),\gamma_2(y)\big)\big\rangle \Big| \leq \sqrt{\frac{2\log(2/\delta)}{r}}
$$

*Proof.* Given that $k_1(x,y) = \kappa_1(x-y)$ is a shift-invariant kernel that satisfies $\kappa_1(0) = 1$, we can deduce that

$$
k_1\big(\gamma_1(x),\gamma_1(y)\big) = \mathbb{E}_{\omega_1\sim\widehat{\kappa}_1}\Big[\cos\big(\omega_1^\top(\gamma_1(x) - \gamma_1(y))\big)\Big].
$$

Similarly, we can observe that $k_2\big(\gamma_2(x), \gamma_2(y)\big) = \mathbb{E}_{\omega_2 \sim \widehat{\kappa}_2}\Big[\cos\big(\omega_2^\top(\gamma_2(x) - \gamma_2(y))\big)\Big]$, which results in

$$
\begin{aligned}
&k_1\big(\gamma_1(x), \gamma_1(y)\big) \cdot k_2\big(\gamma_2(x), \gamma_2(y)\big) \\
&= \mathbb{E}_{\omega_1 \sim \widehat{\kappa}_1}\Big[\exp\big(i\omega_1^\top(\gamma_1(x) - \gamma_1(y))\big)\Big] \cdot \mathbb{E}_{\omega_2 \sim \widehat{\kappa}_2}\Big[\exp\big(i\omega_2^\top(\gamma_2(x) - \gamma_2(y))\big)\Big] \\
&= \mathbb{E}_{(\omega_1, \omega_2) \sim \widehat{\kappa}_1 \times \widehat{\kappa}_2}\Big[\exp\big(i\omega_1^\top(\gamma_1(x) - \gamma_1(y))\big) \cdot \exp\big(i\omega_2^\top(\gamma_2(x) - \gamma_2(y))\big)\Big] \\
&= \mathbb{E}_{(\omega_1, \omega_2) \sim \widehat{\kappa}_1 \times \widehat{\kappa}_2}\Big[\exp\big(i\omega_1^\top(\gamma_1(x) - \gamma_1(y)) + i\omega_2^\top(\gamma_2(x) - \gamma_2(y))\big)\Big] \\
&= \mathbb{E}_{(\omega_1, \omega_2) \sim \widehat{\kappa}_1 \times \widehat{\kappa}_2}\Big[\exp\big(i\big((\omega_1^\top\gamma_1(x) + \omega_2^\top\gamma_2(x)) - (\omega_1^\top\gamma_1(y) + \omega_2^\top\gamma_2(y))\big)\big)\Big] \\
&= \mathbb{E}_{(\omega_1, \omega_2) \sim \widehat{\kappa}_1 \times \widehat{\kappa}_2}\Big[\cos\big((\omega_1^\top\gamma_1(x) + \omega_2^\top\gamma_2(x)) - (\omega_1^\top\gamma_1(y) + \omega_2^\top\gamma_2(y))\big)\Big] \\
&\quad + i \cdot \mathbb{E}_{(\omega_1, \omega_2) \sim \widehat{\kappa}_1 \times \widehat{\kappa}_2}\Big[\sin\big((\omega_1^\top\gamma_1(x) + \omega_2^\top\gamma_2(x)) - (\omega_1^\top\gamma_1(y) + \omega_2^\top\gamma_2(y))\big)\Big] \\
&= \mathbb{E}_{(\omega_1, \omega_2) \sim \widehat{\kappa}_1 \times \widehat{\kappa}_2}\Big[\cos\big((\omega_1^\top\gamma_1(x) + \omega_2^\top\gamma_2(x)) - (\omega_1^\top\gamma_1(y) + \omega_2^\top\gamma_2(y))\big)\Big] \\
&= \mathbb{E}_{(\omega_1, \omega_2) \sim \widehat{\kappa}_1 \times \widehat{\kappa}_2}\Big[\cos\big(\omega_1^\top\gamma_1(x) + \omega_2^\top\gamma_2(x)\big) \cdot \cos\big(\omega_1^\top\gamma_1(y) + \omega_2^\top\gamma_2(y)\big) \\
&\quad + \sin\big(\omega_1^\top\gamma_1(x) + \omega_2^\top\gamma_2(x)\big) \cdot \sin\big(\omega_1^\top\gamma_1(y) + \omega_2^\top\gamma_2(y)\big)\Big].
\end{aligned}
$$

Therefore, since $\big|\cos(\cdot)\big| \leq 1$, the application of Hoeffding's inequality implies that:

$$
\begin{aligned}
&\mathbb{P}\bigg(\Big|k_1\big(\gamma_1(x), \gamma_1(y)\big) \cdot k_2\big(\gamma_2(x), \gamma_2(y)\big) - \big\langle\phi_r\big(\gamma_1(x), \gamma_2(x)\big), \phi_r\big(\gamma_1(y), \gamma_2(y)\big)\big\rangle\Big| \geq \epsilon\bigg) \\
&= \mathbb{P}\bigg(\Big|\mathbb{E}_{(\omega_1, \omega_2) \sim \widehat{\kappa}_1 \times \widehat{\kappa}_2}\Big[\cos\big((\omega_1^\top\gamma_1(x) + \omega_2^\top\gamma_2(x)) - (\omega_1^\top\gamma_1(y) + \omega_2^\top\gamma_2(y))\big)\Big] \\
&\quad - \frac{1}{r}\sum_{j=1}^{r}\Big[\cos\big((\omega_1^{(j)\top}\gamma_1(x) + \omega_2^{(j)\top}\gamma_2(x)) - (\omega_1^{(j)\top}\gamma_1(y) + \omega_2^{(j)\top}\gamma_2(y))\big)\Big]\Big| \geq \epsilon\bigg) \\
&\leq 2\exp\Big(\frac{-2r\epsilon^2}{4}\Big) \\
&= 2\exp\Big(\frac{-r\epsilon^2}{2}\Big)
\end{aligned}
$$

If we let $\delta = 2\exp\big(\frac{-r\epsilon^2}{2}\big)$, i.e., $\epsilon = \sqrt{\frac{2\log(2/\delta)}{r}}$, then we can equivalently write

$$
\begin{aligned}
&\mathbb{P}\bigg(\Big|k_1\big(\gamma_1(x), \gamma_1(y)\big) \cdot k_2\big(\gamma_2(x), \gamma_2(y)\big) - \big\langle\phi_r\big(\gamma_1(x), \gamma_2(x)\big), \phi_r\big(\gamma_1(y), \gamma_2(y)\big)\big\rangle\Big| \\
&\quad \geq \sqrt{\frac{2\log(2/\delta)}{r}}\bigg) \leq \delta
\end{aligned}
$$

which completes the proof. $\qquad\square$

Following Theorem 2, we can extend the random projection fusion of uni-modal embeddings to the settings with shift-invariant kernels that possess an infinite-dimensional feature map. In this case, the fusion will consider the jointly-drawn RFFs according to (10) of embeddings $\gamma_1$ with a shift-invariant kernel $k_1$ and $\gamma_2$ with a shift-invariant kernel $k_2$. This $2r$-dimensional vector will be the proxy fusion of the embedding maps (using the shift-invariant kernel similarity functions $k_1, k_2$). Theorem 1 also proves that the proxy kernel function of this RFF-based fusion is, with high probability, close to the product of marginal kernel functions that is the kernel function of the Kronecker product of the embeddings.

To further extend this discussion to the KrossFuse fusion of the cross-modal $\gamma = (\gamma_X, \gamma_T)$ and uni-modal $\psi_X$, we derive the following formulation. We first generate the RFF features in (10) to obtain the $2r$-dimensional map $\psi_r(x)$ for the shared modality. Extending the RP-KrossFuse

definition to shift-invariant kernels, we obtain the following fused embeddings $\widetilde{\psi}_X : \mathcal{X} \rightarrow \mathbb{R}^{8r}$ and $\widetilde{\psi}_T : \mathcal{T} \rightarrow \mathbb{R}^{8r}$ for the uni-modal embedding $\psi_X$ functioning only only the modality $X$ and cross-modal embedding $\gamma = (\gamma_X, \gamma_T)$:

$$
\begin{aligned}
\widetilde{\psi}_X(x) := \frac{1}{\sqrt{2r}} \Big[ & \sqrt{C}\cos\big(\psi_X(x)^\top \omega_2^{(1)}\big) + \cos\big(\gamma_X(x)^\top \omega_1^{(1)} + \psi_X(x)^\top \omega_2^{(1)}\big), \\
& \sqrt{C}\cos\big(\psi_X(x)^\top \omega_2^{(1)}\big) - \cos\big(\gamma_X(x)^\top \omega_1^{(1)} + \psi_X(x)^\top \omega_2^{(1)}\big), \\
& \sqrt{C}\cos\big(\psi_X(x)^\top \omega_2^{(1)}\big) + \sin\big(\gamma_X(x)^\top \omega_1^{(1)} + \psi_X(x)^\top \omega_2^{(1)}\big), \\
& \sqrt{C}\cos\big(\psi_X(x)^\top \omega_2^{(1)}\big) - \sin\big(\gamma_X(x)^\top \omega_1^{(1)} + \psi_X(x)^\top \omega_2^{(1)}\big), \\
& \sqrt{C}\sin\big(\psi_X(x)^\top \omega_2^{(1)}\big) + \cos\big(\gamma_X(x)^\top \omega_1^{(1)} + \psi_X(x)^\top \omega_2^{(1)}\big), \\
& \sqrt{C}\sin\big(\psi_X(x)^\top \omega_2^{(1)}\big) - \cos\big(\gamma_X(x)^\top \omega_1^{(1)} + \psi_X(x)^\top \omega_2^{(1)}\big), \\
& \sqrt{C}\sin\big(\psi_X(x)^\top \omega_2^{(1)}\big) + \sin\big(\gamma_X(x)^\top \omega_1^{(1)} + \psi_X(x)^\top \omega_2^{(1)}\big), \\
& \sqrt{C}\sin\big(\psi_X(x)^\top \omega_2^{(1)}\big) - \sin\big(\gamma_X(x)^\top \omega_1^{(1)} + \psi_X(x)^\top \omega_2^{(1)}\big) \\
& , \ldots, \\
& \sqrt{C}\cos\big(\psi_X(x)^\top \omega_2^{(r)}\big) + \cos\big(\gamma_X(x)^\top \omega_1^{(r)} + \psi_X(x)^\top \omega_2^{(r)}\big), \\
& \sqrt{C}\cos\big(\psi_X(x)^\top \omega_2^{(r)}\big) - \cos\big(\gamma_X(x)^\top \omega_1^{(r)} + \psi_X(x)^\top \omega_2^{(r)}\big), \\
& \sqrt{C}\cos\big(\psi_X(x)^\top \omega_2^{(r)}\big) + \sin\big(\gamma_X(x)^\top \omega_1^{(r)} + \psi_X(x)^\top \omega_2^{(r)}\big), \\
& \sqrt{C}\cos\big(\psi_X(x)^\top \omega_2^{(r)}\big) - \sin\big(\gamma_X(x)^\top \omega_1^{(r)} + \psi_X(x)^\top \omega_2^{(r)}\big), \\
& \sqrt{C}\sin\big(\psi_X(x)^\top \omega_2^{(r)}\big) + \cos\big(\gamma_X(x)^\top \omega_1^{(r)} + \psi_X(x)^\top \omega_2^{(r)}\big), \\
& \sqrt{C}\sin\big(\psi_X(x)^\top \omega_2^{(r)}\big) - \cos\big(\gamma_X(x)^\top \omega_1^{(r)} + \psi_X(x)^\top \omega_2^{(r)}\big), \\
& \sqrt{C}\sin\big(\psi_X(x)^\top \omega_2^{(r)}\big) + \sin\big(\gamma_X(x)^\top \omega_1^{(r)} + \psi_X(x)^\top \omega_2^{(r)}\big), \\
& \sqrt{C}\sin\big(\psi_X(x)^\top \omega_2^{(r)}\big) - \sin\big(\gamma_X(x)^\top \omega_1^{(r)} + \psi_X(x)^\top \omega_2^{(r)}\big) \Big]
\end{aligned}
$$

and

$$
\begin{aligned}
\widetilde{\psi}_T(t) := \frac{1}{\sqrt{2r}} \Big[ & \sqrt{C} + \cos\big(\gamma_T(t)^\top \omega_1^{(1)}\big), \sqrt{C} - \cos\big(\gamma_T(t)^\top \omega_1^{(1)}\big), \sqrt{C} + \sin\big(\gamma_T(t)^\top \omega_1^{(1)}\big) \\
& , \sqrt{C} - \sin\big(\gamma_T(t)^\top \omega_1^{(1)}\big), \\
& \cos\big(\gamma_T(t)^\top \omega_1^{(1)}\big), -\cos\big(\gamma_T(t)^\top \omega_1^{(1)}\big), \sin\big(\gamma_T(t)^\top \omega_1^{(1)}\big), -\sin\big(\gamma_T(t)^\top \omega_1^{(1)}\big) \\
& , \ldots, \\
& \sqrt{C} + \cos\big(\gamma_T(t)^\top \omega_1^{(r)}\big), \sqrt{C} - \cos\big(\gamma_T(t)^\top \omega_1^{(r)}\big), \sqrt{C} + \sin\big(\gamma_T(t)^\top \omega_1^{(r)}\big) \\
& \sqrt{C} - \sin\big(\gamma_T(t)^\top \omega_1^{(r)}\big), \\
& \cos\big(\gamma_T(t)^\top \omega_1^{(r)}\big), -\cos\big(\gamma_T(t)^\top \omega_1^{(r)}\big), \sin\big(\gamma_T(t)^\top \omega_1^{(r)}\big), -\sin\big(\gamma_T(t)^\top \omega_1^{(r)}\big) \Big]
\end{aligned}
$$

## A.5 Algorithm of Fusing Cross-modal Embeddings and Uni-modal Embeddings

**Algorithm 2** Kernel Feature Fusion of Cross-modal Embeddings and Uni-modal Embeddings

---

1: **Input:** Image samples $\{x_i\}_{i=1}^{N_x}$, Text samples $\{y_j\}_{j=1}^{N_y}$, Cross-modal encoder $\gamma_1$, Uni-image encoder $\gamma_2$ and Uni-text encoder $\gamma_3$, Kernel maps $\phi_1, \phi_2, \phi_3$, Constant $c$, Projected dim $l$
2: $U_i \sim \text{Uniform}[-\sqrt{3}, \sqrt{3}]^{d_{\phi_i} \times l}/\sqrt{l}, \quad \text{for } i = 1, 2, 3.$
3: Initialize $Z^{\text{img}} \in \mathbb{R}^{N_x \times l}$, $Z^{\text{text}} \in \mathbb{R}^{N_y \times l}$
4: **for** batch $\mathcal{B}_x$ in $\{x_i\}$ **do** $\qquad\qquad\qquad\qquad\qquad\qquad\qquad$ ▷ Process Image Samples
5: $\qquad \psi_{1,\mathcal{B}_x} \leftarrow \phi_1(\gamma_1(\mathcal{B}_x)), \psi_{2,\mathcal{B}_x} \leftarrow \phi_2(\gamma_2(\mathcal{B}_x)), \psi_{3,\mathcal{B}_x} \leftarrow c \cdot \mathbf{1}_{|\mathcal{B}_x| \times 2d_{\phi_3}}$
6: $\qquad \psi'_{2,\mathcal{B}_x} \leftarrow \text{concat}(c\mathbf{1} + \psi_{2,\mathcal{B}_x}, c\mathbf{1} - \psi_{2,\mathcal{B}_x})$
7: $\qquad Z^{\text{img}}_{\mathcal{B}_x} \leftarrow (\psi_{1,\mathcal{B}_x} U_1) \odot (\psi'_{2,\mathcal{B}_x} U_2) \odot (\psi_{3,\mathcal{B}_x} U_3)$
8: **end for**
9: **for** batch $\mathcal{B}_y$ in $\{y_j\}$ **do** $\qquad\qquad\qquad\qquad\qquad\qquad\qquad\quad$ ▷ Process Text Samples
10: $\qquad \psi_{1,\mathcal{B}_y} \leftarrow \phi_1(\gamma_1(\mathcal{B}_y)), \psi_{2,\mathcal{B}_y} \leftarrow c \cdot \mathbf{1}_{|\mathcal{B}_y| \times 2d_{\phi_2}}, \psi_{3,\mathcal{B}_y} \leftarrow \phi_3(\gamma_3(\mathcal{B}_y))$
11: $\qquad \psi'_{3,\mathcal{B}_y} \leftarrow \text{concat}(c\mathbf{1} + \psi_{3,\mathcal{B}_y}, c\mathbf{1} - \psi_{3,\mathcal{B}_y})$
12: $\qquad Z^{\text{text}}_{\mathcal{B}_y} \leftarrow (\psi_{1,\mathcal{B}_y} U_1) \odot (\psi_{2,\mathcal{B}_y} U_2) \odot (\psi'_{3,\mathcal{B}_y} U_3)$
13: **end for**
14: **Return** $Z^{\text{img}}, Z^{\text{text}}$

---

# B    Implementation Details

## B.1    Experiments Setup

We tested several pre-trained embeddings in our experiments, including multiple variants of CLIP [60], DINOv2 [53], Unicom [3], Sroberta [62], E5 [74], and Siglip [80]. Unless otherwise specified, we primarily report results in the main text using standard CLIP(ViT-B/32), DINOv2(ViT-B/14), and Sroberta. The projection matrices $U_i$ we generate once using the uniform distribution over $[-\sqrt{3}, \sqrt{3}]$, that has unit variance. The projection dimension $l$ is 3000 for all datasets except 5000 in the case of ImageNet. The parameter $C$ has been determined using cross-valdiation over $\{10^{-3}, 10^{-2}, \ldots, 10^3\}$. All experiments were run on 2 RTX-4090 GPUs.

## B.2    Baselines Setup

Our experiments analyze the fusion of cross-modal and uni-modal representations where we utilize the pretrained expert encoders without additional fine-tuning. In our numerical analysis, we consider the last-layer output for every attempted encoder. To the best of our knowledge, the task of fusing cross-modal and uni-modal embeddings has not been exclusively analyzed in the literature. Therefore, we emphasize that the baseline methods discussed in our analysis are fusion methods proposed for two unimodal embeddings, which cannot be applied to the zero-shot classification tasks.

The baseline methods in our analysis are: (1) *Kronecker Product of Marginal Random Projection (KPoMRP).* This baseline applies an independent random projection separately to each embedding output, followed by a Kronecker product to obtain the fused representation. Note that our proposed RP-KrossFuse functions differently and samples the random features jointly for the output of the two embeddings. (2) *Gated Fusion.* This baseline represents the application of the Mixture-of-Experts (MoE) method discussed in [66], where a gating mechanism dynamically controls the contribution of different feature sources. Specifically, each feature is first passed to an MLP layer, then the outputs are passed through a sigmoid gating function to compute a dynamic fusion weight. (3) *Attentional Fusion.* The baseline follows the self-attentional fusion framework proposed in [84]. Projected features from each encoder are concatenated and passed through a self-attention module to dynamically aggregate information. (4) *COMM.* This baseline follows the fusion framework proposed in [33]. They employ an MLP layer to project the features of DINOv2 and concatenate the output features with that of CLIP.

# C  Additional Experimental Results

## C.1  Unimodal Clustering

We present the complete set of our numerical results of kernel-based clustering on the image datasets: CUB-200-2011 [73], Oxford Flowers [51], DTD [11], Image-Woof [23] consisting of ten dog breeds from ImageNet-1K [13], GTSRB [69] and typographic attack images by introducing mislabeled red text into 10 ImageNet subclasses following the reference [47]. Table 4 reports the clustering performance scores of different methods on six image datasets, evaluated by Normalized Mutual Information [48] (NMI), Adjusted Mutual Information [71] (AMI), and Adjusted Rand Index [27] (ARI). Across all datasets and metrics, KrossFuse consistently performs better than individual CLIP and DINOv2, reaching the highest scores across the datasets. Notably, KrossFuse shows improvements on known challenging dataset cases such as Typo-Attacked ImageNet, ImageNet-Dogs and DTD, indicating its capability in capturing discriminative features for clustering tasks. The following figures further illustrate the detailed clustering results on these datasets. On top of it, we visualize the kernel matrices, as well as the distribution of the embeddings using t-SNE and UMAP, providing qualitative insights into the effectiveness of each method in separating different classes.

Table 4

| Metric | Method | CUB200 | Flowers102 | DTD | ImageNet-Dogs | GTSRB | Typo-Attacked ImageNet |
|--------|--------|--------|------------|------|---------------|-------|------------------------|
| NMI | CLIP | 63.2 | 80.0 | 50.9 | 49.7 | 49.7 | 20.1 |
|  | DINOv2 | 85.2 | 98.7 | 60.5 | 86.7 | 40.2 | 81.9 |
|  | KrossFuse | 85.6 | 99.1 | 62.9 | 88.3 | 50.0 | 87.4 |
| AMI | CLIP | 45.7 | 73.2 | 47.7 | 49.2 | 46.4 | 19.3 |
|  | DINOv2 | 78.3 | 98.2 | 57.8 | 86.6 | 36.2 | 81.7 |
|  | KrossFuse | 79.0 | 98.8 | 60.4 | 88.2 | 46.7 | 87.3 |
| ARI | CLIP | 21.1 | 55.5 | 27.2 | 37.1 | 18.6 | 10.4 |
|  | DINOv2 | 55.9 | 94.6 | 28.1 | 84.8 | 12.3 | 65.9 |
|  | KrossFuse | 56.3 | 97.0 | 36.4 | 86.3 | 19.5 | 79.6 |

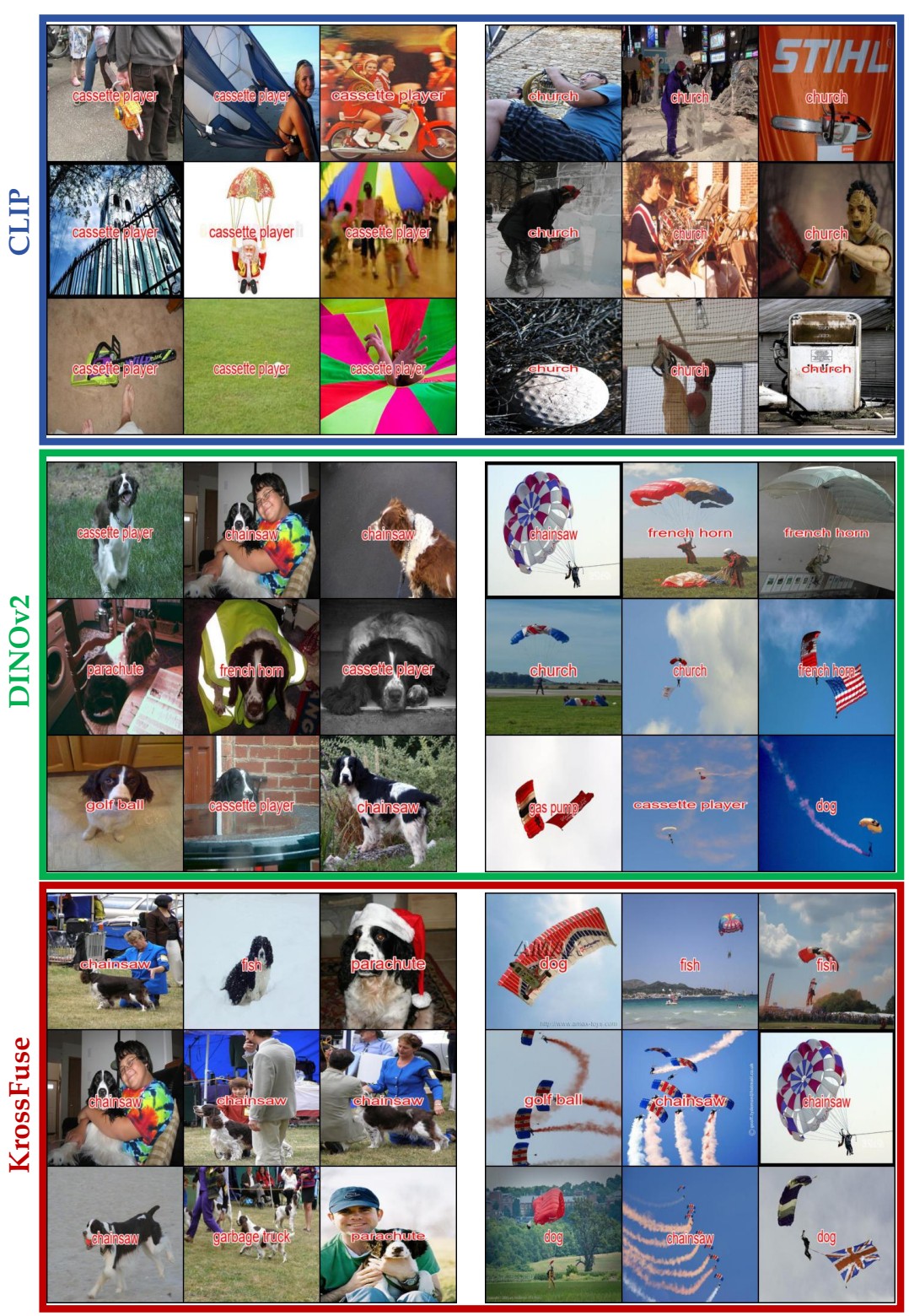

Figure 7: Clustering results of CLIP, DINOv2 and KrossFuse embeddings for typographic attacked images (red text is the misleading labels that simulate the attack) from 10 ImageNet classes. KrossFuse could cluster the attacked image classes like DINOv2 while CLIP is mislead by text.

**Typo-attacked ImageNet Dataset**

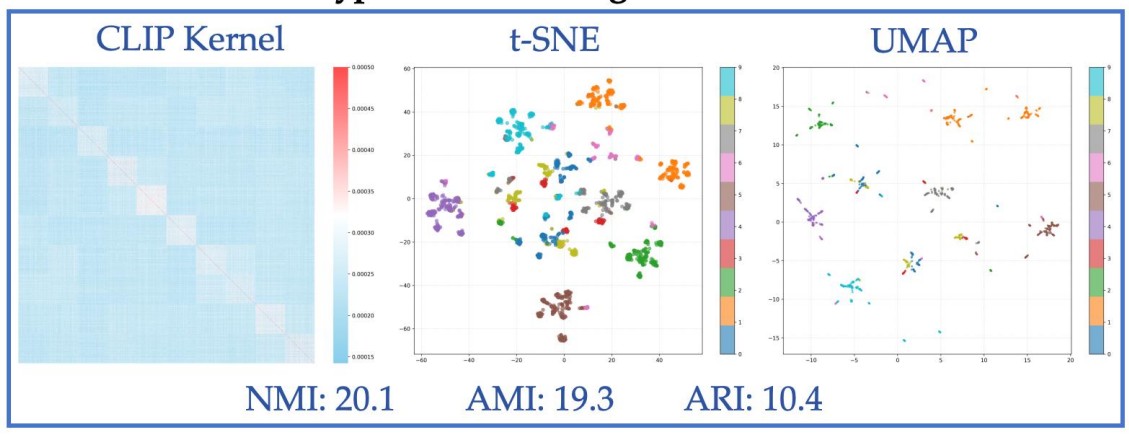

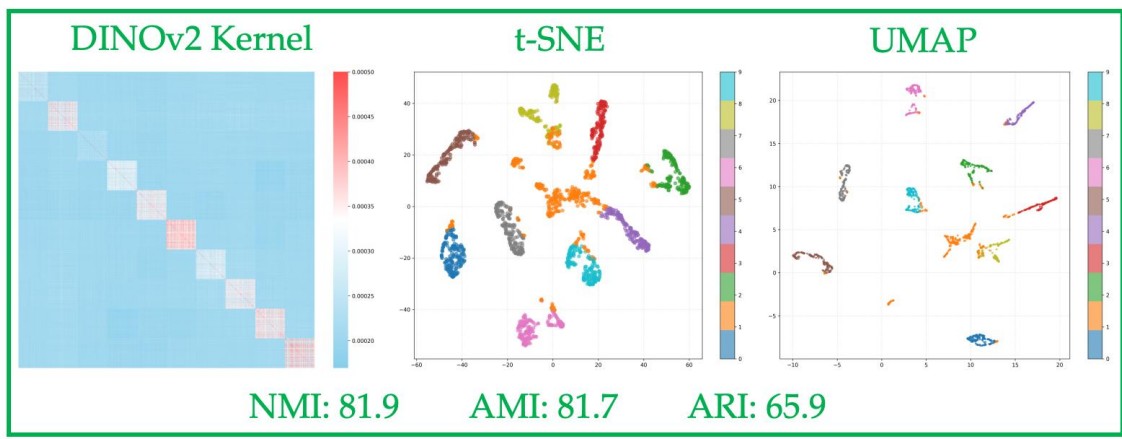

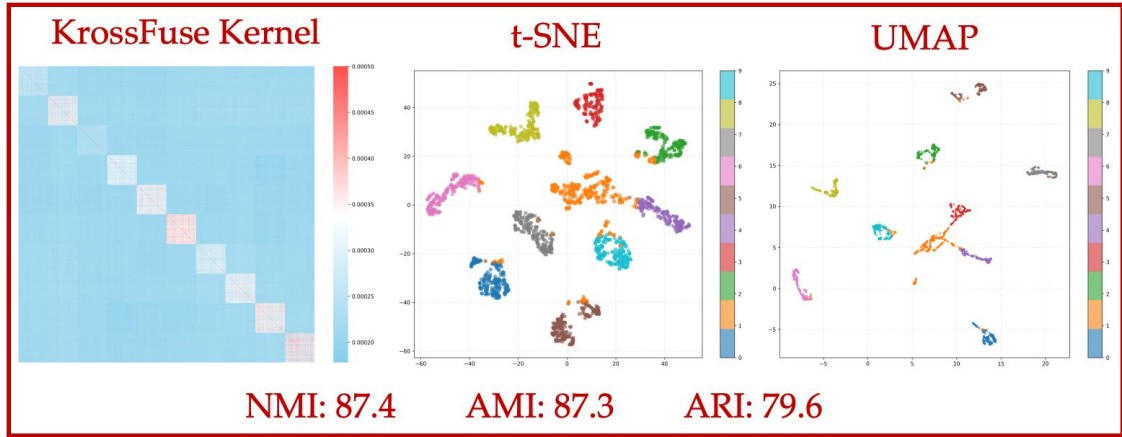

Figure 8: Comparison among CLIP, DINOv2 and KrossFuse embeddings for typographic attacked images from 10 ImageNet classes. (Left) Heatmaps of RBF kernel similarity matrices, (Middle) t-SNE visualization, (Right) UMAP visualization. KrossFuse could cluster the attacked image classes like DINOv2 while CLIP is mislead by text.

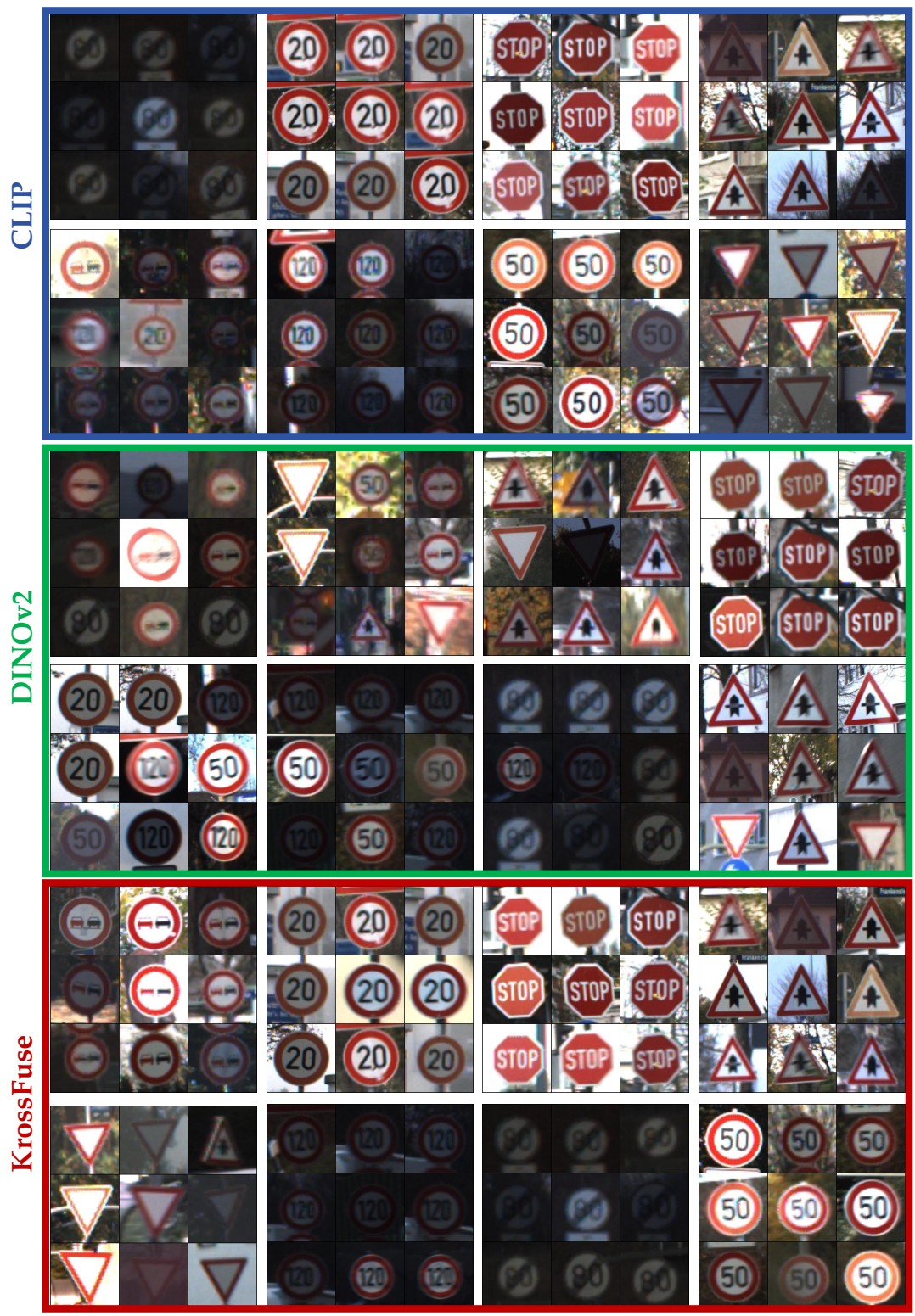

Figure 9: Clustering results of CLIP, DINOv2 and KrossFuse embeddings for GTSRB dataset with eight clusters. KrossFuse could cluster the eight image classes like CLIP while DINOv2 can't distinguish them.

**GTSRB Dataset**

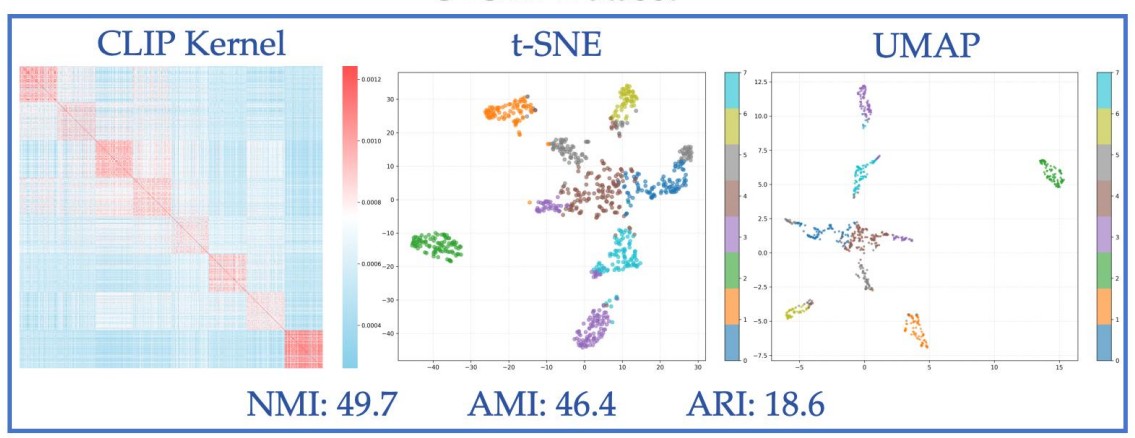

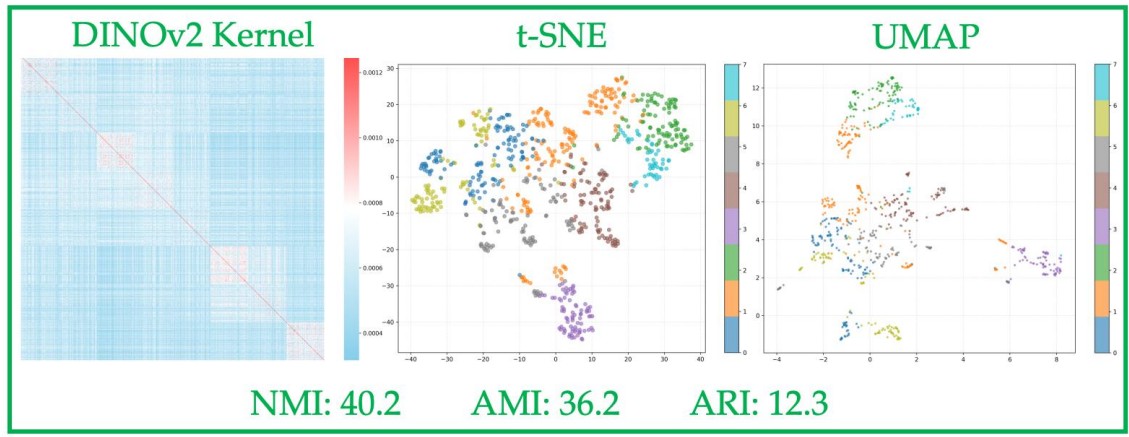

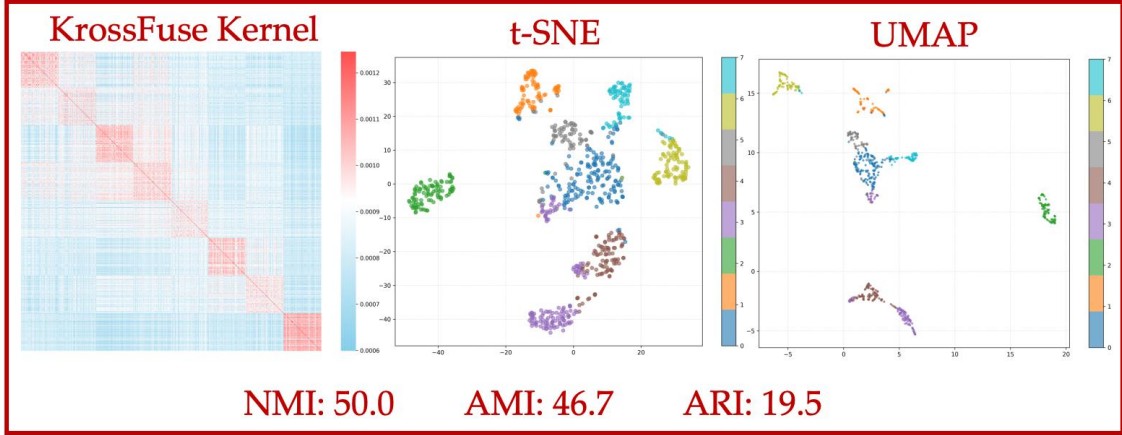

Figure 10: Comparison among CLIP, DINOv2 and KrossFuse embeddings for GTSRB dataset with eight clusters. (Left) Heatmaps of RBF kernel similarity matrices, (Middle) t-SNE visualization, (Right) UMAP visualization. KrossFuse could cluster the eight image classes like CLIP while DINOv2 can't distinguish all of them.

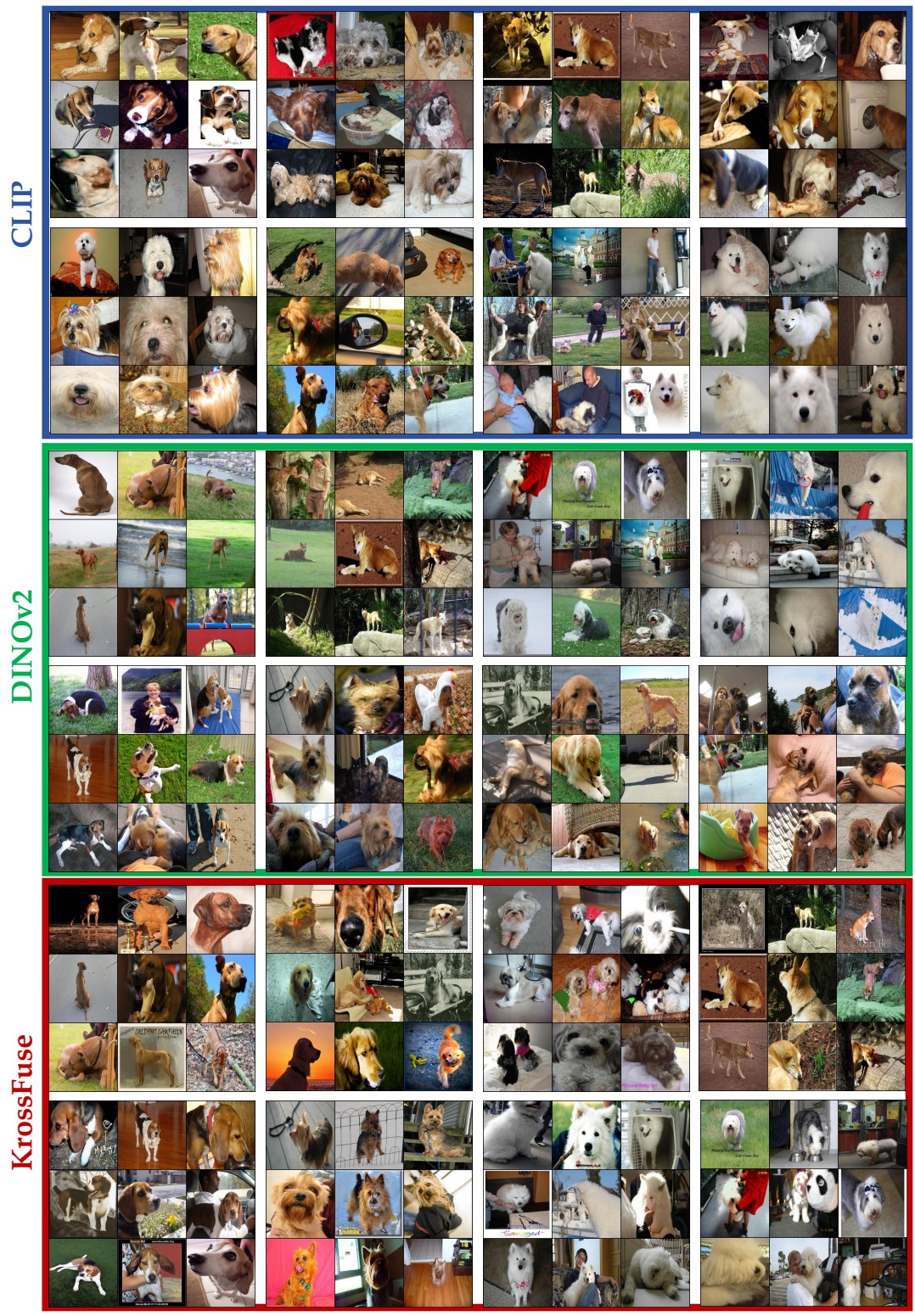

Figure 11: Clustering results of CLIP, DINOv2 and KrossFuse embeddings for ImageNet-dog breeds dataset. KrossFuse could cluster them like DINOv2 while CLIP can't distinguish all of them.

**ImageNet-Dogs Dataset**

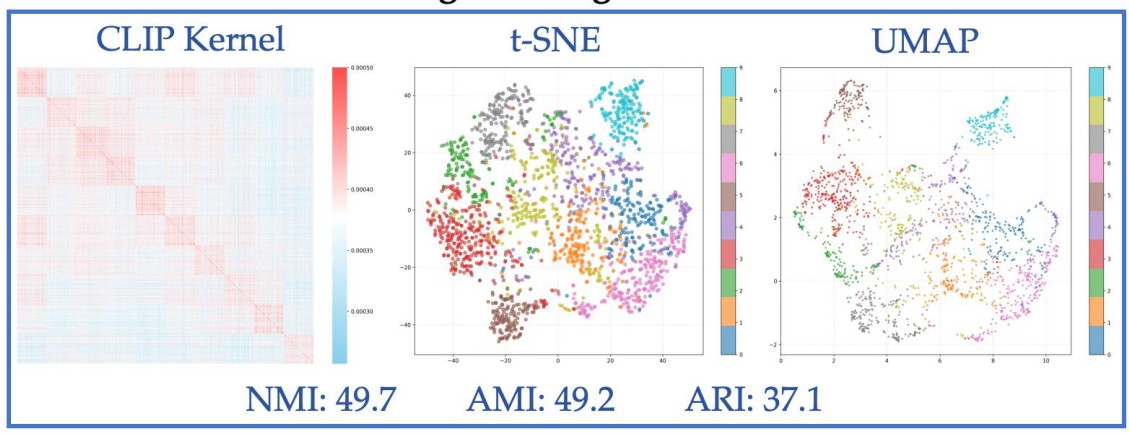

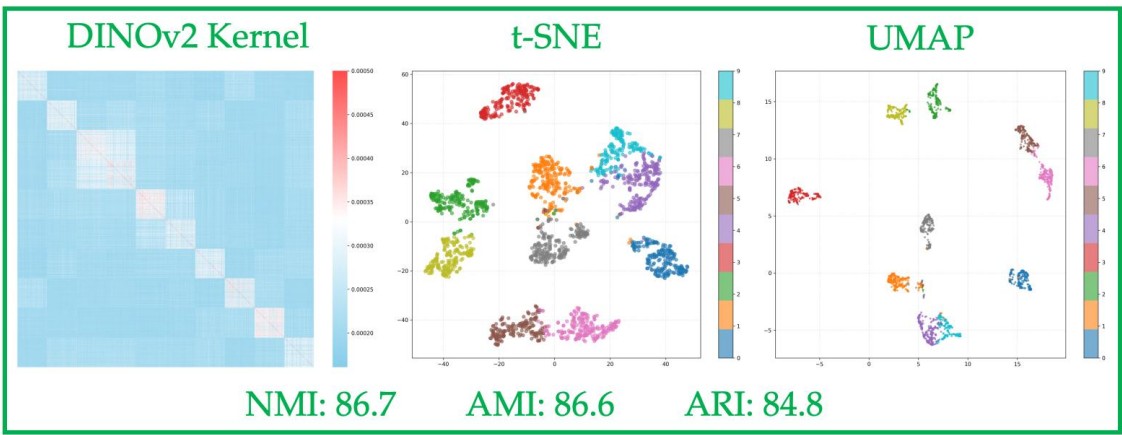

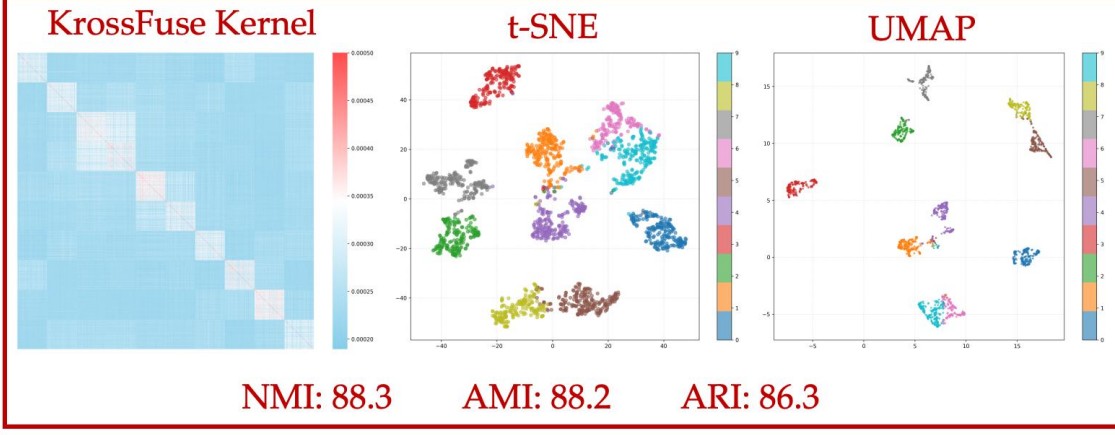

Figure 12: Comparison among CLIP, DINOv2 and KrossFuse embeddings for ImageNet-dog breeds dataset. (Left) Heatmaps of RBF kernel similarity matrices, (Middle) t-SNE visualization, (Right) UMAP visualization. KrossFuse could cluster the different dog classes like DINOv2 while CLIP can't distinguish all of them.

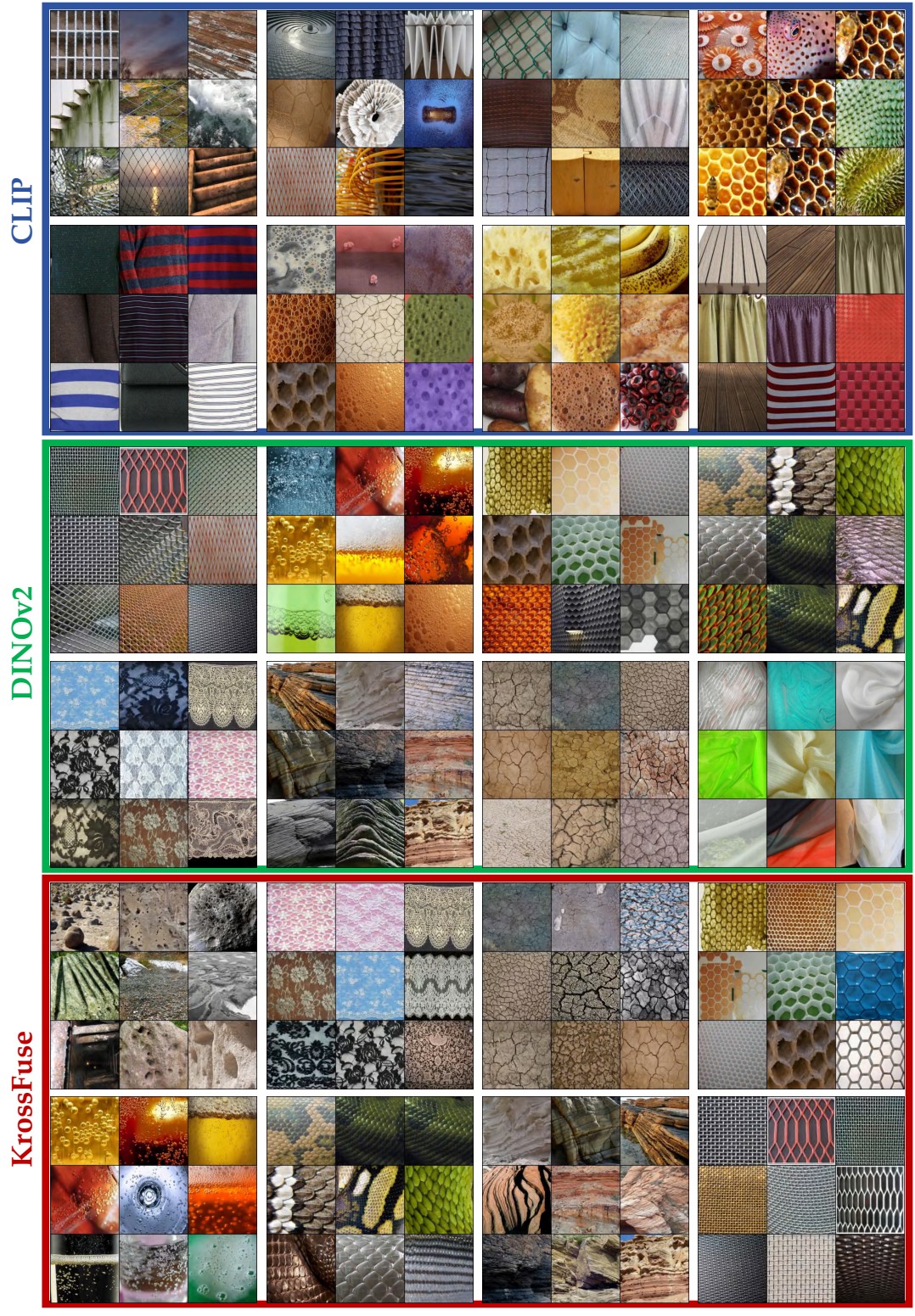

Figure 13: Clustering results of CLIP, DINOv2 and KrossFuse embeddings for DTD dataset. Kross-Fuse could cluster them like DINOv2 while CLIP can't distinguish all of them.

**DTD Dataset**

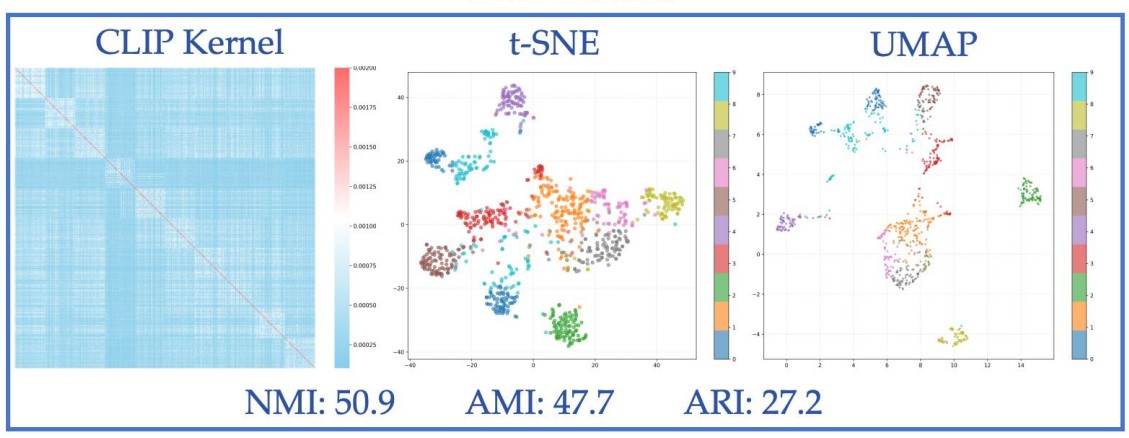

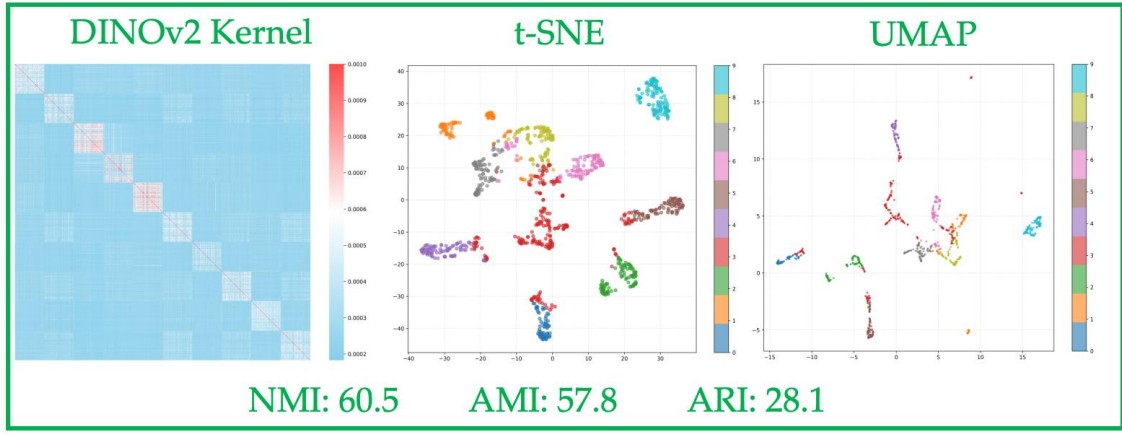

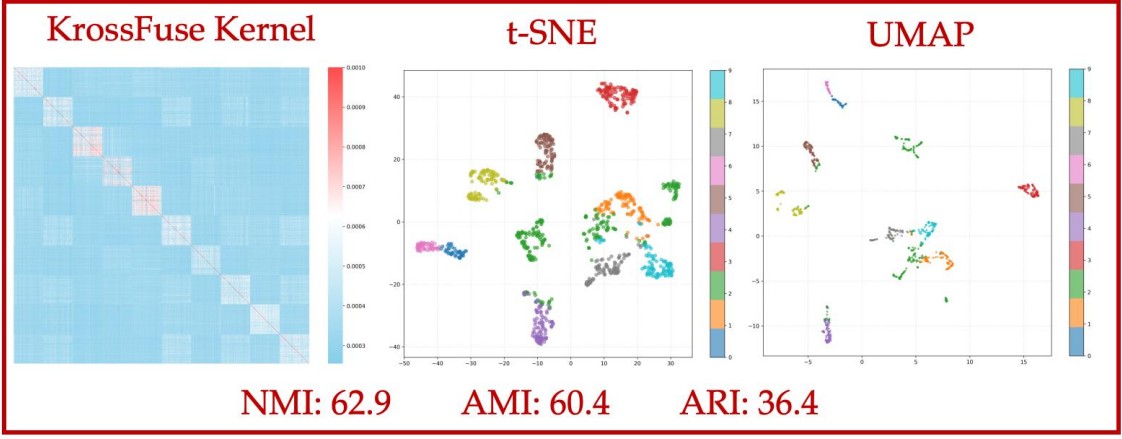

Figure 14: Comparison among CLIP, DINOv2 and KrossFuse embeddings for DTD dataset. (Left) Heatmaps of RBF kernel similarity matrices, (Middle) t-SNE visualization, (Right) UMAP visualization. KrossFuse could cluster the different texture classes like DINOv2 while CLIP can't distinguish all of them.

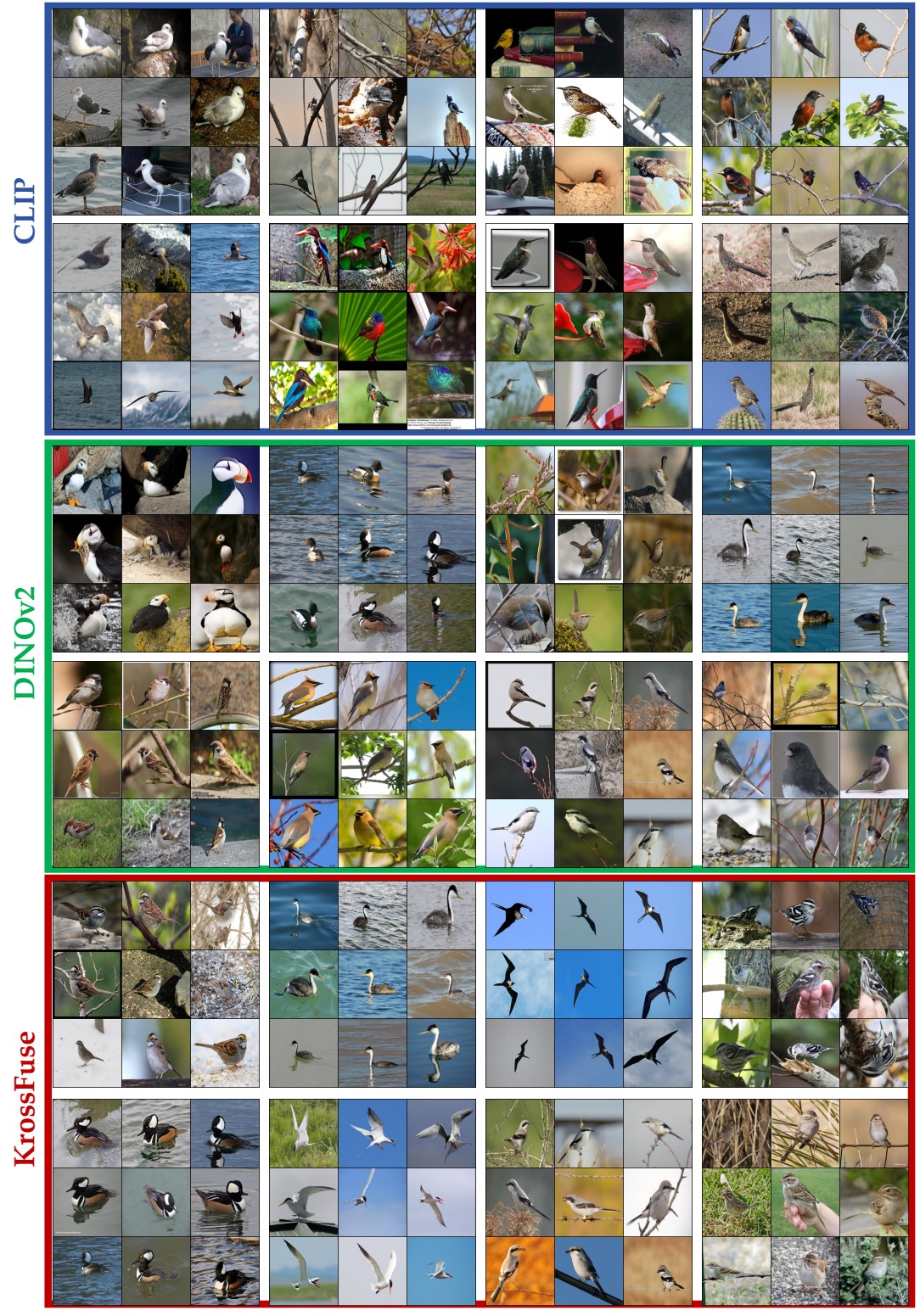

Figure 15: Clustering results of CLIP, DINOv2 and KrossFuse embeddings for CUB200 dataset. KrossFuse could cluster them like DINOv2 while CLIP can't distinguish all of them.

**CUB200 Dataset**

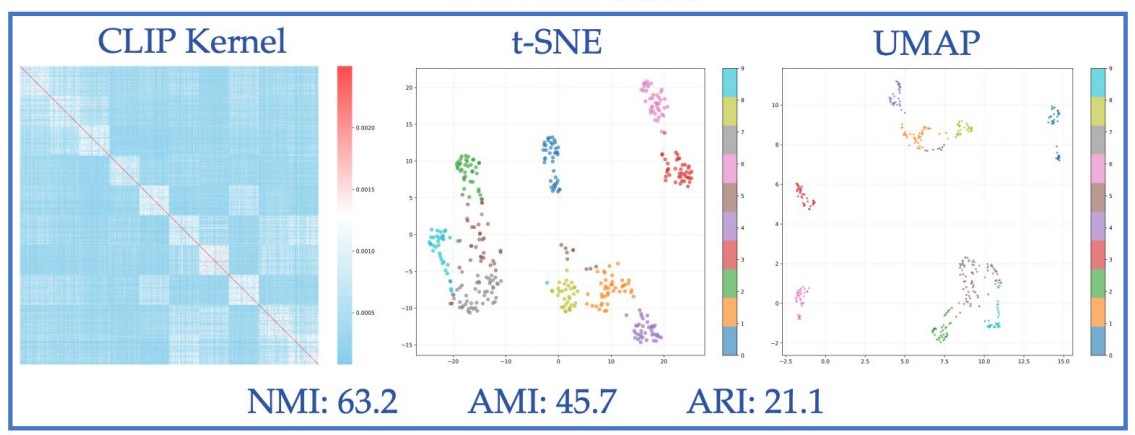

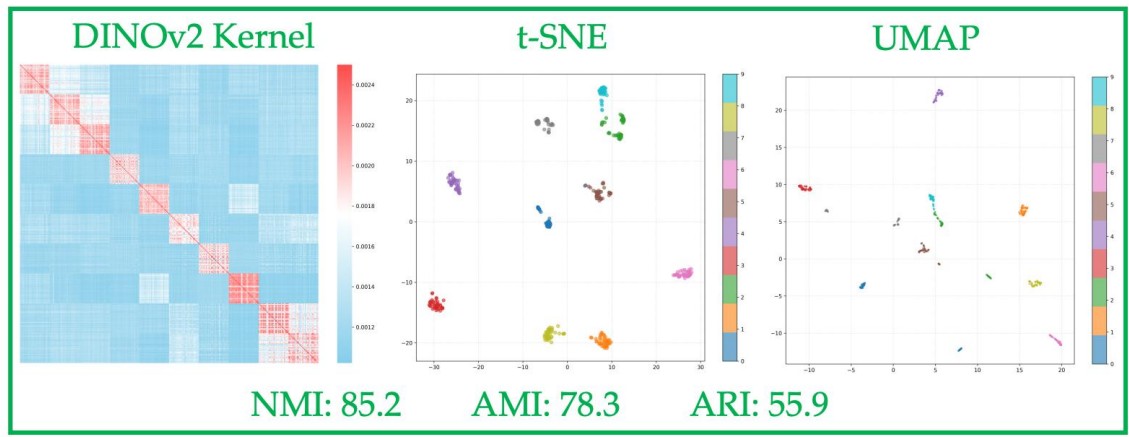

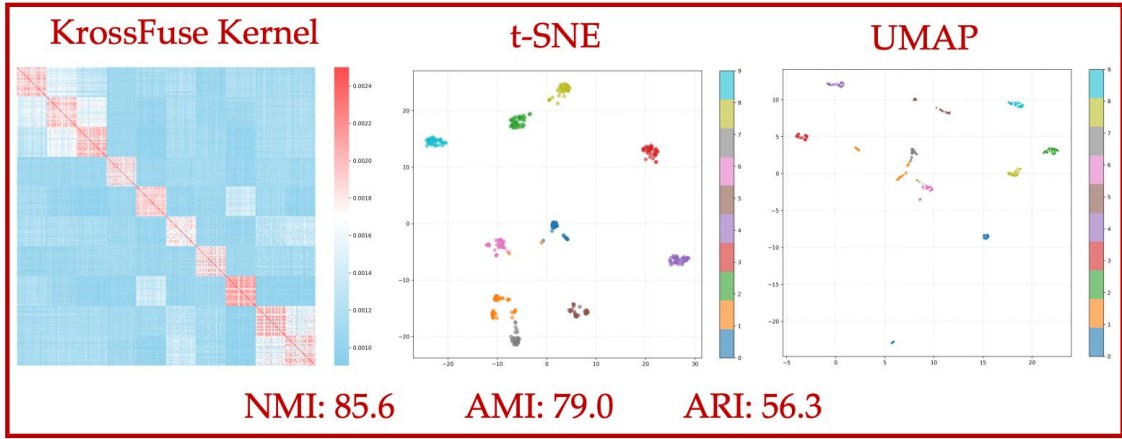

Figure 16: Comparison among CLIP, DINOv2 and KrossFuse embeddings for CUB200 dataset. (Left) Heatmaps of RBF kernel similarity matrices, (Middle) t-SNE visualization, (Right) UMAP visualization. KrossFuse could cluster the different bird classes like DINOv2 while CLIP can't distinguish all of them.

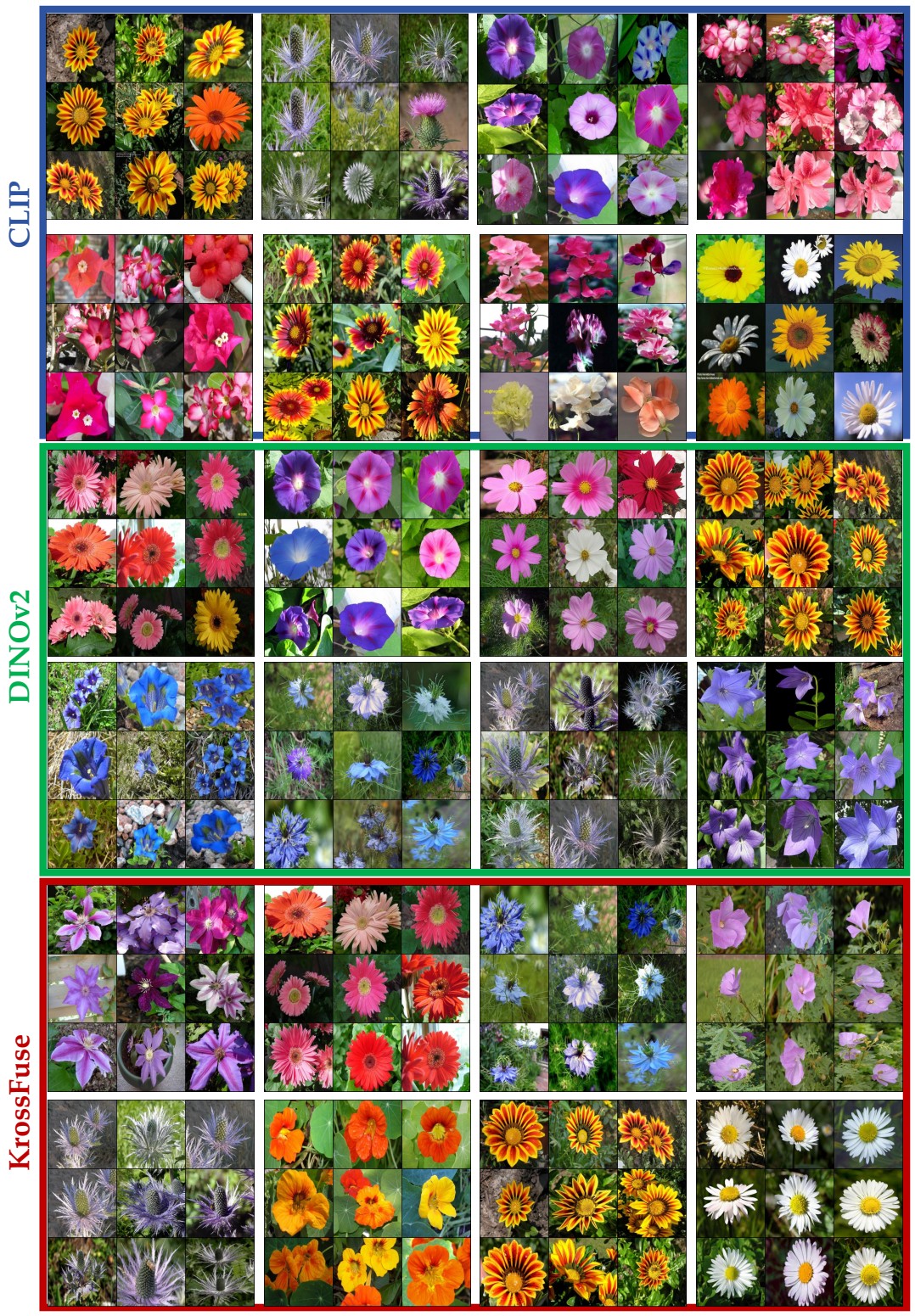

Figure 17: Clustering results of CLIP, DINOv2 and KrossFuse embeddings for Flowers102 dataset. KrossFuse could cluster them like DINOv2 while CLIP can't distinguish all of them.

**Flowers Dataset**

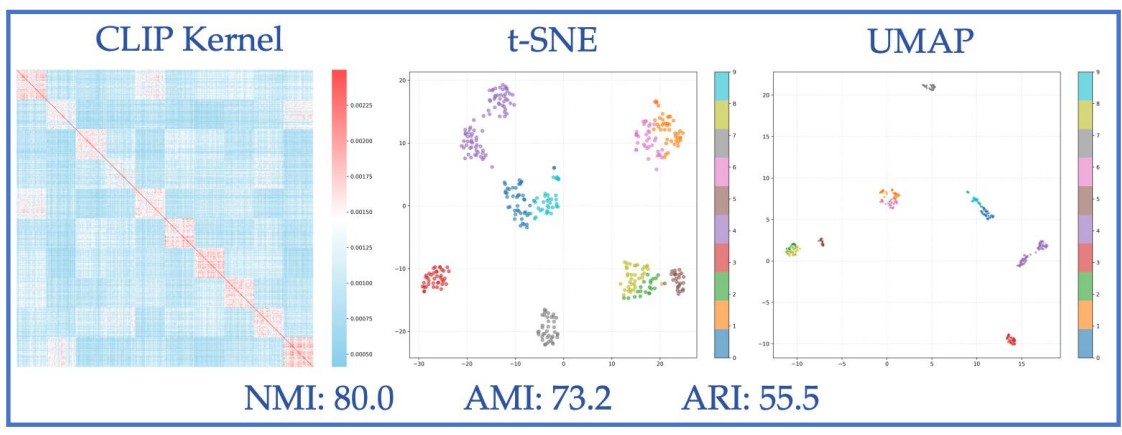

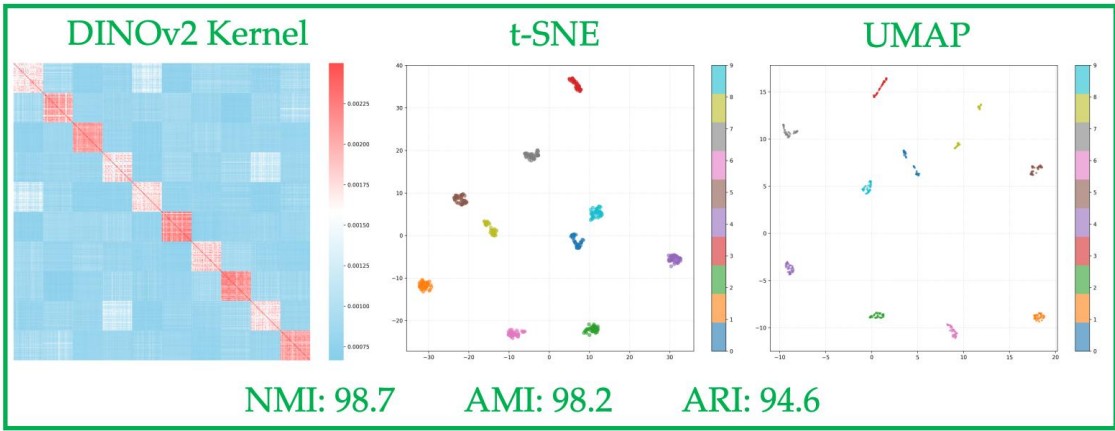

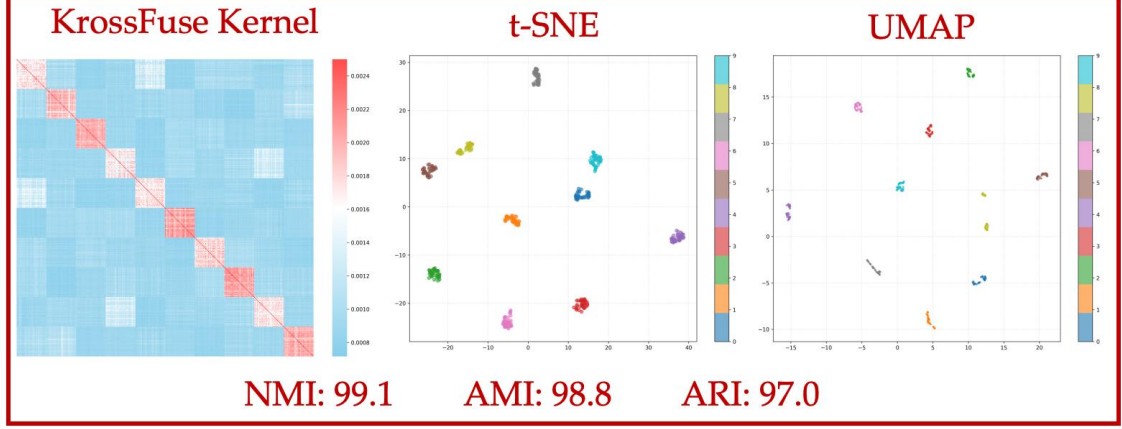

Figure 18: Comparison among CLIP, DINOv2 and KrossFuse embeddings for Flowers102 dataset. (Left) Heatmaps of RBF kernel similarity matrices, (Middle) t-SNE visualization, (Right) UMAP visualization. KrossFuse could cluster the different flower classes like DINOv2 while CLIP can't distinguish all of them.

## C.2 Cross-modal Clustering

While our method achieves comparable performance to CLIP in zero shot image-text retrieval tasks, we observed an interesting phenomenon in cross-modal clustering: despite normalization to the unit hypersphere, image and text embeddings tend to form separate clusters due to the modality gap [41, 83]. This separation persists even when both modalities represent semantically identical concepts, indicating a systematic misalignment in the embedding spaces[3]. To address this challenge, we propose a learned unitary transformation applied to the normalized text embeddings. This rotation operation preserves the geometric structure within each modality while aligning the text embedding space with the image embedding space. Our approach ensures that the semantic relationships are maintained across modalities, enhancing the model's ability to perform cross-modal tasks effectively. We validated our method on the MSCOCO dataset, where we applied t-SNE visualization to demonstrate the elimination of the modality gap. As shown in Figure 19, after applying our unitary transformation, image and text embeddings of the same semantic concepts are well-aligned in the shared embedding space while maintaining their internal structure.

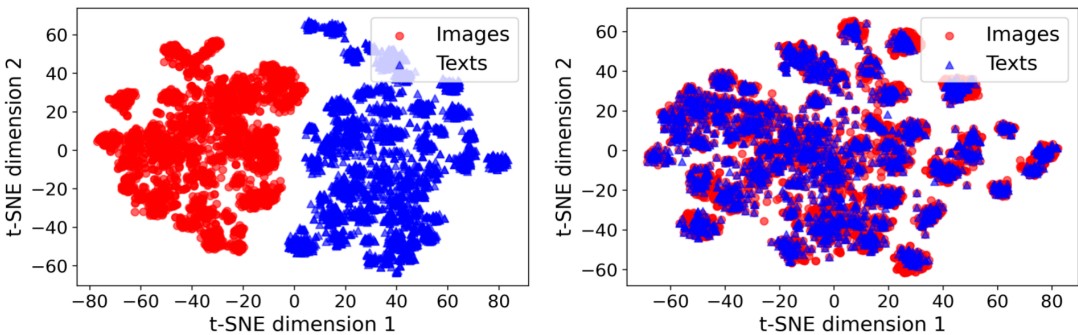

Figure 19: t-SNE visualization of image and text embeddings on MS COCO dataset. **Left:** image embeddings (red circle) and text embeddings (blue triangular) form separate clusters due to modality gap. **Right:** After applying our unitary transformation to text embeddings, the two modalities align well in the embedding space while preserving their internal structure, enabling effective cross-modal clustering.

In the following figures, we showcase the clustering results on the COCO dataset after applying our transformation. This transformation aligns the embedding spaces, enhancing the clustering of semantically similar image-text pairs.

---

[3]We emphasize that this step is limited to the cross-modal clustering setting and was not used in any of the other experiments in the paper.

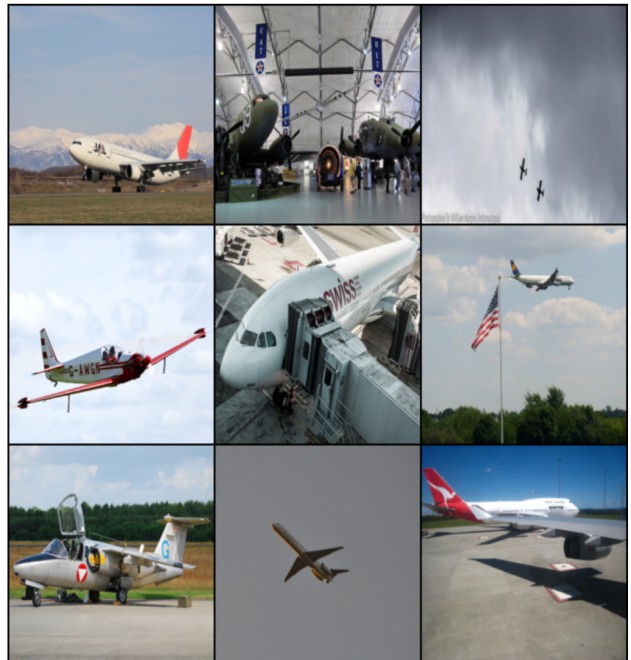

Large commercial cargo plane sits on tarmac next to radar equipment.

A side view of a plane flying in the clear blue sky.

A white airplane parked on a runway under a blue sky.

An airplane leaving a trail in the sky.

Two planes in runway and another plane in the sky.

Some people in high visibility jackets putting suitcases onto a conveyer belt from a container.

A black and white image of a shipyard with some boats. A boat traveling into a water filled tunnel.

A couple of airplanes flying under a cloudy sky.

Figure 20: Post-transformation alignment of image and text embeddings on the COCO dataset. The cluster samples shown are all related to planes, demonstrating complete alignment of visual and textual data.

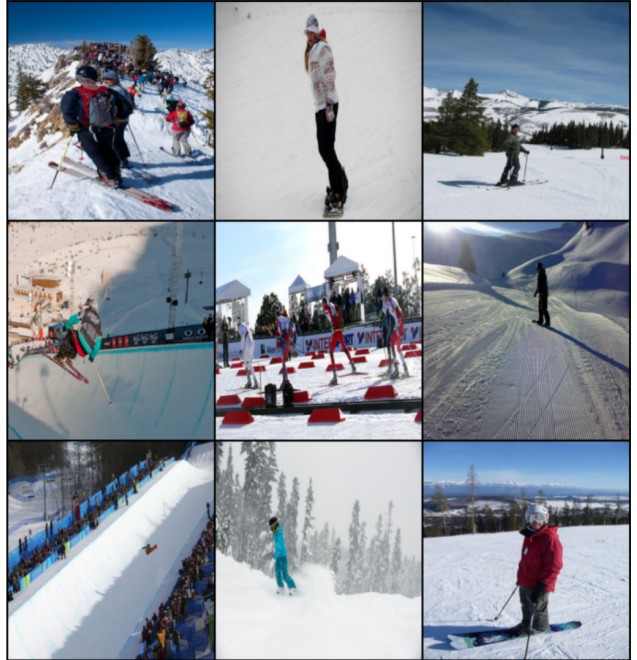

A man on his skis on a snowy slope.

A person skiing through a snowy forest with tall trees.

A man is in his skiing gear, while posing for the camera.

a group of beginner snow skiers having class

there is a male skier going down a hill

a group of people riding skis on a snowy surface

A woman with a snowboard with a man standing next to her on a ski slope.

a person standing snow wearing a snow suit and skis

A woman posing for the camera standing on skis.

Figure 21: Post-transformation alignment of image and text embeddings on the COCO dataset. The cluster samples shown are all related to ski, demonstrating complete alignment of visual and textual data.

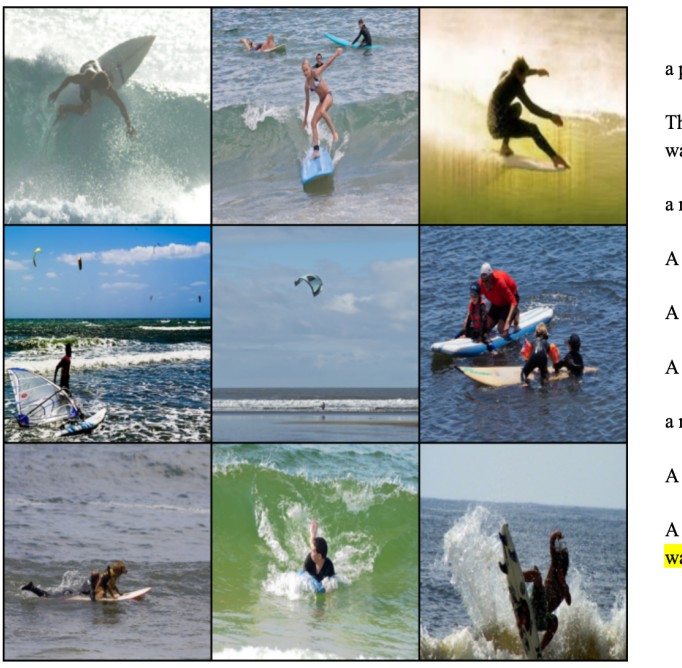

a person parachute surfing on a large body of water

There is a blurry photo of a surfer walking out of the water

a man riding a blue surfboard on top of a wave.

A man is surfing on a wave in the ocean.

A kid on a surboard riding a little wave

A man in a wet suit riding a wave on a surfboard.

a man wearing a wet suit riding the wave

A man and a dog riding a surf board in the water .

A man on a blue raft attempting to catch a ride on a large wave.

Figure 22: Post-transformation alignment of image and text embeddings on the COCO dataset. The cluster samples shown are all related to surf, demonstrating complete alignment of visual and textual data.

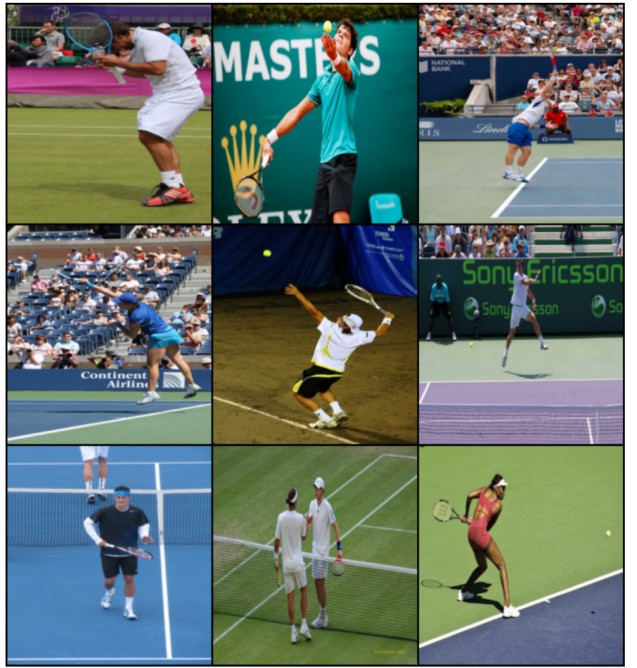

A man standing on a clay tennis court with a racquet.

A beautiful young woman swinging a tennis racquet on a tennis court.

A woman prepares to serve a tennis ball.

A man holding a tennis racket in the air

A man hitting a tennis ball with a tennis racquet.

Venus Williams is swinging at the tennis ball at the game.

A young boy holding a tennis racquet on a court.

A man holding a tennis racquet on top of a tennis court.

A woman holding a tennis racquet while wearing glasses.

Figure 23: Post-transformation alignment of image and text embeddings on the COCO dataset. The cluster samples shown are all related to tennis, demonstrating complete alignment of visual and textual data.

## C.3 Text Linear Probe Results

To validate the effectiveness of RP-KrossFuse for improving the uni-modal performance, we also conducted linear probe experiments in text domain. We used the SentEval toolkit [12] to evaluate the RP-KrossFuse embeddings compared to CLIP, Sroberta and other baselines on the following NLP classification benchmarks: MR [58], CR [24], SUBJ [57], MPQA [76], SST2 [68], TREC [72], and MRPC [14]. The linear probe results are provided in the Table 5. The results show that RP-KrossFuse significantly enhances CLIP's text understanding capabilities, achieving an average improvement of 6% over CLIP itself. In comparison to the baseline, our method demonstrates superior performance across all evaluated tasks.

Table 5: Linear probe evaluation of frozen features of variants of CLIP, Sroberta, RP-KrossFuse and three baselines using the SentEval toolkit on text benchmarks. Test accuracy (%) are based on a 5-fold cross-validation.

| Embedding | Arch | Fused | MR | CR | SUBJ | MPQA | SST2 | TREC | MRPC | Avg |
|---|---|---|---|---|---|---|---|---|---|---|
| CLIP [60] | ViT-B/32 | ✗ | 75.8 | 83.1 | 92.5 | 86.4 | 82.0 | 83.0 | 70.1 | 81.8 |
| | ViT-L/14 | ✗ | 78.1 | 85.3 | 93.8 | 87.0 | 83.9 | 86.4 | 67.7 | 83.2 |
| KPoMRP | ViT-B/32 | ✔ | 73.8 | 78.5 | 86.6 | 82.1 | 80.3 | 74.8 | 70.8 | 78.1 |
| | ViT-L/14 | ✔ | 72.5 | 80.3 | 86.0 | 83.1 | 74.0 | 78.2 | 68.1 | 77.5 |
| GATE [66] | ViT-B/32 | ✔ | 84.8 | 87.2 | 94.4 | 88.5 | 91.8 | 89.3 | 65.5 | 85.9 |
| | ViT-L/14 | ✔ | 84.3 | 87.8 | 94.5 | 88.3 | 91.4 | 89.3 | 67.6 | 86.2 |
| ATTN [84] | ViT-B/32 | ✔ | 85.7 | 86.3 | 93.4 | 88.8 | 91.9 | 88.5 | 66.2 | 85.8 |
| | ViT-L/14 | ✔ | 85.7 | 85.9 | 94.3 | 87.8 | 92.4 | 86.5 | 65.9 | 85.5 |
| RP-KrossFuse | ViT-B/32 | ✔ | 85.8 | 88.7 | 94.4 | 89.1 | 89.7 | 95.0 | 73.6 | **88.0** |
| | ViT-L/14 | ✔ | 86.0 | 88.1 | 94.8 | 89.3 | 89.8 | 95.2 | 73.6 | **88.1** |
| SRoBERTa [62] | TF-L24 | ✗ | 85.1 | 86.8 | 93.7 | 87.7 | 89.1 | 93.2 | 68.1 | 86.2 |

## C.4 Zero Shot Image-to-text and Text-to-image Retrievals

To evaluate the cross modal alignment ability of ours RP-KrossFuse method, we conduct zero-shot image-to-text and text-to-image retrieval experiments on MSCOCO [42] and Flickr30k [77]. In our experiments, we utilize several variants of CLIP, including the base, large, and large with 336 pixel models, as well as the large and huge versions of OpenCLIP, to ensure a comprehensive comparison. The results in table 6 shows that the differences between CLIP and RP-KrossFuse are mostly below 1%, suggesting that RP-KrossFuse maintains strong zero shot cross-modal alignment, with retrieval performance comparable to CLIP.

## C.5 Improvement of Image Representation vs Alignment of Zero Shot Classification

We performed linear probe and zero-shot classification tasks on multiple datasets covering different visual recognition settings: ImageNet [13], CIFAR-10, CIFAR-100 [35], Caltech101 [16], Food101 [9], Oxford Flowers [51], Oxford-IIIT Pet [59], and DTD [11]. The detailed results are shown in Table 7. As shown in Table 7(a), RP-KrossFuse improves the averaged linear probe accuracy over several image benchmarks from 83.3% to 91.2%. A more detailed breakdown of per-dataset image improvement is visualized in Figure 24(left) where we observe more than 10% increases on 4 out of 8 datasets. Notably, on the ImageNet dataset, RP-KrossFuse irpved over CLIP from 73.2% to 84.1%, suggesting a considerable improvement in the image representation alignment with the actual label. On the other hand, the averaged zero-shot accuracy of RP-KrossFuse drops by 0.5% compared to CLIP(see Table 7(a)). This is indeed expected because we involved uni-modal expert's embedding which is not aligned with the other modality whereas CLIP is trained to align image and text explicitly. Also, we note that most lower scores are underperforming by at most 1% (on 7 out of 8 datasets), which are relatively outweighed by the gains in uni-modal representations shown in Figure 24 (left). For example, for ImageNet where RP-KrossFuse suffers the gap of -1.57%, the boost in uni-modal representation of 10. 90% is the second highest accuracy over the datasets.

Table 6: Zero-shot image-text retrieval performance on Flickr30K and MSCOCO. We report Recall@K (%) for Image→Text and Text→Image retrieval. Fuse denotes our RP-KrossFuse fusion method. Superscripts denote the backbone variant: B = ViT-B/32, L = ViT-L/14, L+ = ViT-L/14@336px, H = ViT-H/14.

| | Flickr30K | | | | | | MSCOCO | | | | | |
| | Image→Text | | | Text→Image | | | Image→Text | | | Text→Image | | |
| Method | R@1 | R@5 | R@10 | R@1 | R@5 | R@10 | R@1 | R@5 | R@10 | R@1 | R@5 | R@10 |
|---|---|---|---|---|---|---|---|---|---|---|---|---|
| CLIP$^B$ | 76.9 | 94.3 | 97.8 | 57.9 | 82.9 | 89.0 | 48.4 | 73.8 | 81.6 | 29.8 | 54.0 | 65.0 |
| Fuse$^B$ | 75.7 | 94.6 | 97.7 | 58.0 | 82.5 | 89.0 | 48.9 | 72.7 | 81.3 | 29.1 | 53.3 | 64.4 |
| CLIP$^L$ | 85.9 | 97.3 | 99.2 | 64.5 | 87.1 | 91.9 | 56.9 | 79.6 | 86.7 | 35.7 | 60.4 | 70.5 |
| Fuse$^L$ | 85.0 | 97.0 | 99.0 | 64.6 | 86.6 | 91.9 | 56.5 | 79.0 | 86.2 | 35.7 | 60.3 | 70.2 |
| CLIP$^{L+}$ | 87.7 | 98.5 | 99.4 | 66.8 | 88.9 | 93.3 | 57.3 | 80.2 | 87.5 | 35.9 | 60.7 | 70.7 |
| Fuse$^{L+}$ | 87.2 | 98.2 | 99.4 | 67.0 | 88.5 | 93.0 | 57.9 | 80.3 | 87.0 | 35.8 | 60.3 | 70.4 |
| OpenCLIP$^L$ | 89.0 | 98.5 | 99.3 | 74.9 | 92.4 | 95.6 | 61.8 | 83.6 | 89.9 | 45.5 | 70.4 | 79.0 |
| Fuse$^{OpenL}$ | 89.0 | 98.4 | 99.4 | 74.6 | 92.5 | 95.6 | 62.3 | 83.8 | 90.0 | 45.2 | 70.2 | 78.8 |
| OpenCLIP$^H$ | 90.7 | 99.2 | 99.7 | 77.6 | 94.2 | 96.6 | 66.1 | 86.5 | 91.9 | 48.5 | 72.8 | 81.0 |
| Fuse$^{OpenH}$ | 91.0 | 99.3 | 99.8 | 77.5 | 94.0 | 96.7 | 66.3 | 86.5 | 92.0 | 48.4 | 72.7 | 81.1 |

Table 7: Comparison of CLIP and RP-KrossFuse in the linear probe and zero-shot classification setting. LP: linear probe. ZS: zero shot.

**(a) Average.**

| | LP | ZS |
|---|---|---|
| CLIP | 83.3 | 69.4 |
| RP-KrossFuse | 91.2 | 68.9 |
| $\Delta$ | +7.9 | -0.5 |

**(b) ImageNet.**

| | LP | ZS |
|---|---|---|
| CLIP | 73.2 | 57.9 |
| RP-KrossFuse | 84.1 | 56.3 |
| $\Delta$ | +10.9 | -1.6 |

**(c) CIFAR-10.**

| | LP | ZS |
|---|---|---|
| CLIP | 95.0 | 88.8 |
| RP-KrossFuse | 98.6 | 88.4 |
| $\Delta$ | +3.6 | -0.4 |

**(d) CIFAR-100.**

| | LP | ZS |
|---|---|---|
| CLIP | 80.0 | 61.7 |
| RP-KrossFuse | 90.5 | 61.0 |
| $\Delta$ | +10.5 | -0.7 |

**(e) Caltech101.**

| | LP | ZS |
|---|---|---|
| CLIP | 92.6 | 85.3 |
| RP-KrossFuse | 95.6 | 85.2 |
| $\Delta$ | +3.0 | -0.1 |

**(f) Food101.**

| | LP | ZS |
|---|---|---|
| CLIP | 87.3 | 79.2 |
| RP-KrossFuse | 90.9 | 78.5 |
| $\Delta$ | +3.6 | -0.7 |

**(g) Oxford Flowers.**

| | LP | ZS |
|---|---|---|
| CLIP | 84.9 | 63.5 |
| RP-KrossFuse | 99.7 | 62.8 |
| $\Delta$ | +14.8 | -0.7 |

**(h) Oxford-IIIT Pet.**

| | LP | ZS |
|---|---|---|
| CLIP | 87.9 | 75.5 |
| RP-KrossFuse | 95.3 | 75.7 |
| $\Delta$ | +7.4 | +0.2 |

**(i) DTD.**

| | LP | ZS |
|---|---|---|
| CLIP | 65.3 | 43.0 |
| RP-KrossFuse | 75.2 | 43.3 |
| $\Delta$ | +9.9 | +0.3 |

## C.6 Analysis of Ablation Studies

To test the effect of the different RP-KrossFuse components, we evaluated the classification accuracy on ImageNet and the average accuracy across seven NLP benchmarks in SentEval.

**Effect of Image Expert Embeddings.** To assess the impact of incorporating additional image expert embeddings on the performance of the image modality, we conducted a linear probe on ImageNet by fusing the CLIP embedding with UniCom [3] as an alternative image expert. As illustrated in Figure 25(a), integrating UniCom with CLIP increases the accuracy from 73.2% to 79.17%, demonstrating nearly 6% improvement over using CLIP alone. Furthermore, fusing CLIP with DINOv2 yields even higher accuracy than with UniCom, indicating that the effectiveness of RP-KrossFuse is closely related to the quality of the expert embeddings employed. Specifically,

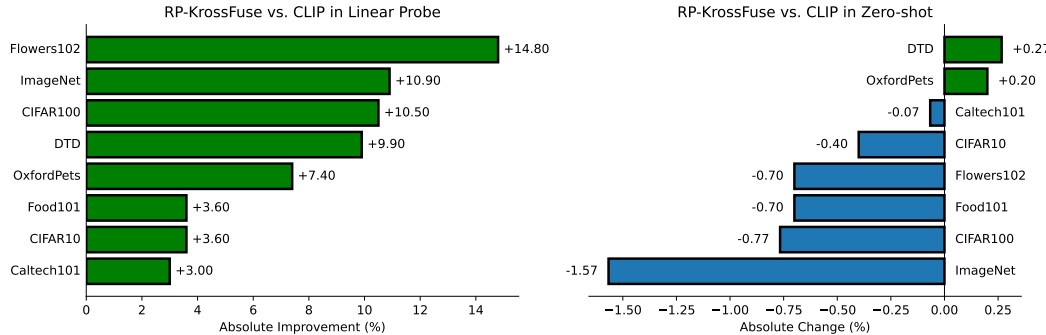

Figure 24: Comprehensive comparisons of RP-KrossFuse and CLIP in the linear probe and zero shot classification settings. (Left) RP-KrossFUse is able to gain consistent improvements over CLIP in linear probe on all datasets. (Right) RP-KrossFuse's declines in zero shot accuracy are mostly under 1%, which are far outweighed by the gains in linear probe.

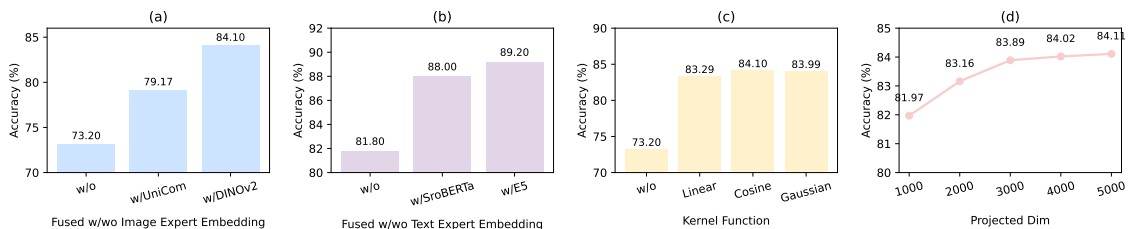

Figure 25: Ablation Studies. (a) (b) Effect of fusing different image and text expert embeddings. (c) Effect of kernel function. (d) Effect of random projected dimension.

since DINOv2 provides stronger representations than UniCom, the overall performance gain is more pronounced when DINOv2 is used as the expert embedding.

**Effect of Text Expert Embeddings.** To evaluate the generalizability of our approach with different text experts, we replace RoBERTa with E5 in the RP-KrossFuse framework and conduct linear probe experiments using the SentEval toolkit on seven NLP benchmarks: MR [58], CR [24], SUBJ [57], MPQA [76], SST2 [68], TREC [72], and MRPC [14]. As shown in Table 8, incorporating either RoBERTa or E5 as an additional text expert consistently improves the performance over the original CLIP text encoder, with E5 yielding an average accuracy gain of 7.4%. These results demonstrate that RP-KrossFuse can effectively leverage various strong text experts to enhance text representation.

Table 8: Linear probe evaluation of CLIP, E5, SRoBERTa and RP-KrossFuse using the SentEval toolkit on text benchmarks. Test accuracy (%) are based on a 5-fold cross-validation. CLIP+E5 and CLIP+SRoBERTa are RP-KrossFuse methods with different text embeddings.

| Embedding | MR | CR | SUBJ | MPQA | SST2 | TREC | MRPC | Avg |
|---|---|---|---|---|---|---|---|---|
| CLIP | 75.8 | 83.1 | 92.5 | 86.4 | 82.0 | 83.0 | 70.1 | 81.8 |
| E5 | 87.1 | 91.1 | 94.7 | 90.2 | 92.0 | 95.7 | 73.1 | 89.1 |
| CLIP+E5 | 86.8 | 91.3 | 94.3 | 90.5 | 91.9 | 95.6 | 74.1 | 89.2 |
| SRoBERTa | 85.1 | 86.8 | 93.7 | 87.7 | 89.1 | 93.2 | 68.1 | 86.2 |
| CLIP+SRoBERTa | 85.8 | 88.7 | 94.4 | 89.1 | 89.7 | 95.0 | 73.6 | 88.0 |

**Effect of Cross-modal Embeddings.** To assess the generalization capability of RP-KrossFuse in enhancing unimodal performance while maintaining cross-modal alignment, we further conducted linear probe and zero-shot experiments on the ImageNet dataset using several multimodal embedding

baselines. As shown in Table 9, RP-KrossFuse consistently improves linear probe accuracy across different backbones, indicating stronger unimodal representation, while maintaining comparable zero-shot performance, demonstrating preserved cross-modal alignment. These results highlight the generality and effectiveness of RP-KrossFuse when applied to various cross-modal embedding architectures.

Table 9: Comparison of linear probe and zero-shot accuracy (%) between baseline cross-modal embeddings and their RP-KrossFuse counterparts on ImageNet.

| Embedding Model | Linear Probe | | Zero-Shot | |
|---|---|---|---|---|
| | Baseline | RP-KrossFuse | Baseline | RP-KrossFuse |
| SigLIP [80] | 81.7 | **83.7** | 73.1 | 72.8 |
| MobileCLIP [70] | 82.3 | **84.5** | 71.1 | 70.9 |
| OpenVision [40] | 79.3 | **84.2** | 65.9 | 65.1 |

**Effect of Kernel Function.** To validate the role of kernel function in RP-KrossFuse framework, we conducted linear probe experiments among different kernel functions. As shown in Figure 25(c), those kernel-based fusions can led to almost 10% classification accuracy improvements on ImageNet dataset, with the Cosine and RBF kernels yielding slightly higher gains than the linear kernel. In the main text, we use cosine kernel for classification and RBF kernel for clustering.

**Effect of Random Projected Dimension.** We investigate how the choice of random projection dimension affects the performance of RP-KrossFuse. As shown in Figure 25(d), increasing the projection dimension leads to a steady improvement in classification accuracy, which gradually saturates as the dimension becomes larger. Notably, when the dimension reaches around 3000, the performance of RP-KrossFuse closely approaches that of the KrossFuse, indicating that a sufficiently high projection dimension can effectively preserve the information required for optimal fusion.

**Effect of Hyperparameter $C$.** The constant $C$ in Equation (3) balances the influence of uni-modal embeddings on the fused shared-modality kernel similarity and maintains balance between similarity scores for shared and non-shared modalities. Specifically, when considering images and text as shared and non-shared modalities, a smaller $C$ increases the weight of the image uni-modal embedding, but reduces the balance between (image, image), (image, text), and (text, text) similarity scores. To empirically validate this effect, we evaluated the cosine similarity distributions of image-text pairs on the MSCOCO validation set under different values of $C$. As shown in Figure 26, smaller values of $C$ lead to a slightly larger separation between the similarity curves of CLIP and RP-KrossFuse, whereas larger $C$ results in more overlapping curves, indicating improved cross-modal alignment.

## C.7 Computational Efficiency of RP-KrossFuse

We provide additional analysis of the computational efficiency of our proposed RP-KrossFuse scheme. Theoretically, RP-KrossFuse reduces the computational complexity from $\mathcal{O}(d_1 d_2 l)$ (naive random projection) to $\mathcal{O}((d_1 + d_2)l)$ by leveraging structured random projections, which enables more scalable multimodal fusion. Regarding inference efficiency, it is important to note that, assuming sufficient memory, our method allows for the two embedding models to operate in parallel, thereby avoiding increased inference latency. To support this, we conducted benchmarking experiments on two NVIDIA RTX 3090 GPUs with a batch size of 128. The original KrossFuse pipeline required 491.3 ms per batch and 18.406 GB of peak memory, while RP-KrossFuse (with $l = 5000$) achieved comparable performance with just 413.1 ms per batch and a significantly reduced memory footprint of 3.39 GB. Notably, the fusion step's overhead was reduced from 22.0 ms to only 0.14 ms. For reference, the baseline CLIP and DINOv2 models individually required 73.6 ms and 392.1 ms per batch, respectively.

## C.8 Concatenation-Only Baseline

We additionally analyze a concatenation-only baseline to better understand the advantages of the proposed RP-KrossFuse fusion strategy. From a theoretical perspective, concatenating two embeddings corresponds to defining a kernel similarity function that is the *sum* of the individual kernels. This formulation implies that two samples will have high similarity in the fused space only if both original

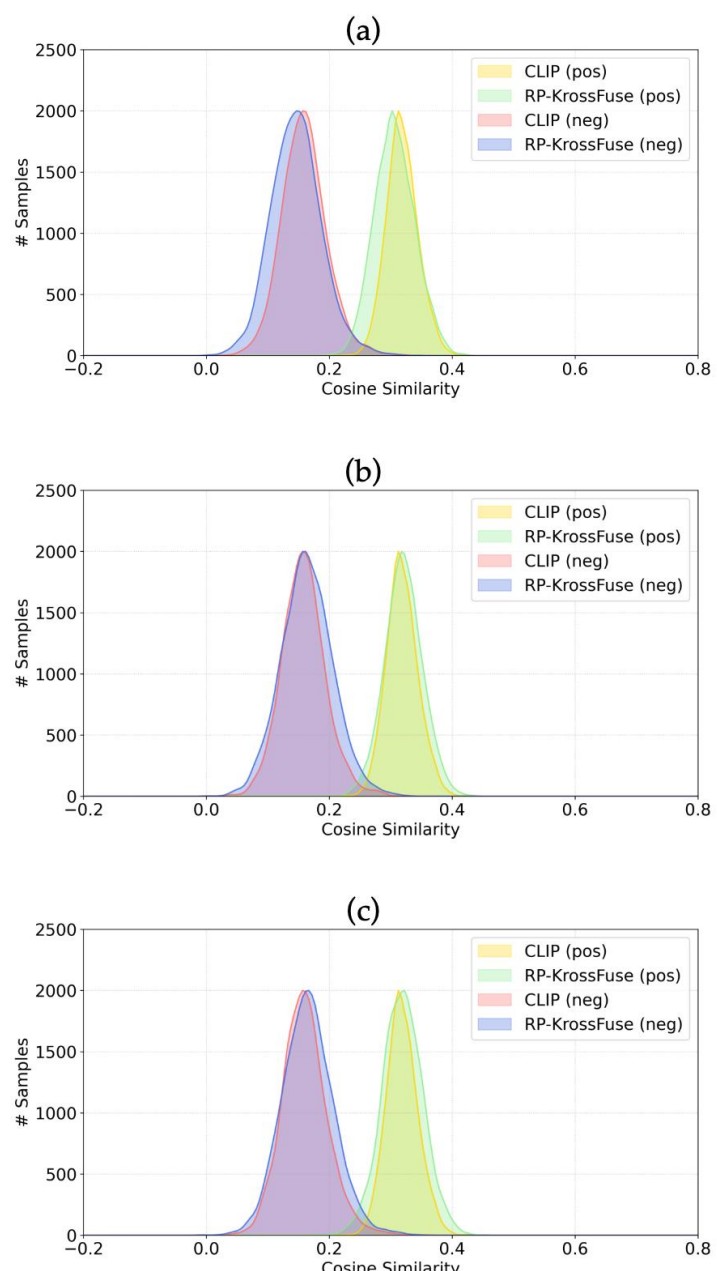

Figure 26: The cosine similarity distributions of positive/negative image text pairs on MSCOCO dataset across various values of hyperparameter $C$. (a) $C = 0.1$. (b) $C = 0.5$. (c) $C = 1$.

Table 10: Benchmarking of inference and fusion efficiency.

| Method | Time/Batch (ms) | Peak Memory (GB) |
|---|---|---|
| CLIP | 73.6 | 2.077 |
| DINOv2 | 392.1 | 3.199 |
| KrossFuse Pipeline | 491.3 | 18.406 |
| KrossFuse (Overhead) | 22.0 | 17.203 |
| Parallel RP-KrossFuse Pipeline | 413.1 | 3.387 |
| RP-KrossFuse (Overhead) | 0.14 | 1.729 |

embeddings assign them high similarity, effectively modeling the *intersection* of the semantic concepts captured by each modality. In contrast, the Kronecker-based fusion employed in RP-KrossFuse leads to a *product* of kernel similarities, which enables distinguishing two samples as long as either embedding distinguishes them, thereby modeling the *union* of their semantic representations.

In supervised classification settings, the concatenation baseline is expected to perform comparably to the stronger of the individual embeddings. This is because the concatenated feature vector preserves the full representational capacity of both embeddings, while the limited VC dimension of the linear probe mitigates overfitting. However, in unsupervised or weakly supervised scenarios such as clustering, the product-based Kronecker fusion can better capture complementary cross-modal structures, leading to improved representation quality and broader concept coverage.

Empirically, we compared the concatenation baseline with RP-KrossFuse across multiple tasks. For supervised ImageNet linear probing, both methods achieve similar performance (concatenation: 83.6%, RP-KrossFuse: 84.1%), which aligns with theoretical expectations. In multimodal classification tasks, RP-KrossFuse consistently outperforms the concatenation baseline on the MVSA dataset [52], achieving 74.3% vs. 71.3% on MVSA-Single and 66.6% vs. 63.7% on MVSA-Multiple. The largest gains are observed in unsupervised clustering tasks, where RP-KrossFuse shows clear improvements across datasets, as summarized in Table 11.

Table 11: Comparison between the concatenation-only baseline and RP-KrossFuse on unsupervised clustering tasks, evaluated using NMI, AMI, and ARI metrics (%).

| Method | Metric | Flowers102 | DTD | ImageNet-Dogs | GTSRB | Typo-Attacked IN |
|---|---|---|---|---|---|---|
| Concatenation | NMI | 98.4 | 60.5 | 76.3 | 43.8 | 44.4 |
| RP-KrossFuse | NMI | **99.1** | **62.9** | **88.3** | **50.0** | **87.4** |
| Concatenation | AMI | 97.9 | 57.9 | 76.1 | 40.2 | 43.9 |
| RP-KrossFuse | AMI | **98.8** | **60.4** | **88.2** | **46.7** | **87.3** |
| Concatenation | ARI | 93.9 | 35.3 | 64.9 | 14.1 | 28.4 |
| RP-KrossFuse | ARI | **97.0** | **36.4** | **86.3** | **19.5** | **79.6** |

## C.9 Application of RP-KrossFuse in Text-to-Image Diffusion Models

To further demonstrate the practical utility of the proposed RP-KrossFuse embedding fusion, we apply it to conditional image generation with text-to-image diffusion models. Specifically, we adapt the Vendi Score Guidance (VSG) framework proposed by Askari *et al.* [4], which guides the reverse diffusion process using the Vendi diversity score of the generated samples. In the original implementation, CLIP embeddings were used to compute the Vendi diversity score. We replace this embedding with the Kronecker-fused representation of DINOv2 and CLIP, thereby enriching the diversity guidance with complementary visual and multimodal semantics.

Experiments were conducted on the ImageNet dataset using a class-conditional Diffusion Transformer (DiT-XL/2) as the backbone. Two guidance configurations were compared: (1) the original CLIP-based contextualized Vendi Score Guidance (c-VSG), and (2) our KrossFuse-based version, which employs the fused CLIP and DINOv2 embeddings. As reported in Table 12, incorporating RP-KrossFuse improves both diversity and fidelity of the generated samples. Specifically, the diversity metrics—Recall [36] and Coverage [49]—increase notably, while the quality metrics—Precision [36] and Density [49]—also improve over the CLIP-only baseline.

Table 12: Comparison of Vendi Score diversity guidance methods for text-to-image diffusion on ImageNet using DiT-XL/2. Metrics follow [36, 49].

| Diversity Guidance Method | Precision | Recall | Density | Coverage |
|---|---|---|---|---|
| c-VSG Guidance (CLIP) | 0.913 | 0.413 | 1.206 | 0.552 |
| KrossFuse (CLIP + DINOv2) | **0.932** | **0.484** | **1.252** | **0.613** |

