# OpenReview forum: "When Kernels Multiply, Clusters Unify: Fusing Embeddings with the Kronecker Product"
_NeurIPS.cc/2025/Conference — NeurIPS 2025 poster_

### Official Review · Reviewer_m3ub · 2025-06-30

**Clarity:** 2
**Significance:** 2
**Originality:** 4
**Rating:** 3
**Confidence:** 4

**Summary:**

This paper focuses on the performance trade-off between cross-modal embeddings and single-modality embeddings in the field of multi-modal learning. Cross-modal embeddings excel at aligning representations across modalities, but may underperform compared to specialized single-modality embeddings on modality-specific tasks. To address this, the authors propose the KrossFuse method, whose core idea is to leverage the product of kernel functions and fuse the feature maps of different modal embeddings through the Kronecker product. This method allows the fused embedding to achieve fine-grained distinction between samples if either of the original embeddings can distinguish them. The paper further introduces the scalable RP-KrossFuse method. RP-KrossFuse cleverly applies random projections to reduce the dimensionality of the fused representation while approximately preserving the kernel similarity values of KrossFuse. RP-KrossFuse maintains CLIP's zero-shot cross-modal alignment capability and also achieves superior performance over baselines like CLIP and DINOv2 in cross-modal few-shot learning tasks.

**Questions:**

1.	While this study provides theoretical grounds concerning kernel function products and Kronecker products, a more thorough theoretical analysis is still needed regarding the deeper mechanisms of Kronecker product in semantic information fusion and the practical impact of approximation errors from random projections on fine-grained feature distinction capabilities, with the aim of transcending the current preliminary explanations based on kernel function products. Please provide a more intuitive and in-depth explanation from an information-theoretic or geometric perspective. Furthermore, the practical impact of approximation errors introduced by random projection on fine-grained feature distinction capabilities is not sufficiently discussed in the paper. Please analyze in detail under what conditions this approximation error might become significant and potentially lead to performance degradation.
2.	The paper emphasizes RP-KrossFuse's "training-free" characteristic and computational efficiency. However, there is a lack of direct quantitative comparison with existing training-based fusion methods regarding inference time and resource consumption. Please provide specific resource consumption for RP-KrossFuse at typical dataset scales, and quantify this against single-modality embedding methods such as CLIP or DINOv2, to fully demonstrate its efficiency advantages.
3.	In Figures 1 and 3, the paper utilizes kernel matrix heatmaps to demonstrate KrossFuse's clustering capability. Although the text description claims that the Kronecker product can better separate classes, the visual effect of the figures themselves lacks sufficient persuasiveness in terms of discriminability, making it difficult for readers to intuitively perceive the significant advantage of RP-KrossFuse. Please improve the visual effectiveness of the charts.
If the authors can provide a more in-depth and intuitive theoretical explanation, offer a more detailed analysis and discussion of the error effects of random projection, and improve the visual effectiveness of the figures, this will significantly enhance the paper's theoretical depth and comprehensibility, and I may increase the clarity rating by one point.
4.	If the CLIP visual encoder, widely used in LLaVA and other vision-language models, were to be replaced with RP-KrossFuse's fused image embeddings, would this significantly enhance the model's visual understanding capabilities? Since RP-KrossFuse aims to bridge the performance gap between cross-modal embeddings like CLIP and single-modality expert embeddings like DINOv2, theoretically, its fused image representations should possess stronger fine-grained visual recognition capabilities while maintaining cross-modal alignment. Have the authors considered or preliminarily explored such a replacement? For instance, in downstream tasks for VLMs, such as visual question answering, image captioning, or instruction following, could using RP-KrossFuse image embeddings lead to performance improvements, especially in scenarios requiring detailed visual discrimination?
If the authors can provide preliminary experimental results or compelling theoretical analysis demonstrating that RP-KrossFuse fused image embeddings can indeed enhance the visual capabilities of VLMs (such as LLaVA), especially in tasks requiring stronger visual detail understanding, I will increase the significance and quality ratings.

**Ethical Concerns:**

["NO or VERY MINOR ethics concerns only"]

**Final Justification:**

After careful consideration of their responses and alignment with fellow reviewers, I will maintain my original rating.

**Limitations:**

yes

**Quality:**

3

**Strengths And Weaknesses:**

Strengths：
1. The paper proposes a novel, training-free embedding fusion method called KrossFuse, which is based on the Kronecker product. This contrasts with most existing fusion strategies that rely on additional training, offering a new and more efficient paradigm for multimodal representation learning.
2. The paper proposes that when the fused kernel function is the product of two original kernel functions, its feature map is the Kronecker product of these two original feature maps. To address the problem of missing non-shared modalities in single-modality embeddings, the paper solves this by constructing a symmetrized cross-modal embedding, enabling the application of Kronecker product fusion while maintaining unchanged similarity scores between shared and non-shared modalities.
3. In RP-KrossFuse, the integration of random projection with the Kronecker product, achieved through the Hadamard product, aims to efficiently approximate high-dimensional kernel spaces; this represents a novel and practical technical combination.
4. The paper conducted detailed ablation studies to analyze the impact of different expert embeddings, kernel function types, and random projection dimensions on performance. This helps in understanding the role of each component in RP-KrossFuse

Weaknesses:
1. In Figures 1 and 3, the paper utilizes kernel matrix heatmaps to demonstrate KrossFuse's clustering capability. Although the text description claims that the Kronecker product can better separate classes and capture the clustering structures of different embeddings, the visual effect of the figures themselves does not always clearly support this claim. It is difficult for readers to intuitively perceive the significant advantage of KrossFuse compared to single embeddings, and its clarity and distinctiveness are not highly convincing. This visual ambiguity may weaken the persuasiveness of the experimental results and affect the paper's clarity.
2. While this study provides theoretical grounds concerning kernel function products and Kronecker products, a more thorough theoretical analysis is still needed regarding the deeper mechanisms of Kronecker product in semantic information fusion and the practical impact of approximation errors from random projections on fine-grained feature distinction capabilities, with the aim of transcending the current preliminary explanations based on kernel function products.
3. The paper emphasizes RP-KrossFuse's "training-free" characteristic and computational efficiency. However, it lacks a direct quantitative comparison with existing training-based fusion methods regarding actual training/inference time and resource consumption. Although computational cost is mentioned, a specific comparison with CLIP or single-modality embeddings is not provided, which makes the intuitive understanding of its efficiency advantages insufficient.

---

> ### Author Rebuttal · Authors · 2025-07-31
>
> We would like to thank Reviewer m3ub for the constructive and thoughtful feedback on our work. We are glad that the reviewer finds our work to propose a “novel and practical technical combination”. Below is our response and clarifications regarding the questions and comments in the review:
>
> **1- Quantitative comparisons on clustering results for Figures 1 and 3**
>
> We thank the reviewer for pointing this out. Originally, we decided to limit the evaluation of Figure 1 in the introduction to the qualitative comparison of kernel heatmaps. To address the reviewer’s concern, we have measured the clustering score, Adjusted Rand Index, Adjusted and Normalized Mutual Information (ARI, AMI and NMI), and the results are in the following table, indicating the better clustering of the Kronecker product fusion in comparison to each of the CLIP and DINOv2 embeddings.
>
>
> | Method | NMI | ARI | AMI |
> |--------|:---:|:---:|:---:|
> | CLIP | 75.8 | 68.6 | 75.8 |
> | DINOv2 | 71.6 | 59.4 | 71.6 |
> | RP-KrossFuse | **84.4** | **76.5** | **84.4** |
>
>
> **2- Theoretical analysis of Kronecker product in semantic information fusion**
>
> We appreciate the reviewer’s point on the theoretical analysis of the role of Kronecker product in combining the semantic information of the two embeddings. As explained in the text (Propositions 1 and 2), the Kroncker product fusion leads to the product of the two kernel functions. A simple but somewhat insightful example to understand how the semantic features can be merged is when we have a 2-dimensional feature vector $[x,y]$ where each of the two embeddings can capture the information of only one of the coordinates, i.e. $\psi_1([x,y]) = f(x)$  and  $\psi_2([x,y]) = g(y)$ for mutivaraibel functions $f$ and $g$. Then, the Gaussian kernel fusion of the two embeddings will lead to the following kernel functions
>
> $$k_{fuse}([x,y],[x’,y’]) =  \exp(\frac{||f(x)-f(x’)||^2 + ||g(y)-g(y’)||^2}{\sigma^2})$$
>
> As seen, while neither of the embeddings contained the full information, the kernel function of the Kronecker product captures the role of both the coordinates. We will discuss this simple example in the revised text to better highlight how the fused embedding can capture the semantic information of both the embeddings.
>
> **3- Comparison with training-based fusion methods**
>
> We appreciate the reviewer’s comment regarding the comparison with training-based fusion approaches. To clarify, the baselines GATE and ATTN included in our study are indeed training-based fusion methods. However, we understand that the reviewer’s comment could more generally refer to a training-time integration, retraining one embedding model from scratch while conditioning on the other, potentially enabling lower inference-time complexity.
>
> Regarding inference efficiency, it is important to note that, assuming sufficient memory, our method allows for the two embedding models to operate in parallel, thereby avoiding increased inference latency. To support this, we conducted benchmarking experiments on an NVIDIA RTX 3090 GPU with a batch size of 128. The individual CLIP and DINOv2 models required 73.6 ms and 392.1 ms per batch, respectively. Our parallel RP-KrossFuse pipeline achieved a total inference time of 413.1 ms, which is only slightly higher than the slower of the two models. Notably, the fusion step itself adds just 0.14 ms of overhead.
> |                 Method                | Time/Batch (ms) | Peak Memory (GB) |
> |---------------------------------------|-----------------|-----------------|
> |                  CLIP                 |      73.6      |      2.077       |
> |                 DINOv2                |      392.1     |      3.199       |
> | Parallel RP-KrossFuse Pipeline |      413.1     |      3.387       |
> |     RP-KrossFuse ( Overhead)     |       0.14      |      1.729       |
>
> Also, a core advantage of our training-free fusion approach is its potential for improved out-of-distribution (OOD) generalization. Since RP-KrossFuse does not rely on training over a specific dataset, it avoids overfitting to a particular data distribution. In contrast, training-based methods often show sensitivity to their training data. For instance, Table 5 of [1, SLIP] demonstrates a 7.2\% performance gap on ImageNet linear probing: 65.4\% when trained on CC3M, 73.7\% on CC12M, and a drop to 66.5\% when trained on YFCC15M. This highlights the dependence of training-based fusion models on the choice of pretraining data.
> We will clarify these points and include the new results in the revised manuscript.
>
>
>
>
> **4- Practical applications of the proposed method**
>
> We thank Reviewer 9JZK for the suggestions on showing the practical application of our proposed embedding fusion. We agree that the application of KrossFuse fusion to VLMs could be highly interesting. In the rebuttal period, we did not manage to finish the visual instruction tuning of LLAVA-v1.5-7B. However, we performed another experiment on the application of the framework in guided diffusion models as explained in the following paragraph.
>
> To demonstrate the value of the KrossFuse fusion in conditional image generation, we used it to adapt and improve the Vendi score diversity guidance proposed in [2]. The Vendi score guidance method guides the reverse diffusion process using the Vendi diversity score of the generated data. In the original implementation in [3], the authors used the CLIP embedding for computing the Vendi score in diversity guidance, which we updated to the Kronecker product of DINOv2 and CLIP.
>
> In the experiment, we generated ImageNet samples using a class-conditional Diffusion Transformer (DiT-XL/2) with both the CLIP-based Vendi score guidance from [2], using CLIP as the feature extractor, and the KrossFuse embedding (CLIP fused with DINOv2). We found that the diversity of the generated samples, measured with Recall [3] and Coverage [4] scores, increased with the KrossFuse embedding, while the quality scores Precision [3] and Density [4] also improved over the original CLIP-based Vendi score guidance. We will include this experiment in the received manuscript.
>
>
> | Diversity Guidance method       | Precision | Recall | Density | Coverage |
> |---------------------------------|-----------|--------|---------|----------|
> | c-VSG Guidance (CLIP)        | 0.913     | 0.413  | 1.206   | 0.552    |
> | KrossFuse (CLIP and DINOv2) | 0.932  | 0.484  | 1.252   | 0.613    |
>
> References:
>
> [1] Mu et al. Slip: Self-supervision meets language-image pre-training. ECCV 2022.
>
> [2] Askari et al. Improving geo-diversity of generated images with contextualized vendi score guidance. ECCV 2024.
>
> [3] Kynkäänniemi et al. Improved precision and recall metric for assessing generative models. NuerIPS 2019.
>
> [4] Naeem et al. Reliable fidelity and diversity metrics for generative models. ICML 2020.

---

### Official Review · Reviewer_tf3Z · 2025-07-01

**Clarity:** 2
**Significance:** 3
**Originality:** 3
**Rating:** 5
**Confidence:** 3

**Summary:**

The paper proposes a novel approach for fusing uni-modal and multi-modal representations by integrating the kernel properties of the Kronecker product into the fused representation. To mitigate the multiplicatively growing dimensionality, the paper approximates the Kronecker product of the embeddings with the Hadamard product on randomly up-projected representations. By utilizing a random matrix with identity covariance for random projection, the paper derives a probabilistic upper bound for the proposed approximation. Technical claims are supported by empirical evaluations and ablations on classification tasks, showing that the proposed embeddings preserving the classification abilities of the uni-modal embeddings while retaining the zero-shot abilities of the multi-modal embeddings.

**Questions:**

1. How would the fused representation compare to the unimodal representation under the same kernel transformation? For example, how would the fused representation compare to the L2-normalized DINOv2 representation in the linear probing experiment?
2. Are any transformations applied to shift the range of the kernels to positive values? If not, would doing so benefit the linear or cosine kernels?
3. Are the comparisons done on an equal dimensionality footing of the fused embedding space? If not, what are the results if the fused dimensionality is equal to the uni- or multi-modal embeddings?
4. The experiments mainly focus on self-supervised unimodal representations. Would one expect to observe a similar gain in accuracy when fusion is performed with a supervised unimodal representation?
5. Can you comment on how the modality gap is affected by the proposed fusion?

**Ethical Concerns:**

["NO or VERY MINOR ethics concerns only"]

**Final Justification:**

The answers to my concerns as well as those of the other reviewers have clarified a number of important points in a concise fashion. From my perspective, with these additions as well as the added experiments, no significant concerns remain, hence I am upgrading my recommendation to accept. I believe this paper would be a valuable addition to the literature.

**Limitations:**

As mentioned in the "Major weaknesses," there is no dedicated "Limitations" section. As limitations, one could mention (i) introducing additional hyperparameters that need tuning, such as $l$ and $C$, (ii) experiments being limited to image classification tasks, and (iii) inconsistency in gain of accuracy compared to the stronger fused component.

**Paper Formatting Concerns:**

The supplemental material contains a full version of the main paper in addition to the appendix. Other conferences (e.g., CVPR) disallow this, hence I am not sure whether this is permitted by the NeurIPS regulations. A scan of the guidelines did not seem to turn up anything concrete. In any case, I ignored the first 9 pages of the supplemental.

**Quality:**

2

**Strengths And Weaknesses:**

Strengths:
* The method addresses a problem of potentially high significance. Fusing representations of varying modalities, especially those of foundational models, could impact a broad range of applications.
* The lead-up to the RP-KrossFuse method is logically coherent. The design choices are well-motivated, and the extension via random projections is skillfully integrated.
* The empirical results demonstrate improvement over individual components and simple, alternative fusion strategies, including ensemble and attention-based methods.

Major weaknesses:
* The clarity of the text can be improved. Tables 1 and 2 should include citations for the previous methods. For Tables 1 and 2 and Figures 4 and 5, the captions lack sufficient detail about the experimental setup. For example, the figures omit information about which dataset is used. Additionally, lines 265–266 lack citations for the baseline methods and provide only a brief explanation, making it difficult to understand them without consulting the appendix. Similarly, line 225 mentions a range for the random matrix but does not clarify why it was chosen and can only be understood by referring to the appendix. The notation $\psi_{1,2,X}$ etc. in Theorem 1 is apparently not defined, etc. The main text should be self-contained and clear without requiring the reader to refer to the appendix for key details. Other clarity issues: The notion of "properly aligned" in line 153 remains vague. Why is the embedding in Eqs. 4 and 5 suddenly E when Greek letters were used before.
* Lines 49–51 build the motivation for using the Kronecker product as a fusion method. The claim is that if the fused kernel is the product of the individual component kernels, then the fused representation will identify two samples as similar only if they are deemed similar by both component kernels, which would lead to better class separation. This assumption only holds if the range of the kernel functions is non-negative. For example, with cosine or linear kernels, if both individual component kernels output negative values near their lower bound, indicating dissimilarity in both components, the Kronecker product would yield a high similarity score, contradicting the original motivation. Furthermore, Figure 5c shows that the cosine kernel, which violates the assumption, outperforms the Gaussian kernel, for which the assumption holds.
* I may have missed this, but it seems like the value of l, i.e. the dimensionality of the projected features is not specified for the experimental comparison. Hence it is unsure if we are really comparing apples and apples here, in the sense that the projected, fused embeddings may not have the same dimensionality as the uni-modal ones that are being compared to. Unless I overlooked something, e.g., the CLIP and DINOv2 variants considered have differing embedding dimensions so RP-KrossFuse cannot be fairly compared against both. A comparison on an equal dimensionality footing seems quite essential, however.
* Given that the modality gap is a known issue, as the paper discusses in the appendix (Line 630f), I am surprised that the paper does not assess this issue in more detail, e.g. to clarify whether the modality gap is affected by the proposed fusion strategy.
* According to the checklist, the appendix is said to include a dedicated "Limitations" section. However, no such section is present. Further, "Section 7: Limitations and Conclusions" does not address the limitations of the work.

Minor weaknesses:
* Line 518 defines an upper bound for the absolute value of the dot product. However, in the equation following Line 519, the bound is applied to the dot product itself, not its absolute value. Moreover, the denominator of the Hoeffding inequality's exponent should be a function of $2B$, and not $B$, as the dot product would range between $[-B, B]$. Additionally, it is unclear why the denominator of the exponent in the Hoeffding equation is not squared.
* During fusion, the method introduces additional nonlinearities in the form of kernels. It is unclear how the fused representation compares to unimodal representations with the same nonlinearity applied.
* The experiments are limited to classification tasks. It remains unclear how the fused representation would generalize to dense prediction tasks such as semantic segmentation.
* While the fused representation yields robust accuracy across different datasets, the gain over the stronger fused component is not consistent.
* The paper is a bit sloppily written in various places. There are a fair number of typos ("Kroncker", "simialrty", ...) and the "S-RoBERTa" approach is capitalized in numerous different ways.
* The answer to question 7 in the checklist is not convincing. Averaging the results is *not* a form of establishing statistical significance. Similarly, the answers to questions 9 and 10 are not convincing.

---

> ### Author Rebuttal · Authors · 2025-07-31
>
> We would like to thank Reviewer tf3Z for the constructive and thoughtful feedback on our work. We are glad that the reviewer finds our work to address “a problem of potentially high significance” and the design choices “well-motivated” . Below is our response and clarifications regarding the questions and comments in the review:
>
> **1- Writing improvements**
>
> We thank Reviewer tf3Z for pointing out the needed citations and information in Tables 1,2 and the captions of Figures 4,5. We would like to clarify that the notation $\psi_{1,2}$ denotes the KrossFuse embedding of embeddings $\psi_{1}$ and $\psi_2$ which we will state in Theorem 1. The term “properly aligned” refers to how alignment is supposed to be measured in a cross-modal embedding (e.g. via cosine similarity or Euclidean distance). We will make the term clearer in the revision.
>
> **2- Range of kernel values in the KrossFuse fusion**
>
> We appreciate the reviewer’s comment on the potential negative values in the kernel inner products and whether they fit into our argument for the product of kernel functions. As already pointed out by the reviewer, this concern does not apply to the Gaussian (RBF) kernel, which always takes positive values. We note that in all our clustering experiments, we have used the Gaussian kernel, which ensures non-negative values.
>
> In the case of the cosine similarity kernel, the kernel inner product may indeed take negative values. However, we note that for both CLIP and OpenCLIP, the cosine similarity is always non-negative on MS-COCO and ImageNet samples (as also shown in reference [1]). On the other hand, the cosine similarity for DINOv2 can take negative values on ImageNet samples.
>
> The question raised by the reviewer can be viewed as follows: What is the interpretation of the product kernel fusion when kernel similarity scores take negative values? Our answer is that in such cases, the product kernel fusion distinguishes sample types based on the **orthogonality** of their embedded vectors. Please note that we do not claim that this is the universally correct way to define similarity for all embeddings, but rather that it is a natural consequence of fusing kernels via multiplication. As such, the effectiveness of the product kernel fusion depends on whether the orthogonality condition is proper for the fused embeddings, i.e. should clustering within the embedding space be performed based on the orthogonality of the vectors? If an embedding has such a property, then applying the framework with cosine similarity can improve the numerical results. If this is not the case, the framework can instead be used with Gaussian, Laplace, or (normalized) even-degree polynomial kernels, all of which yield positive values bounded by 1, where our intuition on similarity score will directly apply.
>
> **3- The value of random projection dimensionality $l$ in the experiment**
>
> We would like to clarify that we have used dimension $l=3000$ for all the datasets except $l=5000$ in the case of ImageNet. We will make this clear in the revision.
>
> **4- Fairness of RP-KrossFuse comparison with embedding of different dimensions**
>
> If we have correctly understood the reviewer’s comment, it concerns the fairness of comparing embeddings with different dimensions. We respectfully disagree with the notion that such comparisons are inherently unfair. For example, DINOv2 and CLIP already have different embedding dimensions (768 for DINOv2 and 512 for CLIP), yet their comparison is widely accepted in the literature. Following the same logic, it would be inconsistent to deem a comparison between RP-KrossFuse and these individual embeddings as unfair solely due to dimensional differences.
>
> To further clarify our motivation for using random projection in RP-KrossFuse, our goal is to reduce the high dimensionality of the original Kronecker fusion (nearly 800,000 for CLIP and DINOv2) only for the sake of computational efficiency in downstream applications. If enough compute and memory were available, we would even prefer to use the full KrossFuse representation over the projected one. As highlighted in our ablation studies, a lightweight random projection with computational cost $O((d_1 + d_2)l)$ and projection dimension $l = 5000$ is sufficient to preserve the performance of full KrossFuse in the ImageNet case.
>
>
> Continuing our response, if we are constrained to reduce the fused embedding to a bounded dimension, e.g., $l = 512$ to match the lower of the original embedding sizes, then using a pure random projection may not improve the signal-to-noise ratio (SNR). Under such dimensional constraints required by the reviewer's fairness criterion, we would instead apply PCA dimensionality reduction, which provides better SNR at the cost of increased computational cost. Of course, performing PCA directly on the full Kronecker fusion (size $\approx$800,000) is computationally prohibitive. Therefore, in our rebuttal experiments, we applied PCA to reduce the RP-KrossFuse embedding (from 5000 to 512 dimensions) using ImageNet training data. The resulting linear-probe accuracy was 83.7% for the 512-dimensional PCA-reduced embedding, compared to 84.1% for the original RP-KrossFuse at 5000 dimensions. For reference, the linear-probe accuracies of the original CLIP and DINOv2 image encoders are 73.2% and 83.3%, respectively.
>
> To summarize, we apply random projection in RP-KrossFuse solely to reduce computational cost for downstream tasks. We do not believe that exact dimension matching is a strict requirement for fair comparison between embeddings. However, if dimensionality parity is required, we recommend using PCA for compression, as it offers improved SNR, although at a higher computational cost. We will include this discussion in the revised manuscript.
>
> **5- Modality Gap for the fused embedding vs. original CLIP embeddings**
>
> To address the reviewer’s comment, we have measured the modality gaps (as defined in [1]) of the fused embeddings over the MS-COCO dataset. We made the measurement with our used RP-KrossFuse fused embedding with dimension $5000$. The results are in the following table.
>
>
> | Method            | Modality Gap |
> |-------------------|:------------:|
> | CLIP              | 0.875 |
> | RP-KrossFuse      | 0.853 |
>
>
> **6- Limitations**
>
> To address the reviewer’s comment, we will discuss the limitations of the KrossFuse framework in more detail, including the potential cross-validation needed for parameter $C$ and $l$. However, we would like to highlight the following results in the Appendix: the text-image retrieval and clustering results,  the text classification results, that extend the evaluation beyond only the image classification task. Still, we will discuss the further evaluation of the KrossFuse framework in non-image domains in the limitation section.
>
> **7- Typo in the bound of Theorem 1**
>
> We thank the reviewer for catching the typo in the theorem’s equation regarding the power of $B$. We will correct it in the revision.
>
> **8- Question 1 (Results for $L_2$-normalized DINOv2)**
>
> To answer the reviewer’s question, we tested the performance of the $L_2$-normalized DINOv2 embedding and the numerical results are in the following table (continuing Table 1 in the main text). As seen, the numerical performance on linear-probe classification results are mostly similar between normalized and unnormalized DINOv2.
>
> | Method | ImageNet | GTSRB | SVHN | IN-A | IN-R |
> |--------|:--------:|:-----:|:----:|:----:|:----:|
> | DINOv2 (unnormalized) | 83.3 | 72.5 | 60.5 | 48.5 | 68.8 |
> | Normalized DINOv2 | 81.8 | 73.1 | 59.9 | 47.4 | 68.1 |
> | RP-KrossFuse (DINOv2 + CLIP) | 84.1 | 82.7 | 66.9 | 47.6 | 67.4 |
>
>
> **9- Question 4 (Experiments beyond self-supervised embeddings)**
>
> To address this question, we performed zero-shot and linear-probe classification experiments on ImageNet1K by applying RP-KrossFuse to the ImageNet-22k-supervised-trained image model  DeiT-III [2] .The numerical results are in the following table:
>
> | Model | Linear Probe (%) | Zero-Shot (%) |
> |-------|-----------------|--------------|
> |  DeiT-III | 86.5 | NA |
> | CLIP | 73.2 | 57.9 |
> | RP-KrossFuse ( DeiT-III + CLIP) |  86.9| 57.5  |
>
>
>
> References:
>
> [1] Liang et al. Mind the gap: Understanding the modality gap in multi-modal contrastive representation learning. NeurIPS 2022.
>
> [2] Touvron et al. Deit iii: Revenge of the vit. ECCV 2022.

---

> > ### Comment · Reviewer_tf3Z · 2025-08-07
> >
> > I would like to thank the authors for their rebuttal, which has clarified a number of points. Particularly, answers 2, 3, 5, 8, and 9 were very helpful. Regarding answer 4, I understand that it may not be always possible to compare descriptors with equal dimensionalities. That said, the difference in the number of dimensions between DINOv2 and CLIP is much less (factor 1.5) than that to the proposed RP-CrossFuse (~factor 5). However, the additional PCA experiment helps to show that the fused formulation indeed goes beyond the original embeddings even with a lower dimensionality, so this is very helpful and should be added to the paper.
> >
> > Thank you again for the clarifications. I have no further questions at this point.

---

> > > ### Author Response · Authors · 2025-08-09
> > >
> > > We sincerely thank Reviewer tf3Z for their time and feedback on our response and are glad that our responses could clarify several points discussed in the review. As we mentioned in our response, we will revise the manuscript accordingly to reflect the points and clarifications provided in our responses to this and other reviews. We will specifically discuss the sign of the kernel inner product in the Kronecker product fusion and how the KrossFuse operates based on the orthogonality of the embedded vectors measured by the chosen kernel function (Item 2), and include the numerical result with PCA dimensionality reduction to the lower dimension of the embeddings (Item 4).

---

### Official Review · Reviewer_9JZK · 2025-07-02

**Clarity:** 3
**Significance:** 3
**Originality:** 3
**Rating:** 5
**Confidence:** 3

**Summary:**

This paper proposes a method to effectively combine two modality spaces using the Kronecker product. Specifically, it aims to merge cross-modal embeddings (CLIP) with unimodal embeddings (DINO) to leverage the advantages of both methods. To do this, the paper introduces Kronecker fusion to incorporate the unimodal embedding's missing modality. It also proposes using Random Projection to reduce the  computational cost of the Kronecker product space. The proposed method demonstrated highly effective performance improvements, outperforming both CLIP and DINO on various downstream tasks.

**Questions:**

Are there other domains where these integrated embeddings could be highly effective? For example, could Vision-Language Models (VLMs) or text-to-image diffusion models achieve better performance by using these integrated embeddings?

**Ethical Concerns:**

["NO or VERY MINOR ethics concerns only"]

**Final Justification:**

I think this is a good paper that shows consistent performance gains across many uni- and multi-modal scenarios. While there are some worries about the efficiency of this method, I think it is not a strong drawback.

**Limitations:**

Yes

**Quality:**

3

**Strengths And Weaknesses:**

Strength
- The paper is well-written and easy to understand.
- The proposed methods are sound, and practical scalability is well-considered.
- Experimental results show effectiveness of proposed method on various downstream tasks

Weakness
- Further experiments are needed to determine if the method performs well with more recent cross-modal embeddings, such as SigLiP.
- Evaluation on more downstream tasks is required. Please refer to the questions.

---

> ### Author Rebuttal · Authors · 2025-07-31
>
> We would like to thank Reviewer 9JZK for the constructive and thoughtful feedback on our work. We are pleased to hear that the reviewer finds our work “well-written” and the proposed method to be  “sound” . Below is our response and clarifications regarding the questions and comments in the review:
>
> **1- Experiments on the recent SigLIP embedding**
>
> In the Appendix C.6, we have already presented the numerical results of the zero-shot and Linear-probe classification accuracy of SigLIP embeddings on ImageNet. We will move these results to the main text to address the reviewer’s comment. In the rebuttal period, we also ran more experiments on recent embedding models MobileCLIP[1] and OpenVision[2], and the results support the method:
>
> | Model | Linear Probe (%) | Zero-Shot (%) |
> |-------|-----------------|--------------|
> | MobileCLIP | 82.3 | 71.1 |
> | RP-KrossFuse (w/ MobileCLIP) | 84.5 | 70.9 |
> | OpenVision | 79.3 | 65.9 |
> | RP-KrossFuse (w/ OpenVision) | 84.2 | 65.1 |
>
> In the revised paper, we will add these results to the main text.
>
>
> **2- Application of the proposed method on text-to-image diffusion models**
>
> We thank Reviewer 9JZK for the suggestion on showing the practical application of our proposed embedding fusion to diffusion models. To demonstrate the value of the proposed fusion in conditional image generation, we used our approach to adapt and improve the Vendi score diversity guidance proposed in [3]. The Vendi score guidance method guides the reverse diffusion process using the Vendi diversity score of the generated data. In the original implementation in [3], the authors used the CLIP embedding for computing the Vendi score in diversity guidance, which we updated to the Kronecker product of DINOv2 and CLIP.
> In the experiment, we generated ImageNet samples using a class-conditional Diffusion Transformer (DiT-XL/2) with both the CLIP-based Vendi score guidance from [3], using CLIP as the feature extractor, and the KrossFuse embedding (CLIP fused with DINOv2). We found that the diversity of the generated samples, measured with Recall [4] and Coverage [5] scores, increased with the KrossFuse embedding, while the quality scores Precision [4] and Density [5] also improved over the original CLIP-based Vendi score guidance. We will include this experiment in the received manuscript.
>
>
> | Diversity Guidance method       | Precision | Recall | Density | Coverage |
> |---------------------------------|-----------|--------|---------|----------|
> | c-VSG Guidance (CLIP)        | 0.913     | 0.413  | 1.206   | 0.552    |
> | KrossFuse (CLIP and DINOv2) | 0.932  | 0.484  | 1.252   | 0.613    |
>
>
>
>
> References:
>
> [1] Vasu et al. Mobileclip: Fast image-text models through multi-modal reinforced training. CVPR 2024.
>
> [2] Li et al. Openvision: A fully-open, cost-effective family of advanced vision encoders for multimodal learning. arXiv preprint (2025).
>
> [3] Askari et al. Improving geo-diversity of generated images with contextualized vendi score guidance. ECCV 2024.
>
> [4] Kynkäänniemi et al. Improved precision and recall metric for assessing generative models. NuerIPS 2019.
>
> [5] Naeem et al. Reliable fidelity and diversity metrics for generative models. ICML 2020.

---

> > ### Comment · Reviewer_9JZK · 2025-08-06
> > **Thanks for the rebuttal**
> >
> > Thank you for the authors' rebuttal. I especially appreciate their experiments on text-to-image diffusion models. I also have read the responses to the other reviewers' questions, and find no significant drawbacks. I will maintain my score.

---

> > > ### Author Response · Authors · 2025-08-09
> > >
> > > We sincerely thank Reviewer 9JZK for their time and feedback on our response and are glad that the reviewer finds the responses satisfactory. As we mentioned in the response, we will revise the manuscript accordingly to reflect the points and clarifications provided in our responses to this and other reviews, including the application of promoting diversity in text-to-image diffusion sample generation (Item 2 in the response following the reviewer's suggestion).

---

### Official Review · Reviewer_MP5Z · 2025-07-05

**Clarity:** 2
**Significance:** 3
**Originality:** 4
**Rating:** 4
**Confidence:** 2

**Summary:**

This paper introduces RP-KrossFuse, a method to fuse cross-modal embeddings with uni-modal embeddings to transfer the capabilities of both sets of embeddings into one unified set. The key motivation is that cross-modal embeddings enable cross-modal tasks like zero-shot image classification and cross-modal retrieval, however their unimodal capabilities take a hit, whereas uni-modal embeddings excel within their domain and modality, however are not applicable to cross-modal tasks. RP-KrossFuse utilizes random projection-based Kronecker product to integrate cross-modal embeddings with their uni-modal counterparts in an efficient manner. There are several experiments conducted to showcase the benefits of the fused embeddings on both uni-modal and cross-modal tasks.

**Questions:**

- In C.2 there is a learned unitary transformation that maps the text embeddings onto the space of the image embeddings. What are the downstream implications of such a method? Both [Udandarao et al](https://www.mlmi.eng.cam.ac.uk/files/2021-2022_dissertations/understanding_and_fixing_the_modality_gap_in_vision-language_models_reduced.pdf) and [Schrodi et al](https://arxiv.org/pdf/2404.07983) showed that the downstream effects of aligning the embeddings might not be the most effective for improving downstream task performance? There should be a discussion added that addresses the key motivation for performing this unitary transformation if it indeed does not positively affect downstream performance.
- Is there a baseline where the embeddings are directly just concatenated and then there is a learnable projection at the end that is tuned? My concern with the current baselines is that they are all quite involved, and there is no simple "concatenation-only" baseline that has been tested.

**Ethical Concerns:**

["NO or VERY MINOR ethics concerns only"]

**Final Justification:**

The rebuttal addressed all my concerns. Further, upon reading the other reviews, I do not find any significant flaws / drawbacks in the paper. Therefore, I am leaning towards accepting the paper.

**Limitations:**

Yes

**Quality:**

2

**Strengths And Weaknesses:**

Strengths:
- The motivation and outlined problem that the paper tackles is important and well-justified. Enabling representations to perform well across both cross-modal and uni-modal tasks is extremely significant, especially in an era where the community is moving towards unified end-to-end models.
- The proposed KrossFuse and RP-KrossFuse approaches are novel to the best of my knowledge, and introduce an efficient method to fuse embeddings.
- The method is theoretically grounded.

Weaknesses:
- The embedding models used in the paper are somewhat outdated. While there is definitely good signal from using these models like CLIP/OpenCLIP, there are much more advanced and newer vision encoders that have been released, which improve downstream performance using better data and training recipes. Some examples include [AIMv2](https://arxiv.org/pdf/2411.14402), [OpenVision](https://arxiv.org/abs/2505.04601) (although admittedly OpenVision was only released a week prior to the NeurIPS deadline, so I don't expect the authors to use those models for experiments), [MobileCLIP](https://arxiv.org/abs/2311.17049) etc.
- A key important weakness in my eyes regarding the clustering results and the downstream performance results is that they are not directly comparable due to the differences in data distribution that was used for training the CLIP models and the DinoV2 models. See [WebSSL](https://arxiv.org/pdf/2504.01017). Hence, directly making claims about the differences being about cross-modal and uni-model encoders is somewhat flawed in my opinion, since the differences stem not necessarily only from the training recipe but the training data distribution used. I would urge the authors to add a discussion and caveating their claims surrounding the differences.
- The paper does not compare with any methods that directly try to fuse the benefits of cross-modal and uni-modal recipes, at training time. Some exemplar methods include [SLIP](https://arxiv.org/abs/2112.12750), [TIPS](https://arxiv.org/abs/2410.16512), [SILC](https://arxiv.org/abs/2310.13355), [SigLIP-2](https://arxiv.org/abs/2502.14786). The reason those methods might be more beneficial and preferred at inference time is due to the inference time cost: for any new samples, running the RP-KrossFuse approach incurs extra cost since it requires forwarding through two encoder models, as compared to just forwarding through one model in the case of these approaches.

---

> ### Author Rebuttal · Authors · 2025-07-31
>
> We would like to thank Reviewer MP5Z for the constructive and thoughtful feedback on our work. We are glad to hear that the reviewer finds our work’s motivation “well-justified” and our proposed method to be “theoretically grounded” . Below is our response and clarifications regarding the questions and comments in the review:
>
> **1- The method’s application to recent embeddings models**
>
> We appreciate the reviewer’s suggestion regarding the application of our fusion method to more recent embedding models. We would like to emphasize that RP-KrossFuse is a general method that can be applied to fuse any combination of unimodal and cross-modal embeddings. Our focus on CLIP and its variants is due to CLIP’s widespread adoption and integration in many AI and machine learning frameworks.
>
> Having said that, we agree that evaluating the method on newer cross-modal embeddings is valuable. In fact, we have already included results on the more recent SigLIP model in Appendix C.6 as part of our ablation studies.
> Additionally, during the rebuttal period, we conducted further experiments on the reviewer’s suggested models, OpenVision and MobileCLIP. Below, we report the Linear-Probe and Zero-Shot classification results on ImageNet. These results also support the effectiveness of RP-KrossFuse when applied to more recent embedding architectures. We will include these results in the revised paper.
>
> | Model | Linear Probe (%) | Zero-Shot (%) |
> |-------|-----------------|--------------|
> | MobileCLIP | 82.3 | 71.1 |
> | RP-KrossFuse (DINOv2 + MobileCLIP) | 84.5 | 70.9 |
> | OpenVision | 79.3 | 65.9 |
> | RP-KrossFuse (DINOv2 +OpenVision) | 84.2 | 65.1 |
>
>
> **2- Role of different training data distributions of the embedding models in their Kronecker product fusion**
>
> We fully understand the reviewer’s point regarding the differing training data distributions of CLIP and DINOv2, which naturally result in different behaviors when clustering or distinguishing between samples. In fact, this observation results in a key motivation behind our proposal to fuse these embeddings using the Kronecker product.
> As discussed in the introduction, the kernel similarity score of our fused representation is the product of the individual similarity scores from the two embeddings. This means that two samples will be distinguishable in the fused embedding space as long as either CLIP or DINOv2 assigns them sufficiently small similarity scores.
>
> Given that CLIP and DINOv2 are trained on different datasets, it is expected that they will specialize in distinguishing different semantic or visual concepts. This diversity enables the Kronecker fusion to effectively combine their complementary strengths. For example, Figure 1 in the paper shows that CLIP is more effective at separating traffic sign categories, while DINOv2 performs better at differentiating between dog breeds. When we apply Kronecker fusion, the resulting representation could successfully distinguish both traffic signs and dog breeds, which neither model could fully achieve on its own.
>
> Therefore, we argue that the difference in training data distributions between the embedding models is not a limitation, but rather could be a motivation for fusion. The Kronecker product allows us to construct a fused embedding that inherits the union of the discriminative capabilities of the models. We will revise the introduction to make this intuition more clear, and we also kindly refer the reviewer to Figures 8 and 10 in the Appendix, which provide additional experiments supporting this point across more diverse sample types.
>
> **3- Comparison with the training-based fusion methods**
>
> We appreciate the reviewer’s comment regarding the comparison with training-based fusion approaches. To clarify, the baselines GATE and ATTN included in our study are indeed training-based fusion methods. However, we understand that the reviewer’s comment refers to a training-time integration, retraining one embedding model from scratch while conditioning on the other, potentially enabling lower inference-time complexity. We would like to raise two key points in response:
>
> First, regarding inference efficiency, it is important to note that, assuming sufficient memory, our method allows for the two embedding models to operate in parallel, thereby avoiding increased inference latency. To support this, we conducted benchmarking experiments on an NVIDIA RTX 3090 GPU with a batch size of 128. The individual CLIP and DINOv2 models required 73.6 ms and 392.1 ms per batch, respectively. Our parallel RP-KrossFuse pipeline achieved a total inference time of 413.1 ms, which is only slightly higher than the slower of the two models. Notably, the fusion step itself adds just 0.14 ms of overhead.
>
> |                 Method                | Time/Batch (ms) | Peak Memory (GB) |
> |-----|---------------|-------|
> |                  CLIP                 |      73.6      |      2.077       |
> |                 DINOv2                |      392.1     |      3.199       |
> | Parallel RP-KrossFuse Pipeline |      413.1     |      3.387       |
> |     RP-KrossFuse ( Overhead)     |       0.14      |      1.729       |
>
> Second, a core advantage of our training-free fusion approach is its potential for improved out-of-distribution (OOD) generalization. Since RP-KrossFuse does not rely on training over a specific dataset, it avoids overfitting to a particular data distribution. In contrast, training-based methods often show sensitivity to their training data. For instance, Table 5 of [1, SLIP] demonstrates a 7.2\% performance gap on ImageNet linear probing: 65.4\% when trained on CC3M, 73.7\% on CC12M, and a drop to 66.5\% when trained on YFCC15M. This highlights the dependence of training-based fusion models on the choice of pretraining data.
>
> We will clarify these points and include the new results in the text.
>
> **4- Unitary transformation in Appendix C.2 experiments**
>
> We would like to clarify that the unitary transformation mentioned in Appendix C.2 was applied solely for the cross-modal clustering experiments reported in that subsection. None of the experimental results presented in the main text or other parts of the appendix involve this transformation.
>
> The motivation for applying a unitary transformation in Appendix C.2 is due to the known “modality gap” between the image and text embedding spaces of CLIP, as discussed in previous work [2]. To enable meaningful cross-modal clustering via spectral clustering, it was necessary to learn and correct this modality misalignment. The unitary transformation serves this purpose by aligning the modalities in a shared space before clustering.
>
> We emphasize that this step is limited to the cross-modal clustering setting and was not used in any of the other experiments in the paper. We will make this more clear in the revised paper.
>
> **5- "concatenation-only" baseline**
>
> We thank the reviewer for raising the point about the concatenation-only baseline. Theoretically, concatenating two embeddings results in a kernel similarity function that corresponds to the *sum* of the individual kernels. This implies that two samples will have high similarity in the fused space only if both original embeddings deem them similar, effectively modeling the *intersection* of the embeddings’ concepts. In contrast, our proposed Kronecker fusion leads to a *product* of the kernel similarities, which enables distinguishing two samples as long as either embedding distinguishes them, modeling the union of their concepts.
>
> In terms of numerical performance, we note that the concatenation baseline is expected to perform comparably to the stronger of the individual embeddings in supervised classification settings. This is because the concatenated vector retains all entries from both embeddings, and the limited VC dimension of the linear probe helps prevent overfitting.
>
> However, in unsupervised or weakly supervised scenarios such as clustering, the product-based Kronecker fusion can better capture the complementary structure from both embeddings. This often results in improved performance, as it reflects broader concept coverage. To empirically validate this, we conducted several experiments comparing the concatenation baseline with RP-KrossFuse across various tasks:
>
> 1. For supervised learning tasks like ImageNet linear probing, both methods perform comparably (concatenation: 83.6%, RP-KrossFuse: 84.1%), which is theoretically expected.
> 2. In multimodal classification tasks, RP-KrossFuse shows clear advantages over the concatenation on  MVSA dataset [3] (Sentiment Analysis on Multi-view Social Data):
>
>    - MVSA-Single: 74.3% vs 71.3%
>    - MVSA-Multiple: 66.6% vs 63.7%
>
> 3. Most notably, in unsupervised clustering tasks, RP-KrossFuse significantly outperforms concatenation across the datasets. We have measured the clustering score, Adjusted Rand Index, Adjusted and Normalized Mutual Information (ARI, AMI and NMI), and the results are in the following table.
>
> | Method        | Metric | Flowers102 | DTD   | ImageNet-Dogs | GTSRB | Typo-Attacked ImageNet |
> |--|---|---|--|----|--|------|
> | Concatenation | NMI    | 98.4       | 60.5  | 76.3          | 43.8   | 44.4                 |
> | RP-KrossFuse  | NMI    | 99.1       | 62.9  | 88.3          | 50.0   | 87.4                 |
> | Concatenation | AMI    | 97.9       | 57.9  | 76.1          | 40.2   | 43.9                 |
> | RP-KrossFuse  | AMI    | 98.8       | 60.4  | 88.2          | 46.7   | 87.3                 |
> | Concatenation | ARI    | 93.9       | 35.3  | 64.9          | 14.1   | 28.4                 |
> | RP-KrossFuse  | ARI    | 97.0       | 36.4  | 86.3          | 19.5   | 79.6                 |
>
> References:
>
> [1] Mu et al. Slip: Self-supervision meets language-image pre-training. ECCV 2022.
>
> [2] Liang et al. Mind the gap: Understanding the modality gap in multi-modal contrastive representation learning. NeurIPS 2022.
>
> [3] Niu et al. Sentiment analysis on multi-view social data. MMM 2016.

---

> > ### Comment · Reviewer_MP5Z · 2025-08-08
> > **Response**
> >
> > Thanks to the authors for the clarifying points. Overall, the rebuttal sufficiently addresses most of my concerns. The one remaining point that I would stress on is regarding point 2: Role of different training data distributions of the embedding models in their Kronecker product fusion. While the authors' arguments in this rebuttal text is convincing, I think is is extremely important that these points are conveyed in the main text so that readers do not get an incomplete take-away message. Hence, I would urge the authors to add this discussion back into the main text.
> > Given the rebuttal and the other reviews, I am happy to raise my score.

---

> > > ### Author Response · Authors · 2025-08-09
> > >
> > > We sincerely thank Reviewer MP5Z for their time and feedback on our response and are glad that our responses could address the reviewer’s concerns. As we mentioned in our response, we will revise the manuscript accordingly to reflect the discussion on the role of embeddings’ different training data in motivating the Kronecker product fusion (Item 2 in the response) and the other points and clarifications in response to the reviewer’s feedback.

---

### Official Review · Reviewer_kcnR · 2025-07-07

**Clarity:** 3
**Significance:** 3
**Originality:** 3
**Rating:** 5
**Confidence:** 4

**Summary:**

This paper introduces a method to integrate cross-modal embedding with modality-specific embeddings called RP-KrossFuse. The main idea is to combine the embeddings using Kronecker product, so that the sample-pairwise similarity scores of the fused embeddings is the product of cross-modal embedding similarity and single-modal embedding similarity. This property allows the fused embeddings to form fine-grained clusters that inherit the discriminative information of both parent embeddings. Considered the extremely high dimension of the Kronecker product, the authors further proposes the application of random projection to lower the dimension of the fused embedding while preserving the similarity relationships. Theorems are provided to show that random projected features can preserve such relationship with high probability. Extensive experiments are conducted to test the fused embedding, including image classification, text classification, zero-shot image-text alignment and cross-modal few-shot learning. The results indicate that the fused embedding combined the strength of cross-modal embedding and modality-specific embedding.

**Questions:**

What is the implementation of the kernel map $\phi$?

**Ethical Concerns:**

["NO or VERY MINOR ethics concerns only"]

**Limitations:**

yes

**Paper Formatting Concerns:**

No formatting issues.

**Quality:**

3

**Strengths And Weaknesses:**

Strengths:
1. Combining representations from different models is an interesting question with broad potential applications. The paper provide an effective solution to this problem. The training-free nature of the proposed method improves its utility in real-world scenarios.
2. The theoretical argument that randomly projected features preserve sample-wise similarity relationships is a strong foundation for the proposed method.
3. The experiments are extensive with acceptable results. Despite that the fused embedding performs occasionally underperforms either cross-modal embedding or modality-specific embedding on certain tasks, the overall results indicate that the fused embedding is generally better than any of the individual representations.
4. The paper is written in a decent way, having good structure and being easy to follow. Although the "Numerical Results" section has a minor issue (see Weakness 1), the execution is otherwise solid.

Weaknesses:
1. In section 6 (Numerical Results), 'see appendix' appears too frequently. Introducing the experiment without showing the result in the main paper affects reading experience. The author should carefully decide which result to be the most important and to be presented in the main text.
2. The authors should report the actual dimension $l$ they choose in implementation, as Figure 5 (d) clearly shows that higher dimension leads to better performance. On one hand, the dimension of the fused embedding determines the computational cost and the efficiency of the method. On the other hand, a larger dimension used in the fused embedding may raise concerns regarding whether certain comparison is fair. For instance, a larger linear layer with more parameters could be applied in the linear probing experiments, leading to better performance compared to training a small linear layer on the CLIP embedding.

---

> ### Author Rebuttal · Authors · 2025-07-31
>
> We would like to thank Reviewer kcnR for the constructive and thoughtful feedback on our work. We are delighted that the reviewer finds our work to provide “an effective solution” and “written in a decent way”. Below is our response and clarifications regarding the questions and comments in the review:
>
> **1- Referrals to the Appendix in the main text**
>
> We thank the reviewer for pointing this out. In the revision, we will keep the more important numerical results on the zero and few shot classification and clustering performance in the main text and reduce the Appendix referrals to only one time at the end of the numerical results section.
>
> **2- The role of random projection dimension $l$ in the RP-KrossFuse method**
>
> We appreciate the reviewer’s comment regarding the importance of the random projection dimension $l$ in the RP-KrossFuse method. We would like to clarify that we have used dimension $l=3000$ for all datasets except $l=5000$ in the case of ImageNet. In the revised version of the paper, we will clearly state the value of $l$ alongside the tables and figures.
>
> To further explain the role and selection of dimension $l$, we note that a key challenge of the Kronecker product fusion is the resulting high dimensionality: the fused representation has size $2d_1d_2$ for embeddings of dimensions $d_1$ and $d_2$. For example, combining CLIP and DINOv2 embeddings leads to a fused dimension close to 800,000 ($2 \times 512 \times 768$).
>
> To address this, we introduce random projection in RP-KrossFuse, which is *not to ensure fairness* in comparison with the original embeddings, *but to reduce the computational overhead* to meet the computational budget. If computationally feasible, we would even prefer to use the full Kronecker product without random projection. However, this would incur significant memory and compute costs.
>
> Also, we highlight that our proposed RP-KrossFuse scheme reduces the computational complexity from $O(d_1d_2 l)$ (naive random projection) to $O((d_1 + d_2)l)$, enabling more efficient fusion. To empirically validate this efficiency gain, we conducted benchmarking experiments on an NVIDIA RTX 3090 GPU with a batch size of 128. The original KrossFuse pipeline required 491.3 ms per batch and 18.406 GB of peak memory, while RP-KrossFuse (with $l = 5000$) achieved comparable performance with just 413.1 ms per batch and a significantly reduced memory footprint of 3.39 GB. Notably, the fusion step’s overhead was reduced from 22.0 ms to only 0.14 ms. For reference, the baseline CLIP and DINOv2 models individually required 73.6 ms and 392.1 ms per batch, respectively.
>
>
> |                 Method                | Time/Batch (ms) | Peak Memory (GB) |
> |---------------------------------------|-----------------|------------------|
> |                  CLIP                 |      73.6      |      2.077       |
> |                 DINOv2                |      392.1     |      3.199       |
> | KrossFuse Pipeline |      491.3     |      18.406      |
> | KrossFuse (Overhead) |      22.0      |      17.203      |
> | Parallel RP-KrossFuse Pipeline |      413.1     |      3.387       |
> |     RP-KrossFuse (Overhead)     |       0.14      |      1.729       |
>
>
> **3-** *“a larger linear layer with more parameters could be applied in the linear probing experiments, leading to better performance compared to training a small linear layer on the CLIP embedding”*
>
> We would like to clarify in the linear probe classification experiments, we did not use any non-linear activation functions applied to the random projection features. Therefore, in the comparison with the CLIP model, even if we first apply a linear transformation (denoted by $g_w$) to transform the CLIP features to a higher dimension and then apply the linear probe (denoted by $f_{\theta}$) to the linearly-transformed features, the composition $g_w \circ f_{\theta}$ still remains to be a linear function of the CLIP features, and therefore the linear-probe classification accuracy will not improve over applying it directly to the CLIP embedding.
>
>
>
> **4- Question on the kernel map $\phi$**
>
> We thank the reviewer for their question regarding the kernel map $\phi$. In our experiments, we evaluated two kernel functions:
> 1) The normalized linear (cosine similarity) kernel, implemented as $\phi(x) = \frac{x}{\Vert x \Vert_2}$. This implementation is straightforward and directly normalizes the input embeddings.
> 2) The Gaussian (RBF) kernel, for which we used random Fourier features as described in Appendix Section A.4.
> We would like to emphasize that our implementation of the Gaussian kernel map $\phi$ does not simply apply RP-KrossFuse to a low-dimensional vector of precomputed random Fourier features. Instead, we designed the implementation to unify the generation of random Fourier features and the random projection in a single step. This unified approach effectively simulates operating in a feature space as if the number of Fourier features were infinite, while still being computationally efficient.
> This design allows RP-KrossFuse to retain the expressivity of the Gaussian kernel without incurring the cost typically associated with large numbers of explicit random features.

---

> > ### Comment · Reviewer_kcnR · 2025-08-08
> >
> > I thank the authors for the detailed and thoughtful responses to my comments. Their clarifications regarding the random projection technique and the implementation of the two kernel maps have effectively addressed my concerns. At this stage, I have no further questions, but I encourage the authors to incorporate the promised refinements into the manuscript. Overall, I find the paper’s originality and quality to meet the conference’s acceptance standards, and I maintain my initial recommendation.

---

> > > ### Author Response · Authors · 2025-08-09
> > >
> > > We sincerely thank Reviewer kcnR for their time and feedback on our response and are glad that our responses could address the concerns. As we mentioned in our response, we will revise the manuscript accordingly to reflect the points and clarifications provided in our responses to the comments raised in this and other reviews.

---

### Comment · Area_Chair_Vtnb · 2025-08-06
**Discussion Required for This Paper**

Dear all,
The authors have submitted their rebuttal. We would appreciate it if you could kindly review their response ASAP and let us know if it affects your assessment or if you have any additional comments. Your input is greatly valued in the decision process. Thank you again for your time and contribution.
Best,
AC

---

### Note · Authors · 2025-08-13

We would like to thank the reviewers for their insightful feedback on our work and rebuttal. Below, we summarize the key points we aimed to highlight during the discussion with the reviewers:

1. **Motivation behind Kronecker product fusion** As noted by the reviewers, state-of-the-art cross-modal and unimodal embeddings are trained on different datasets. We emphasized that this is indeed a motivation for using Kronecker product fusion, as it multiplies the kernel similarity scores of the embeddings and can inherit the cluster structures of the fused embedding maps. We will make this point clear in the revised draft.
2. **Complexity of random projection for dimensionality reduction in RP-KrossFuse** We highlighted the affordable computational complexity of random projection to dimension $l$ in the RP-KrossFuse method, which is $\mathcal{O}\bigl((d_1 + d_2)l\bigr)$, compared to a naive random projection with cost $\mathcal{O}\bigl(d_1 d_2 l\bigr)$. We also referred to the discussion in Appendix A.4 on how this result extends to shift-invariant Gaussian kernels.
3. **Choice of reduced dimension $l$ in RP-KrossFuse** We clarified that the goal of random projection to $l = 5000$ (ImageNet case) is not to ensure fairness in comparison with the baselines, but rather to reduce the significant computational cost of the full Kronecker product in KrossFuse (dimension 800,000) *without compromising the classification performance for moderate values* $l \approx 5000$. If dimensional parity is required, a further reduction from RP-KrossFuse (with $l = 5000$) can be performed using PCA to achieve a higher SNR and preserve the downstream classification scores of KrossFuse.
4. **Performance gains over the concatenation baseline** We highlighted the improvements in unsupervised and weakly supervised learning tasks achieved by Kronecker product fusion compared to the simple concatenation of two embeddings. This improvement is largely due to the multiplication of kernel similarity scores, which better preserves cluster structures than the summation of kernel similarity scores resulting from concatenation.
5. **Application of diversity guidance in diffusion models** We discussed the application of Kronecker product fusion to diversity guidance in image diffusion models. According to our results, using kernel-based Vendi diversity scores computed in the Kronecker product space of CLIP + DINOv2 outperforms the baseline method operating with the CLIP-based Vendi diversity score.

---

### Decision · Program_Chairs · 2025-09-17

**Decision:**

Accept (poster)

**Comment:**

This paper proposes a method for effectively combining two modality spaces using the Kronecker product. The authors introduce Kronecker fusion to address the missing modality in unimodal embeddings and propose leveraging Random Projection to reduce the computational overhead of the Kronecker product space. During the discussion, reviewers acknowledged that the design choices are well-motivated, the method addresses a problem of potentially high significance, and the experimental results demonstrate consistent performance gains across numerous uni- and multi-modal scenarios. The primary concerns raised by reviewers included the motivation behind Kronecker product fusion, the complexity of random projection for dimensionality reduction in RP-KrossFuse, and the choice of reduced dimension in RP-KrossFuse. The authors’ rebuttal effectively addressed most of these concerns, leading to unanimous acceptance by the end of the discussion. Consequently, the Area Chair (AC) recommends acceptance.